# In vivo CRISPR screens reveal Serpinb9 and Adam2 as regulators of immune therapy response in lung cancer

Dzana Dervovic[1], Ahmad A. Malik [1,2], Edward L. Y. Chen[1], Masahiro Narimatsu[1], Nina Adler [3], Somaieh Afiuni-Zadeh[1], Dagmar Krenbek[4], Sebastien Martinez[1], Ricky Tsai [1], Jonathan Boucher[5], Jacob M. Berman[1], Katie Teng[1,2], Arshad Ayyaz[1,6], YiQing Lü [1,2], Geraldine Mbamalu[1], Sampath K. Loganathan [1,7], Jongbok Lee[8], Li Zhang[8,9], Cynthia Guidos [10], Jeffrey Wrana [1,2], Arschang Valipour[11], Philippe P. Roux [5,12], Jüri Reimand[2,3], Hartland W. Jackson [1,2] & Daniel Schramek [1,2] ✉

How the genetic landscape governs a tumor's response to immunotherapy remains poorly understood. To assess the immune-modulatory capabilities of 573 genes associated with altered cytotoxicity in human cancers, here we perform CRISPR/Cas9 screens directly in mouse lung cancer models. We recover the known immune evasion factors Stat1 and Serpinb9 and identify the cancer testis antigen Adam2 as an immune modulator, whose expression is induced by Kras[G12D] and further elevated by immunotherapy. Using loss- and gain-of-function experiments, we show that ADAM2 functions as an oncogene by restraining interferon and TNF cytokine signaling causing reduced presentation of tumor-associated antigens. ADAM2 also restricts expression of the immune checkpoint inhibitors PDL1, LAG3, TIGIT and TIM3 in the tumor microenvironment, which might explain why ex vivo expanded and adoptively transferred cytotoxic T-cells show enhanced cytotoxic efficacy in ADAM2 overexpressing tumors. Together, direct in vivo CRISPR/Cas9 screens can uncover genetic alterations that control responses to immunotherapies.

Lung cancer remains one of the leading causes of cancer-related mortality worldwide, with a 5-year survival rate of <20%. Non-small cell lung cancer (NSCLC) accounts for ~85% of lung cancer and is comprised of lung adenocarcinoma (LUAD), lung squamous cell carcinoma (LUSC) and large cell lung carcinoma (LCLC), among which LUAD is the most prevalent subtype. Exposure to tobacco use is the biggest risk factor and together with other environmental toxins is responsible for the high mutational burden observed in NSCLC. The most frequently mutated genes in LUAD are TP53[1] (44%), KRAS[2] (33%), KEAP1[3–5] (17%), STK11[6] (17%), EGFR[7,8] (14%), NF1[9] (11%), BRAF[10] (10%), PIK3CA[11] (7%), MET (7%)[12–14]. LUAD patients are usually treated with surgery, chemotherapy, radiation therapy, targeted therapy, or a combination of

[1]Centre for Molecular and Systems Biology, Lunenfeld-Tanenbaum Research Institute, Mount Sinai Hospital, Toronto, ON, Canada. [2]Department of Molecular Genetics, University of Toronto, Toronto, ON, Canada. [3]Computational Biology Program, Ontario Institute for Cancer Research, Toronto, ON, Canada. [4]Department of Pathology and Bacteriology, Klinik Floridsdorf, Vienna, Austria. [5]Institute for Research in Immunology and Cancer (IRIC), Université de Montréal, Montreal, QC, Canada. [6]Department of Biological Sciences, University of Calgary, Calgary, AB, Canada. [7]Department of Otolaryngology, Head and Neck Surgery, McGill University, Montreal, QC, Canada. [8]Toronto General Hospital Research Institute, University Health Network, Toronto, ON, Canada. [9]Departments of Laboratory Medicine and Pathobiology, Immunology, University of Toronto, Toronto, ON, Canada. [10]SickKids Research Institute, University Health Network, Toronto, ON, Canada. [11]Karl-Landsteiner-Institute for Lung Research and Pulmonary Oncology, Klinik Floridsdorf, Vienna, Austria. [12]Department of Pathology and Cell Biology, Faculty of Medicine, Université de Montréal, Montreal, QC, Canada. ✉e-mail: schramek@lunenfeld.ca

these treatments. Efforts to generate molecular targeted therapies have largely focused on frequently mutated genes and led to the development of tyrosine kinase inhibitors (e.g. EGFR, ALK, MET, NTRK) or allele-specific inhibitors (e.g. BRAF[V600E], KRAS[G12C])[15–23]. However, prolonged treatment with these targeted therapies often results in the development of acquired drug resistance that limits the duration of their clinical benefit. In addition, the use of these targeted therapies is restricted to a relatively small group of patients whose tumors carry the corresponding genetic alterations.

Immune-checkpoint inhibitors blocking the PD1/PDL1 axis or CTLA-4 are new therapeutic approaches for lung cancer patients[24,25]. While immunotherapy and induction of immune responses alone offer modest overall response rates, durable responses for some patients have been observed with combinatorial approaches. For instance, the PACIFIC study showed that adding the anti-PDL1 monoclonal antibody durvalumab to chemotherapy improved progression-free survival (PFS) and overall survival (OS) for patients with NSCLC[26,27]; the CheckMate 816 study confirmed a significant increase in pathologic complete response with the addition of the anti-PD1 monoclonal antibody nivolumab to neoadjuvant chemotherapy in resectable lung cancer[28,29]; significantly improved disease-free survival was reported in the IMpower010 study that utilized the anti-PDL1 monoclonal antibody atezolizumab and chemotherapy for patients with resected NSCLC[30]. Lastly, a chimeric antigen receptor (CAR) T-cell therapy, targeting several antigens (e.g. MSLN, MUC1, NY-ESO-1, GPC3, PSCA, EGFR, ROR1, HER2, PDL1) with limited expression in normal tissues but high and/or specific expression in tumor cells, is presently being tested against NSCLC[31]. The fact that immune-checkpoint blockade yields durable responses in 10–75% of patients across 17 different malignancies with the highest response rate in Hodgkin lymphoma, underscores the need for deepening our understanding of cancer immune biology[32]. Systematically cataloging immune-regulatory genes altered in cancer holds the promise of improving treatment decisions and stratifying patients into effective immunotherapies. In addition, elucidating molecular mechanisms that render tumors sensitive or resistant to immunotherapy might identify new targets to enhance existing immunotherapies and improve outcomes in lung cancer.

The advent of functional genetic CRISPR/Cas9 screens has provided a platform for unbiased and systematic identification of genes associated with cancer-intrinsic immune evasion, immunotherapy responses, and therapeutic resistance in tumor cells. While extremely valuable, the majority of CRISPR screens are conducted with T cells cocultured with cancer cells in vitro, which fail to fully recapitulate the complexity of the cellular heterogeneity within tissues[33–35]. CRISPR screens utilizing spheroids or organoids are the next level of in vitro systems that better address cellular interactions and mimic in vivo conditions. For example, genome-wide CRISPR screen in human NSCLC cell line-based 3D spheroid and xenograft tumors revealed that carboxypeptidase D (CPD) depletion prevents tumor growth in spheroids and in vivo, but not in 2D culture[36]. However, in vitro spheroids and organoids reiterate in vivo architecture only to a certain degree, suffer from low reproducibility, and still fail to fully recapitulate the complexity of a living organism. In contrast, in vivo screens performed by allografting of syngeneic mouse cancer cells or by using direct or indirect heterotopic or orthotopic xenografting of patient-derived cells (PDXs) resemble more closely a functional tumor microenvironment (TME). These approaches have led to the identification of genes implicated in key immune responses required for cancer cell immune escape and/or resistance to immune-checkpoint blockade (ICB). For instance, in vivo genome-wide or epigenetic screens identified genes involved in IFNγ signaling and antigen presentation (PTPN2, ADAR1, APLNR); suppression of tumor-intrinsic immunogenicity (SETDB1); immune-editing of TME (ASAF1, COP1); and in direct stress response-induced regulation of PDL1 expression (EIF5)[37–44]. Thus, CRISPR-based in vivo screens can be leveraged to

identify immune-regulatory genes that might serve as prognostic, diagnostic or potential drug targets in cancer. However, the wounding responses and distorted 3D architecture that inadvertently ensues upon grafting cancer cells, can confound the results of such screens.

In this work, we overcome the need for orthotopic transplantation and assess the function of genes within cells embedded in their native tissue architecture by developing an autochthonous CRISPR/Cas9 lung cancer screening methodology and combining it with adoptive transfer of cytotoxic T cells specific to a model tumor antigen. This direct autochthonous screen allows for simultaneous functional interrogation of hundreds putative cancer genes within an intact TME comprised of a functional immune system and endogenous signaling provided by the resident tissue surrounding the target cells. We screen 573 genes that are associated with altered immune activity in human tumors[45] and report the identification of several known genes such as *Serpinb9*, that plays a role in immune evasion in LUAD as well as other cancers[46,47]. We also unveil the role of a poorly characterized cancer-testis antigen, *Adam2*, in establishing a cold TME by blocking type I and II interferon and TNFα pathways. Interestingly, in the presence of activated exogenous tumor-specific CD8 T cells, *Adam2* expression results in increased tumor clearance by enhanced cytotoxicity of adoptively transferred CD8 T cells.

## Results
### Direct in vivo CRISPR immune screen in lung cancer

To functionally screen genes that determine the immune response to lung cancer, we established an in vivo CRISPR loss-of-function screening methodology (Fig. 1A). We generated a lentiviral construct that harbors a sgRNA for CRISPR/Cas9-mediated gene editing as well as a Cre-recombinase. For efficient antigen presentation and induction of immune responses by lung epithelial cells, the ovalbumin peptide SIINFEKL (OVA), an experimental tumor-associated antigen (TAA), was cloned into the lentiviral backbone (LV-sgRNA-Cre-OVA). To determine the viral concentration needed to transduce the lung epithelium at clonal density, we administer LV-sgRNA-Cre-OVA at postnatal day 2 (P2) to the lungs of Lox-Stop-Lox (LSL)-Kras[G12D] or LSL-Braf[V600E] mice crossed to multicolor LSL-Confetti mice[48] using intranasal instillation. Cre-mediated excision of the LSL-cassettes induced expression of Kras[G12D] or Braf[V600E] triggered expression of one of four fluorescent proteins encoded in the Confetti reporter cassette (Fig. 1B). Histological analysis revealed the induction of hundreds of independent clones and ultimately ~600 Kras[G12D]- or Braf[V600E]-driven tumors within a single lung (Supplementary Fig. 1a, b). For the CRISPR/Cas9 screen, we then generated LSL-Kras[G12D] or LSL-Braf[V600E]; LSL-Cas9-GFP; LSL-Luc mice, which concomitantly express a conditional Cas9-GFP transgene for CRISPR/Cas9-mediated gene editing as well as a luciferase transgene for noninvasive measurement of lung tumor volume (hereafter termed Kras[G12D];Cas9 and Braf[V600E];Cas9 mice) (Fig. 1A).

All LV-sgRNA-Cre-OVA transduced lung cells also express and present the OVA antigen in the context of major histocompatibility complex class I (MHC-I) H2K[b] molecules, which can be recognized by adoptively transferred cytotoxic OT-I CD8[+] T cells and can ultimately lead to OT-I-mediated lysis of tumor cells. To optimize different adoptive cell transfer (ACT) immune therapy paradigms, we used OT-I T cells isolated from OT-I; mT/mG mice that express a membrane-targeted red fluorescent Tomato protein. These cells were injected in Kras[G12D];Cas9 mice via the tail-vain four weeks after lung tumor induction by LV-Cre-OVA inhalation. These mice were then immunized with OVA-peptide emulsified in Complete Freund's adjuvant (CFA) on day 1 and primed with OVA/Incomplete Freund's adjuvant (IFA) emulsion on day 7 post-ACT to stimulate specific OT-I T-cell responses. We then compared the cytolytic activity of OT-I T cells isolated from the periphery (spleens and LNs) and lungs of Kras[G12D];Cas9 mice without or with the treatment of PDL1 or CTLA4 blocking antibodies by quantifying activation markers and Granzyme (GzmB)-expression

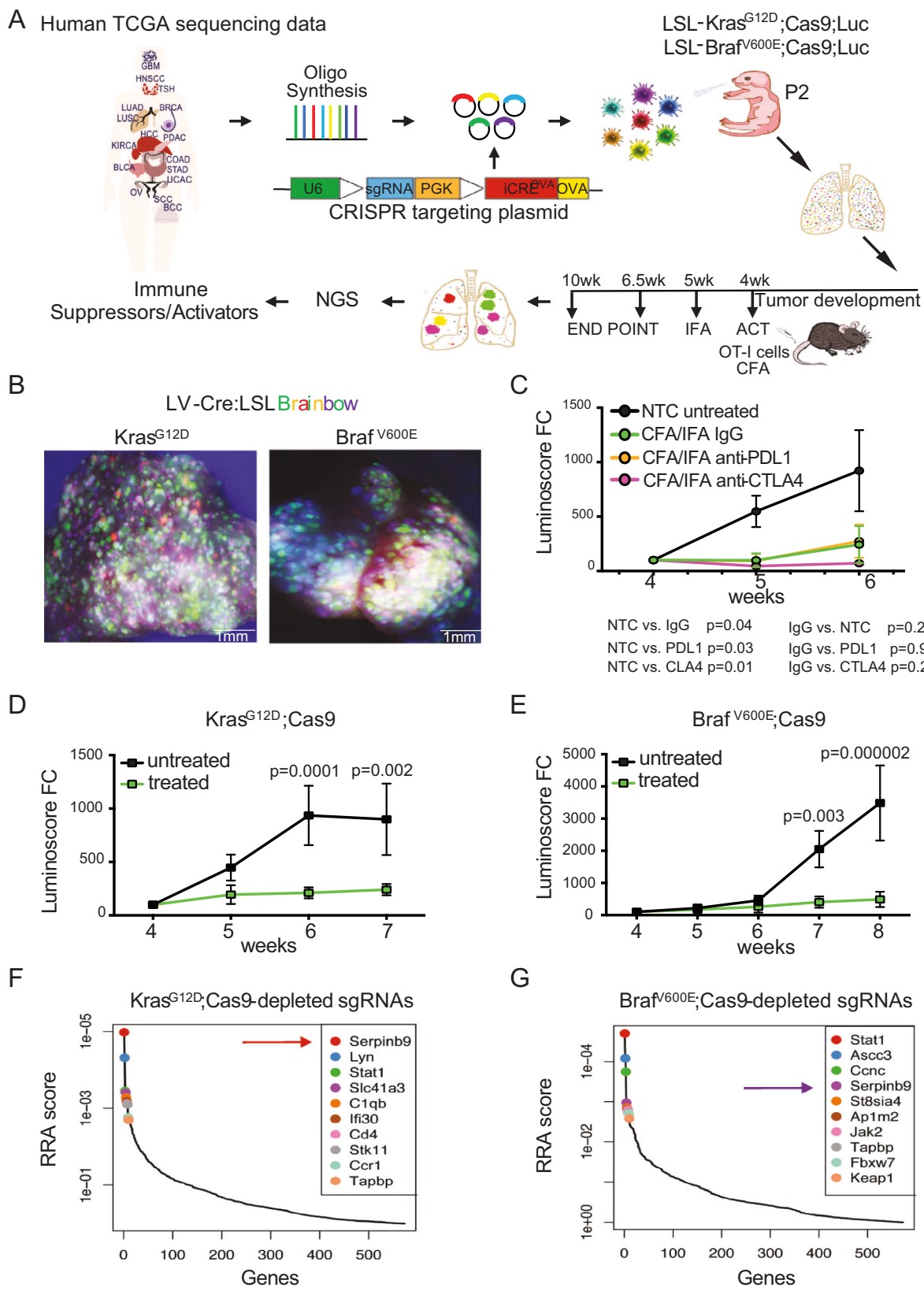

**A** Human TCGA sequencing data

**B** LV-Cre:LSL Brainbow

**C**

NTC vs. IgG    p=0.04        IgG vs. NTC    p=0.2
NTC vs. PDL1 p=0.03          IgG vs. PDL1  p=0.9
NTC vs. CLA4 p=0.01          IgG vs. CTLA4 p=0.2

**D** Kras^G12D;Cas9

**E** Braf^V600E;Cas9

**F** Kras^G12D;Cas9-depleted sgRNAs

**G** Braf^V600E;Cas9-depleted sgRNAs

(Supplementary Fig. 2a–c). Bioluminescent imaging (BLI) was used to quantify lung tumor burden over time (Fig. 1C). Compared to untreated mice, tumor burden in mice treated with OT-I T-cell were significantly smaller, which was further enhanced by CTLA4 but not PDL1 blocking antibodies. Of note, proliferation and activation of OT-I cells were confirmed in vitro and in vivo (Supplementary Fig. 3a–d). We

decided to use the ACT treatment of OVA-CFA/IFA-activated OT-I but without CTLA4 for the ACT treatment, as this treatment regimen would allow us to find genes that block but also enhance the therapeutic effect of ACT.

Next, we generated a pooled lentiviral CRISPR loss-of-function library targeting the mouse homologs of 573 human genes, whose

**Fig. 1 | In vivo CRISPR screen identifies regulators of immune response in lung cancer. A** Experimental workflow of the in vivo CRISPR screen to identify immune-modulatory cancer genes. A lentiviral sgRNA library targeting mouse homologs of 573 human genes, which are associated with altered cytotoxicity in cancer, is introduced into the lung of tumor-prone mice at P2. Mice are treated with OT-I T cells at week 4 and immunized with OVA emulsified in CFA/IFA. sgRNA representation in genomic tumor DNA are quantified by NGS. Human diagram is a modified version of 'Design by Freepik' (www.freepik.com) **B** Representative whole mount immunofluorescence images of the lung from LSL-Kras$^{G12D}$;LSL-Confetti and LSL-Braf$^{V600E}$;LSL-Confetti mice transduced with Cre lentivirus ($n = 6$ biologically independent samples for each group). **C** Growth curves of tumors in Kras$^{G12D}$;Cas9 mice treated with OT-I cells at week 4 in the presence or absence of PDL1 or CTLA4 blocking antibodies ($n = 5$ for each group). The $p$ values for tumor growth were determined by an unpaired two-sided $t$-test. The data are presented as the mean ± SEM. **D** Growth curves of tumors in Kras$^{G12D}$;Cas9 mice. (untreated $n = 7$; OT-I treated $n = 10$). The $p$ values for tumor growth were determined by multiple unpaired $t$-test. The data are presented as the mean ± SEM. **E** Growth curves of tumors in Braf$^{V600E}$;Cas9 mice (untreated $n = 6$; OT-I treated $n = 6$). The $p$ values for tumor growth were determined by multiple unpaired $t$-test. The data are presented as the mean ± SEM. Top 10 genes whose sgRNAs are depleted in lungs of ACT-treated Kras$^{G12D}$;Cas9 (**F**) and -Braf$^{V600E}$;Cas9 mice (**G**) (untreated, $n = 10$; treated, $n = 10$). RRA, Robust Rank Aggregation, which identifies statistically significant depleted genes across the two experimental conditions with the $p$ values gained from the negative binomial (NB) model used by MAGeCK RRA to rank the sgRNAs.

---

expression, mutation or copy number alteration (CNA) is associated with altered immune cytolytic activity across 8709 tumors from 18 different human cancer types in the TCGA dataset (4 sgRNAs/gene, Supplementary Data 1)[45]. We also generated a control library that included 418 non-targeting sgRNAs (termed NTC) as well as an sgRNA targeting the permissive TIGRE locus. These libraries were inhaled into Kras$^{G12D}$;Cas9 or Braf$^{V600E}$;Cas9 mice (Fig. 1A). Next generation sequencing (NGS) confirmed efficient lentiviral transduction of all sgRNAs in the viral library (Supplementary Figs. 4a–c, 5a–c and Supplementary Data 2).

To identify genes that confer resistance or sensitivity to immunotherapy, we quantified sgRNAs at 6.5 weeks after tumor induction, the time point when significant difference in tumor burden was observed between Kras$^{G12D}$- or Braf$^{V600E}$-driven lung tumors treated and untreated with ACT of activated cytotoxic OT-I T cells (Fig. 1D, E). To reveal suppressors of T-cell-mediated killing, we ranked sgRNAs that were depleted in treated versus untreated Kras$^{G12D}$;Cas9 or Braf$^{V600E}$;Cas9 mice (Fig. 1F, G, Supplementary Figs. 4d, e, and 5d, Supplementary Data 3). Kras$^{G12D}$-specific hits included *Serpinb9*, *Lyn*, *Stat1*, *Slc41a* and *C1qb* whereas Braf$^{V600E}$-specific hits included *Stat1*, *Ascc3*, *Ccnc*, *Serpinb9* and *St8sia4*. Depletion of *Stat1*, which has noticeable context-dependent immune-suppressive or -sensitizing role in solid cancers[38,44,49–52], was identified in both Kras$^{G12D}$ and Braf$^{V600E}$ backgrounds. We also observed depletion of sgRNAs targeting genes with known function in tumor immunity and/or resistance to immune-checkpoint therapy, such as *CD4*, C1qb in Kras$^{G12D}$ lungs[53–55]. Genes with yet unidentified immunological function in solid cancers included the Src family kinase *Lyn* and the magnesium transporter (*Slc41a3*) identified in the Kras$^{G12D}$ screen and transcription coactivator complex (*Ascc3*), haploinsufficient tumor suppressor cyclin C (*Ccnc*) and sialyltransferase (*St8sia4*)[56] identified in the Braf$^{V600E}$ screen (Fig. 1F, G). Together, our direct in vivo CRISPR/Cas9 lung cancer screen recapitulated the contribution of previously known immune-regulatory genes and identified potentially new tumor-intrinsic factors involved in sensitizing tumor cells to immune-mediated killing.

### Loss of Serpinb9 increases sensitivity of tumor cells to T-cell-mediated killing

Our top hit in the Kras$^{G12D}$ background and the fourth hit in the Braf$^{V600E}$ background was the serine protease inhibitor (*Serpinb9*). SERPINB9 is the only known intracellular inhibitor of the serine protease granzyme B (GZMB). GZMB is highly expressed by cytolytic CD8$^+$ T cells, natural killer (NK) and γδT cells to eliminate pathogenic and tumor cells and SERPINB9 is usually expressed in these cytolytic effector cells to protect them from apoptosis induced by their own GZMB[57–59]. In addition, SERPINB9 was recently described as an immunosuppressor and shown to be upregulated in several cancers[45–47]. For validation experiments, we therefore first focused on Serpinb9. To reveal the role of Serpinb9 in lung cancer in vivo, we genetically ablated Serpinb9 in the lung of Kras$^{G12D}$;Cas9 or Braf$^{V600E}$;Cas9 mice by inhaling mice with LV-CRE-sgSerpinb9-OVA at P2. Three to four weeks later, at the time when lung tumors could be detected by BLI, these mice and control littermates

transduced with LV-CRE-sgNTC-OVA were injected with OT-I cells followed by OVA-CFA/IFA activation. Efficient depletion of *Serpinb9* was observed in all tested lung tumors (Supplementary Fig. 6a, b).

Genetic ablation of *Serpinb9* significantly decreased lung tumor burden in untreated Kras$^{G12D}$;Cas9 and Braf$^{V600E}$;Cas9 mice (Fig. 2A, B and Supplementary Fig. 6c, d). This is in line with the recently identified immunosuppressive role of Serpinb9 in orthotopic mouse models[47]. While ACT treatment significantly slowed the growth of control (CTRL) Kras$^{G12D}$ and Braf$^{V600E}$ tumors, loss of Serpinb9 further enhanced the effect of ACT treatment and completely blocked tumor growth in both Kras$^{G12D}$;Cas9 and Braf$^{V600E}$;Cas9 mice (Fig. 2A, B). Interestingly, although genetic depletion of *Serpinb9* on its own resulted in a similar extension of survival as ACT treatment in control Kras$^{G12D}$ mice, it did not further extend the overall survival of ACT-treated Kras$^{G12D}$ mice (Fig. 2C). This is likely due to the single infusion of OT-I cells at the outset of tumor growth and further optimization would be needed to maximize ACT efficacy. However, in the Braf$^{V600E}$-lung cancer model, ACT treatment significantly increased the overall survival of mice transduced with sgSerpinb9 compared to control mice (Fig. 2D). Eventually, unedited *Serpinb9*-wildtype escapers and other immune-editing mechanisms lead to tumor growth and death. It is also important to mention that the survival analysis shown in Fig. 2C, D depict overall survival and some of the mice had to be sacrificed not because of lung tumor burden but due to other health complications. Together, our findings unveil that depletion of Serpinb9 enhances CD8 T-cell antitumor immunity in both Kras$^{G12D}$- and Braf$^{V600E}$-driven mouse models of lung adenocarcinoma.

To further validate the function of *Serpinb9*, we genetically ablated *Serpinb9* in the murine Lewis Lung Carcinoma cells (LLC1), which were established from a C57Bl/6 mouse lung tumor, harbor the activating Kras$^{G12D}$ mutation and form immunogenic tumors when transplanted into syngeneic mice[60]. Transduction of LLC1 cells with LV-CRE-sgSerpinb9-OVA resulted in almost complete depletion of *Serpinb9* (96%) (Supplementary Fig. 7a). To assess how Serpinb9 expression modulates cytotoxic T-cell killing, we cocultured *Serpinb9* knockout LLC1 cells with activated OT-I cells. As expected, *Serpinb9* KO cells were significantly more susceptible to OT-I T-cell-mediated killing than control cells (Fig. 2E and Supplementary Fig. 7b–d).

Since about 30% of human LUAD have increased *SERPINB9* copy number, which is associated with significantly reduced CD8 T-cell tumor infiltrates and patients' overall survival (Supplementary Fig. 8a, b), we further validated *SERPINB9* function in human LUAD cell lines[46,61]. We overexpressed human SERPINB9 in human lung cancer cell lines A549 and H125 (Supplementary Fig. 8c–e) and evaluated cytolytic activity of ex vivo activated and expanded γδT cells derived from healthy human donors (UPN119, UPN133), termed DNT cells[62]. As expected, A549 and H125 cell line overexpressing SERPINB9 were significantly more resistant to human DNT cell-mediated killing when compared to control cells (Fig. 2F, G and Supplementary Fig. 8f, g). Together, our in vitro and in vivo functional analyses corroborate the role of SERPINB9 as an important tumor-intrinsic mechanism in promoting resistance to immunotherapy.

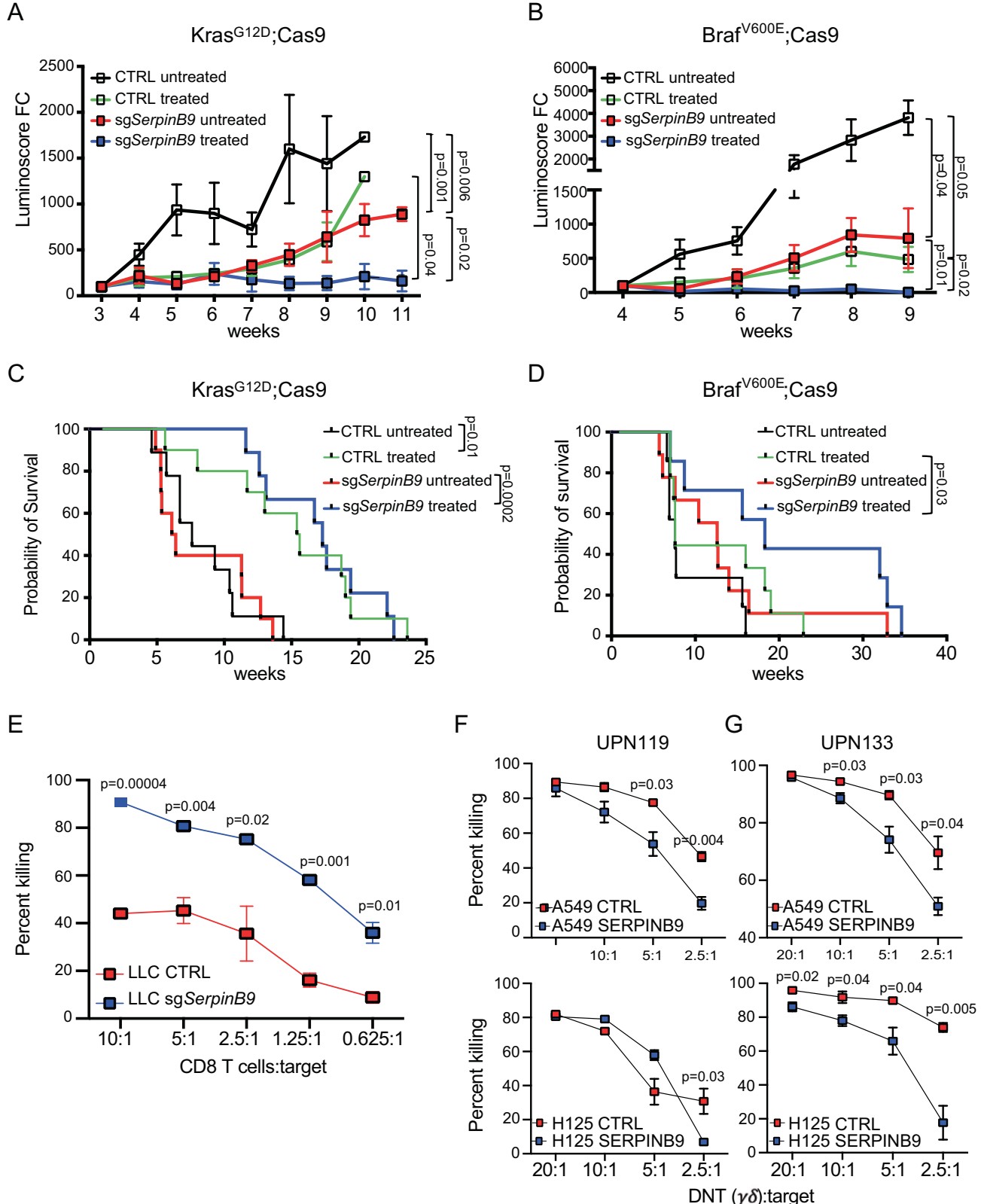

**Loss of *Adam2* impedes T-cell-mediated killing**

Next, we turned to enriched sgRNAs, which delineate genes that enhanced T-cell-mediated killing in our screens. Top hits from the *Kras*[G12D];*Cas9* screen included genes known to be required for immune responses, such as *Tapbl*, encoding a component of the antigen-processing and presentation machinery; and *Vhl* (von-Hippel-Lindau),

a tumor suppressor gene that enhances NK-cell activation in renal cell carcinoma (Fig. 3A)[63]. The top hit from the *Braf*[V600E];*Cas9* screen included *Trem2*, a known regulator of immune response with unknown function in tumor cells (Supplementary Fig. 9a)[55,64,65]. Interestingly, *Adam2* (Disintegrin and Metalloprotease Domain-Containing Protein2), scored as top gene in the *Kras*[G12D] but not in the *Braf*[V600E] screen

**Fig. 2 | SERPINB9 regulates T-cell-mediated killing and lung tumor growth.**
**A** Growth curves of tumors in Kras[G12D];Cas9 mice transduced with sgNTC or
sgSerpinb9 untreated or ACT treated. (CTRL: untreated $n = 10$; treated, $n = 10$ for
each group) The $p$ values for tumor growth were determined by an unpaired two-
sided $t$-test with Welch's correction. The data are presented as the mean ± SEM.
**B** Growth curves of tumors in Braf[V600E];Cas9 mice transduced with sgNTC or
sgSerpinb9 untreated or ACT treated. (CTRL: untreated $n = 7$ and treated $n = 9$;
sgSerpinb9: untreated $n = 6$ and treated $n = 6$). The $p$ values for tumor growth were
determined by an unpaired two-sided $t$-test with Welch's correction. The data are
presented as the mean ± SEM **C** Tumor-free survival of Kras[G12D];Cas9 mice from (A).
**D** Tumor-free survival of Braf [V600E];Cas9 mice from (**B**). Comparison of survival
curves was performed by Log-rank (Mantel–Cox test). **E** The effect of Serpinb9
knockout on T-cell-mediated tumor killing. LLC cells transduced with sgNTC or

sgSerpinb9 and labeled with CFSE were used as targets (*T*) and cocultured with
activated OT-I effector cells (*E*) at different *E:T* ratios for 4 h. Flow cytometry ana-
lysis was used to quantify the percentage of target cells killing at different *E:T* ratios.
Data presents mean ± s.e.m. of three technical replicates analyzed by multiple
unpaired $t$-tests. One representative experiment out of three independent experi-
ments is shown. **F**, **G** The effect of SERPINB9 overexpression on DNT cell-mediated
killing. Human LUAD cell lines, A549 and H125 labeled with CFSE were used as
targets (*T*) and cocultured with activated and expanded DNT effector cells (*E*) from
two different donors (UPN119, UPN133) at different *E:T* ratios for 18 h. Data presents
mean ± s.e.m. of three technical replicates analyzed by multiple unpaired $t$-tests.
One representative experiment out of two independent experiments is shown for
each donor.

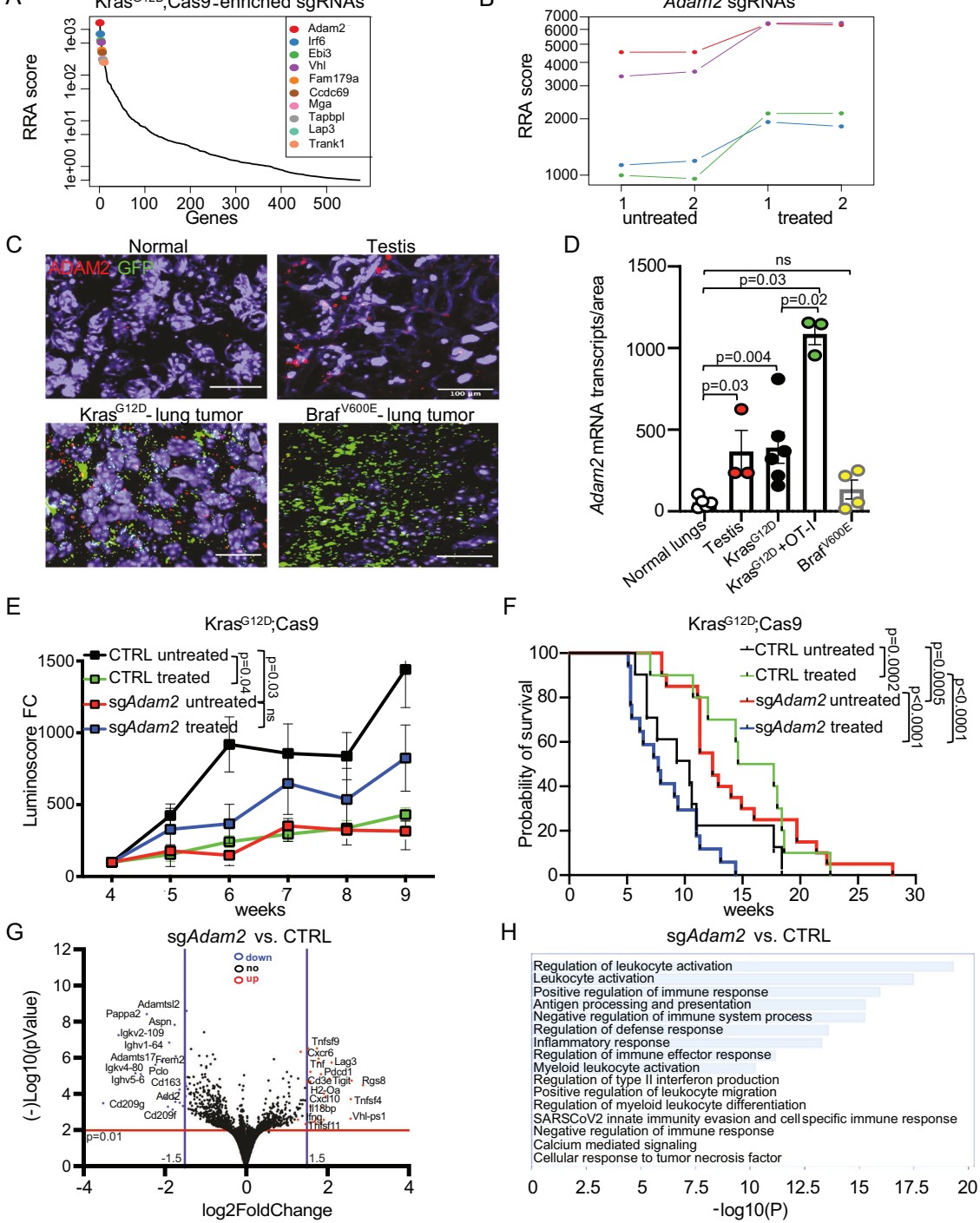

**Fig. 3 ADAM2 regulates leukocyte activation and cytokines in Kras-driven lung cancer. A** Top 10 genes whose sgRNAs are enriched in lungs of ACT-treated *Kras*[G12D];Cas9 mice. (untreated $n = 10$; treated $n = 10$; RRA, Robust Rank Aggregation, which identifies statistically significant enriched genes across the two experimental conditions with the *p* values gained from the negative binomial (NB) model used by MAGeCK RRA to rank the sgRNAs). **B** Relative abundance of 4 different sgRNAs targeting *Adam2* depicted by different colors in 2 replicates of untreated and 2 replicates of ACT-treated *Kras*[G12D] lungs ($n = 5$ lungs for each replicate). **C** RNAscope analysis of *Adam*2 (red) and GFP (green; tumor cells) expression in indicated tissues ($n = 5$ normal lungs; $n = 3$ testis; $n = 6$ *Kras*[G12D]Cas9; $n = 3$ *Kras*[G12D]Cas9 + OT-I; $n = 4$ *Braf*[V600E]Cas9 biologically independent samples). The scale bar represents 100 µm. **D** Quantification of *Adam*2 transcripts from (**C**) with *p* values calculated by unpaired two-sided student's *t*-test; ns non-significant; data are mean ± s.e.m. **E** Growth curves of tumors in *Kras*[G12D];Cas9 mice transduced with sgNTC (CTRL untreated $n = 8$; CTRL treated $n = 12$) or sg*Adam2* (untreated $n = 14$; treated $n = 17$; sg1 and sg2 are shown combined here and are shown

separately in Supplementary Fig. 11d, e). The *p* values for tumor growth were determined by an unpaired two-sided *t*-test with Welch's correction. The data are presented as the mean ± SEM. **F** Tumor-free survival of *Kras*[G12D];Cas9 mice transduced with sgNTC (untreated, $n = 10$; treated, $n = 10$) vs. sgRNA sg*Adam2* (untreated, $n = 20$; treated, $n = 18$; sg1 and sg2 are shown combined here and are shown separately in Supplementary Fig. 11f, g); Comparison of survival curves was performed by Log-rank (Mantel−Cox test). Data are mean ± s.e.m. **G** Volcano plot showing differentially expressed genes (DEGs) in sg*Adam2* KO ($n = 4$) compared to CTRL ($n = 4$) lung tumors isolated from C57BL/6 mice, where log2 FC indicates the mean expression and (−)log10 adjusted *p* value level for each gene. The blue dots denote downregulated gene expression, the red dots denote upregulated expression, and black dots denote the gene expression without marked difference. Data represents three independent biological samples for each group. **H** Bar graph showing Gene Ontology of the DEGs (FC > 2, *p* < 0.05) downregulated in sg*Adam2* KO compared to CTRL lung tumors assigned to Biological Process.

(Fig. 3A, B and Supplementary Fig. 5d). *ADAM2* encodes a non-catalytic metalloprotease-like protein, whose expression is usually restricted to the testis and sperm, where it is essential for the sperm-egg fusion[66]. However, aberrant expression of *ADAM2* was also reported in some malignancies (Supplementary Fig. 9b)[67], indicating that ADAM2 might constitute a poorly characterized cancer-testis antigen.

We first evaluated expression of *Adam2* in our mouse lung cancer models. We examined untreated or ACT-treated lung tumors from *Kras*[G12D];Cas9 and *Braf*[V600E];Cas9 mice using probes against *Adam2* as well as against eGFP to detect tumor cells. We used testis as a positive control and normal lungs as a negative control. As expected, we observed high expression of *Adam2* in testis, but not in healthy lungs. Importantly, *Adam2* was expressed in *Kras*[G12D] lung tumors to a level similar to that seen in testis and its expression was even further increased upon treatment with cytotoxic antigen-specific T cells. In contrast, *Adam2* mRNA levels in *Braf*[V600E] lung tumors were negligible (Fig. 3C, D and Supplementary Fig. 10). This data corroborates that *Adam*2 is a cancer-testis antigen, whose expression is induced in an oncogene-specific manner that can be further increased under selective immune pressure. In addition, the absence of *Adam2* expression in *Braf*[V600E] tumors likely explains why *Adam2* did not surface as a hit in the *Braf* screen.

To functionally investigate whether ADAM2 regulates antitumor immunity, we genetically depleted *Adam2* in lungs of *Kras*[G12D];Cas9 mice using LV-CRE-sg*Adam2*-OVA. Efficient mutagenesis of *Adam2* was confirmed in all tested lung tumors (Supplementary Fig. 11a, b). Next, we evaluated tumor development in *Kras*[G12D];Cas9 mice transduced with sgNTC or 2 independent sgRNAs targeting *Adam2*. We observed that loss of *Adam2* significantly reduced tumor growth and significantly extended the survival of *Kras*[G12D] mice (Fig. 3E, F and Supplementary Fig. 11c−g). In addition, although ACT treatment significantly reduced the tumor burden in control lungs, it did not have a significant therapeutic effect in sg*Adam2*-deficient lungs−in fact there was a trend towards increased growth of treated *Adam2*-deficient tumors (Fig. 3E and Supplementary Fig. 11c−g). Similarly, the overall survival of ACT-treated compared to untreated mice with *Adam2*-knockout tumors was reduced, indicating an adverse effect of the ACT treatment in the background of *Adam2* loss. The survival analysis shown in Fig. 3F depicts overall survival and some of the mice had to be sacrificed due to health complications other than lung tumor burden. Irrespective of genetic background and treatment arm, all tumors exhibited the same histology of invasive nonmucinous adenocarcinoma with a mixture of low-grade (lepidic) and high-grade (solid, micropapillary) patterns (Supplementary Fig. 12). However, the *Adam2*-knockout lung tumors showed more lymphocytic infiltrates especially in the perivascular and peribronchiolar space (Supplementary Fig. 12). Indeed, immunohistochemistry identified significantly increased infiltration of CD8 T cells in treated *Adam2* knockout tumors

but with a concomitant drastic increase expression of the PDL1 exhaustion marker, indicating an immune-suppressive tumor environment (Supplementary Fig. 13). In addition, untreated *Adam2* knockout tumors exhibited significantly decreased CD68[+] and CD206[+] cells compared to untreated control tumors, indicating a decreased infiltration with M2-polarized tumor-promoting and immune-suppressive macrophages, which is in line with the decreased tumor growth of *Adam2* knockout tumors. However, treated compared to untreated *Adam2* knockout tumors exhibited increased CD68[+] and CD206[+] cells, further indicating increased immune-suppressive environment and together with the increased PDL1 expression potentially explaining the reduced efficacy of adoptively transferred cytotoxic OT1 cells (Supplementary Fig. 13).

To probe the mechanisms by which Adam2 modulates ACT responses, we performed whole transcriptome analysis (RNA-seq) on ACT-treated *Adam2*-knockout and control tumors. Gene set enrichment analysis (GSEA) revealed significant upregulation of genes associated with 'Antigen Processing and Presentation of endogenous antigen' and 'Immune System Processing' (Fig. 3G and Supplementary Fig. 14a). MetaScape and gProfiler analysis of the top 200 significantly upregulated genes revealed enrichment of genes associated with 'regulation of leukocyte activation', 'positive regulation of immune response', antigen processing and presentation, 'regulation of type II interferon production', 'T-cell activation', 'cytokine activity' and 'TNF receptor binding' (Fig. 3H, Supplementary Fig. 14a−c and Supplementary Data 4). Interestingly, several immune-checkpoint receptors such as *Pd-1* (*Pdcd1*), *Lag3* and *Tigit* together with several immune-modulatory cytokines such as *Ifnγ*, several *Tnf* ligands (*Tnfa*, *Tnfsf4*, *Tnfsf9*, *Tnfsf11*, *Tnfaip2*), and interferon-stimulated chemokines such as *Cxcl10* and *Ccl5* were amongst the top-upregulated genes, which was confirmed by quantitative RT-PCR (Supplementary Fig. 15a). Importantly, we confirmed upregulated expression of *Pd*-1, *Lag*3, *Tigit* and *Ifnγ* in CD8 T cells isolated from untreated as well as ACT-treated *Adam2*-knockout tumors compared to control tumors by quantitative RT-PCR and upregulated Pd-1 and Lag3 expression by FACS analysis (Supplementary Fig. 15b, c). Collectively, these data show that *Adam2* functions as a tumor promoter in *Kras*[G12D] tumors but is required for cytotoxicity of adoptively transferred T cells. Mechanistically, loss of Adam2 is associated with an elevated cytokine milieu (IFNγ, TNFs, etc.) and elevated expression of several immune-checkpoint receptors.

### ADAM2 suppresses endogenous IFNα/β, IFNγ, and TNFα responses in LUAD in vitro and in vivo

To further study the function of Adam2 during tumor development and immune regulation, we overexpressed Adam2 together with the OVA-peptide SIINFEKL in LLC1 cells (=Adam2 O/E cells). Adam2

overexpression was confirmed by western blotting and IFNγ-induced presentation of OVA bound to MHC-I H2K$^b$ was confirmed by flow cytometry (Supplementary Fig. 16a, b).

We then transplanted Adam2 O/E cells or vector only control cells subcutaneously into immunocompetent C57BL/6 mice. Overexpression of Adam2 resulted in a dramatically faster tumor growth and significantly reduced survival compared to control cells (Fig. 4A and Supplementary Fig. 17a, b). Compared to immunocompetent mice, both Adam2 O/E and control tumors grew faster in immunodeficient Nod-Scid-Gamma (NSG) mice, and, importantly, at similar rates (Fig. 4B and Supplementary Fig. 17c, e). Similarly, Adam2 O/E and control tumors did not show significant difference in tumor growth or overall survival in T-cell-deficient nude mice (Fig. 4C and Supplementary Fig. 17d, e). Since forced expression of Adam2 had no impact on tumor growth or survival in immunodeficient hosts, these data indicate that Adam2 suppresses endogenous immune responses that restrain the growth of LLC cells.

To further elucidate the mechanism through which Adam2 suppresses antitumor immunity, we profiled the transcriptome of control and Adam2 O/E tumors isolated from C57BL/6 mice. GSEA revealed significant reduction of IFNα/β responses (*Cxcl10, Cxcl11, Isg15, Usp18*), IFNγ responses (*Stat1, Stat2, Tap1, Tapbp, HLA-A, Apol6, Ddx60, Ddhx58*), TNFα signaling via NFκβ (*Ccl5, Ifih1, Map3k8, Ifit2, Tap1, Il6, Il1b, Irf1*), IL-2-STAT5 (*Icos, Fgl2, Gbp4, Tnfsf10*), IL6-JAK-STAT3 (*Il18r1, Il2ra, Il1b, Fas*) and complement (*Ltf, C2, C3, Psmb9, Ccl5, Casp1*) pathways (Fig. 4D–F and Supplementary Figs. 18–22). In addition, we found downregulation of cluster of differentiation 3 (*Cd3*), inducible T-cell co-stimulator (*Icos*), several MHC-I molecules, the MHC-I invariant chain *B2m*, the antigen-processing molecule tapasin (*Tap1*) as well as downregulation of the checkpoint molecules PD1-PDL1 (*Pcdc1-Cd274*), *Lag3, Tigit* and *Tim3* (*Havcr2*) (Fig. 4G and Supplementary Fig. 22b). These data indicate that ADAM2 expression suppresses cellular responses to multiple cytokine such as IFNα/β/γ and TNFα and confirms that ADAM2 functions as an oncogene by inhibiting overall immune responses.

The prominent role of IFNα/β, IFNγ and TNFα pathways in the induction of antigen processing and MHC-I-mediated antigen presentation on tumor cells prompted us to test H2K$^b$-OVA expression in CTRL and Adam2 O/E cells in response to IFNβ, IFNγ and TNFα treatment[33,34,37,40,68–77]. Flow cytometry revealed markedly delayed and significantly reduced levels of MHC-I-OVA surface expression in Adam2 O/E cells in comparison to CTRL cells in response to IFNβ, IFNγ or TNFα stimulation (Fig. 4H and Supplementary Fig. 23a–g). Corroborating these findings, we found that IFNγ treatment of Adam2 O/E LLC cells resulted in lower activation/phosphorylation of *Stat1*, which is the major downstream signal mediator of IFNγ, compared to CTRL cells (Supplementary Fig. 23h). Thus, Adam2 overexpression compromises IFNγ, IFNβ and TNFα signaling and MHC-I-mediated antigen presentation, which in turn affect functional innate and adaptive immune responses.

We next tested the expression of other interferon-regulated proteins such as PDL1, CD74 and Tigit. Although IFNγ, IFNα and TNFα-induced surface expression of PDL1 was not affected by Adam2, there was a significant increase of CD74 and Tigit expression in Adam2 O/E cells compared to CTRL cells (Fig. 4I and Supplementary Fig. 24a–f). The pro-inflammatory MIF cytokine receptor CD74 is required for antigen presentation by antigen presenting cells (APCs) in the context of MHC class II. However, expression of CD74 is also associated with epithelial cancer cells, where CD74 blocks the MHC-I peptide binding cleft and inhibits TAA presentation to T cells thus rendering tumors less immunogenic[78,79]. TIGIT (T-cell immunoreceptor with IgG and ITIM domains), whose expression is inhibited by IFNs[80], was recently identified as a cancer stem cell marker in LUAD[81] and TIGIT expression on tumor cells was shown to suppress CD8 T cells and NK cells[82]. These results imply that Adam2

downregulates type I/II IFN and TNFα responses as well as MHC-I expression, while upregulating other key immune-modulatory receptors on tumor cells resulting in reduced cross-presentation of TAA to antigen-specific CD8 T cells.

## ADAM2 dictates responses to ACT and ICB in LUAD in vitro and in vivo

Next, we evaluated how Adam2 modulates cytotoxic T-cell killing. Ex vivo activated and expanded OT-I CD8$^+$-T cells acquired a central memory phenotype, marked by upregulation of activation markers CD25, CD28, CD44, CD62L and GzmB; as well as expression of exhaustion markers CD223 (Lag3) and PD1 (Supplementary Fig. 25a). As expected, these activated OT-I cells efficiently killed OVA-peptide expressing CTRL cells. Interestingly, overexpression of Adam2 resulted in a significantly enhanced T-cell-mediated killing at 4 h ($p = 0.01$) and 8 h ($p = 0.0065$) (Fig. 5A and Supplementary Fig. 25b).

To further understand immunogenic properties of Adam2, CTRL or Adam2 O/E cells were transplanted in immunocompetent C57BL/6 mice followed by adoptive transfer of OT-I cells at day 14 (i.e., 2 days after tumor onset). As previously observed, untreated Adam2 O/E tumors grew significantly faster compared to CTRL tumors. Interestingly, ACT of activated OT-I cells had no effect on growth of CTRL tumors, but drastically restrained the growth and extended survival of Adam2 O/E tumors (Fig. 5B and Supplementary Fig. 26a, b). These results indicated that the reduced IFNα/β/γ responses and the less exhausted tumor microenvironment observed upon overexpression of Adam2 allows for improved ACT efficacy. In support of these findings, blocking IFN signaling by genetic ablation of *Stat1* in the *Kras*$^{G12D}$-driven lung cancer mouse model resulted in a significantly increased overall survival in comparison to control *Kras*$^{G12D}$ mice (Supplementary Fig. 26c). To test how Adam2 impacts not only efficacy of ACT but potentially also immune-checkpoint blockade (ICB), C57BL/6 mice bearing CTRL or Adam2 O/E LLC tumors were treated with PDL1 or CTLA4 blocking antibodies (Ab) once the tumor reached 100mm$^3$. PDL1 as well as CTLA4 blocking Ab significant slowed tumor growth of Adam2 O/E tumors, while control tumors showed little or no reduction in growth. In addition, combining PDL1 or CTLA4 inhibition with ACT led to almost complete tumor stasis of Adam2 overexpressing and control tumors, indicating strong cooperative effects (Supplementary Fig. 26d, e). These results suggest that Adam2 enhances the cytotoxic potential of endogenous or adoptively transferred antigen-specific CD8 T cells.

Notably, we detected increased infiltration of GzmB$^+$ OT1 T cells in ACT-treated Adam2 O/E tumors (Supplementary Fig. 27a, b). In line with these data, we found increased transcript levels of several serine proteases (*GzmB, GzmC, GzmD, GzmE, GzmF, GzmG*) in treated Adam2 O/E tumors compared to treated CTRL tumors isolated from C57BL/6 mice (Fig. 5C, D). In addition, we found upregulation of thrombospondin type1 domain containing 4 (*Thsd4, Adamtsl6*) in ACT-treated Adam2 O/E tumors. The expression of *THSD4* is associated with ICB sensitivity in a TCGA pan-cancer analysis[83,84]. Significantly downregulated genes in treated Adam2 O/E tumors were associated with adipogenesis and metabolism with gene sets associated with 'glucose homeostasis', 'cellular responses to fatty acid', 'metabolic and amino acid biosynthetic processes', 'angiotensin maturation'; and molecular functions associated with 'peptide activity', 'neutrothropin binding' and 'carboxypeptidase and serine type peptidase activity' (Fig. 5D, E and Supplementary Fig. 28a, b). While some of downregulated genes have been previously described as targets for cancer therapy such as asparaginase (*Aspg*)[85], semaphorins (*Sema3c*)[86], Adenosin A1 receptor (*Adora*1)[87], phosphoenolpyruvate (*Pck1*)[88], their association with immune responses are still largely unexplored. Taken together, these data reveal functions of Adam2 in the reprogramming of the tumor cells and TME to augment the cytotoxicity of antigen-specific T cells.

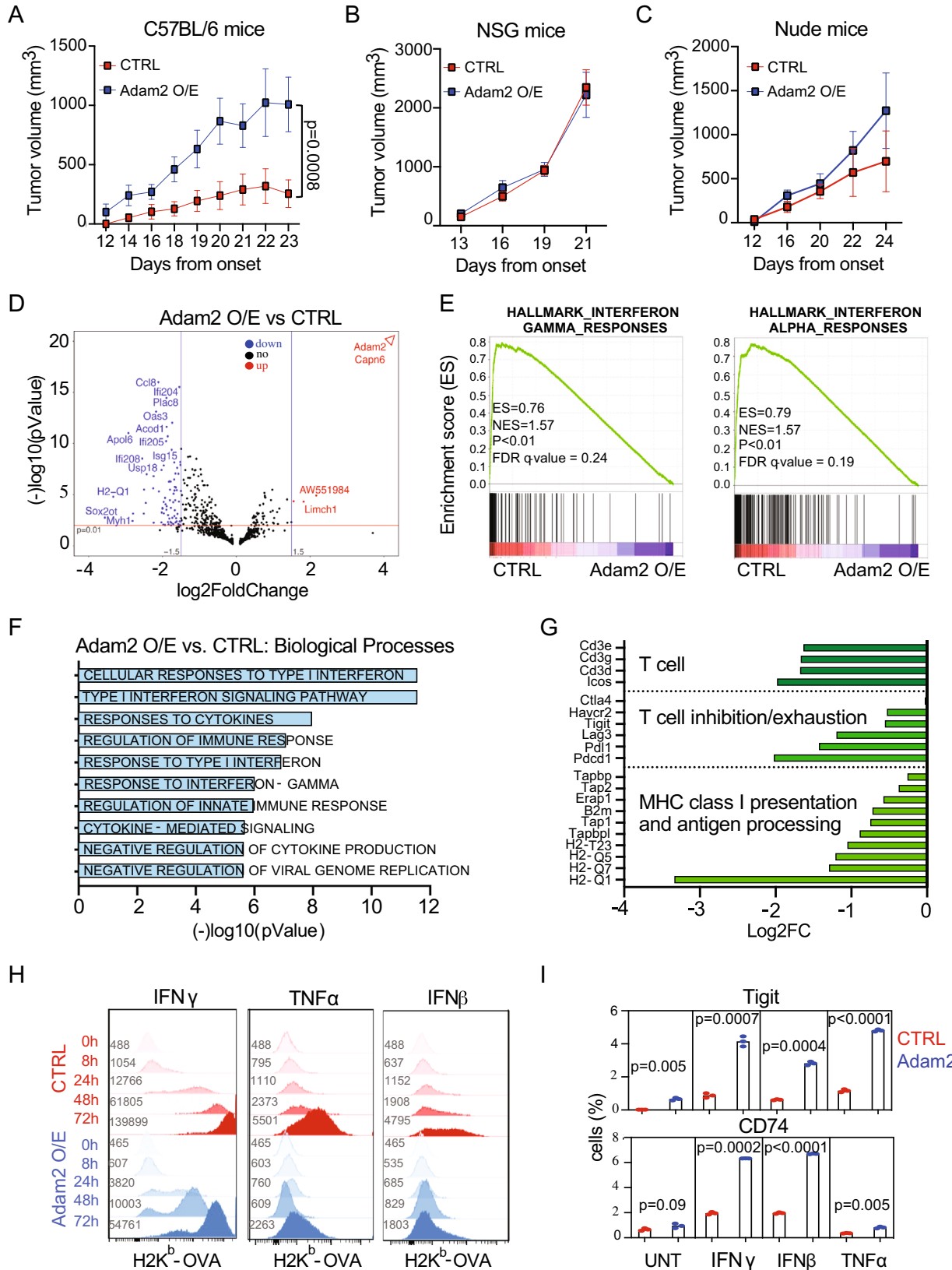

To further understand the impact of Adam2 on reprogramming the TME, cytometry by time of flight (CyTOF) analysis revealed higher abundance of CD11c⁺MHCII⁺CD64⁺Ly6G⁻ cells in Adam2 O/E tumors (Fig. 5F). These cells exhibit a macrophage phenotype as well as functional DC features and are implicated in higher degree of TAA cross-presentation to and priming of effector T cells[89]. This finding was further supported by detection of lower levels of PD1 expression on Ag-specific CD8⁺ T cells, highlighting the importance of Adam2 in decreasing intertumoral exhaustion of antigen-specific T cells (Fig. 5G). Together, Adam2 drives maturation and function of CD11c⁺MHCII⁺CD64⁺Ly6G⁻ macrophages and through regulation of INF-I/INF-II and TNFα

**Fig. 4 | ADAM2 promotes tumorigenesis by suppressing IFN and TNF immune responses. A–C** Tumor growth of CTRL and *Adam*2 O/E LLC cells allografted subcutaneously into C57BL/6 mice (**A**, *n* = 4 CTRL, *n* = 5 *Adam*2), NSG mice (**B**, *n* = 8) and Nude mice (**C**, *n* = 5) examined over three independent experiments. The *p* values for tumor growth were determined by an unpaired two-sided *t*-test with Welch's correction. The data are presented as the mean ± SEM. **D** Volcano plot showing differentially expressed genes (DEGs) in Adam2 O/E compared to CTRL tumors isolated from C57BL/6 mice, where log2 FC indicates the mean expression and (−)log10 adjusted *p*-value level for each gene. The blue dots denote down-regulated gene expression, the red dots denote upregulated expression, and black dots denote the gene expression without marked difference. Data represents three independent biological samples for each group. **E** GSEA plots showing Hallmarks of IFNγ and IFNα pathways between Adam2 O/E compared to CTRL tumors isolated

from C57BL/6 mice (A) with FDR < 25% and *p* < 1%. **F** Bar graph showing Gene Ontology of the DEGs (FC > 2, *p* < 0.05) downregulated in Adam2 O/E compared to CTRL tumors assigned to Biological Process isolated from C57BL/6 mice (**A**). **G** Bar graph depicting selected downregulated genes in Adam2 O/E compared to CTRL tumors categorized by immune function isolated from C57BL/6 mice (**A**). **H** Cell surface expression of MHC-I H2K$^b$-OVA on CTRL and Adam2 O/E LLC cells after treatment with 100 ng/ml of IFNγ, IFNβ or TNFα over the indicated time course. Numbers denote mean fluorescence intensity (MFI). **I** Percentage of CTRL and Adam2 O/E LLC cells expressing CD74 and Tigit after IFNγ, IFNβ or TNFα treatment (100 ng/ml each, 24 h). Data presents mean ± s.e.m. of three technical replicates analyzed by two-sided student's *t*-test examined. One representative experiment out of three independent experiments is shown.

responses contribute in activation/priming of Ag-specific T-cell responses within the TME.

### ADAM2-expressing human LUAD exhibit reduced IFNα, IFNγ and TNFα responses

To extend our findings from mouse to human cancer, we performed pan-cancer analysis for *ADAM2* from The Cancer Genome Atlas (TCGA). Pan-cancer analysis for *ADAM2* showed a high frequency of expression in breast (9.5%), lung (9.6%), bladder (16.7%), prostate (72.4%) and renal (74.7%) cancers (Fig. 6A and Supplementary Data 5). In lung cancer, 71/510 (13.9%) of LUAD and 26/498 (5.2%) of LUSC tumors showed *ADAM2* expression and ~50% of these cases showed *ADAM2* amplifications, indicating that *ADAM2* amplification correlates with ADAM2 expression (Supplementary Fig. 29a and Supplementary Data 5). In addition, 5% of all lung tumors harbored *ADAM2* missense mutations and 34% possessed gains or amplifications of the genomic region encompassing the *ADAM2* (Supplementary Fig. 29b–d). Inter-estingly, *ADAM2* expression was observed commonly in KRAS-mutant LUAD, while only one BRAF-mutant LUAD exhibited *ADAM2* expression (Supplementary Fig. 29e and Supplementary Data 6), reminiscent of our findings in the Kras and Braf-mutant mouse models.

To substantiate the TCGA data, we performed RNAscope ana-lysis on 96 human LUAD samples. While normal human lungs did not show expression of *ADAM2*, 33.9% of human LUAD tumor samples showed *ADAM2* expression akin to human testis (Fig. 6B, C). Fur-thermore, the pathway enrichment analysis revealed significant downregulation of genes associated with inflammatory responses, IFNα and IFNγ responses, TNFα signaling via NFκB, IL6/JAK/STAT3 signaling and complement pathways in *ADAM2*-expressing human LUAD (Fig. 6D, Supplementary Figs. 30–37 and Supplementary Data 7). Interestingly, amplification of the chromosomal region encompassing *ADAM2* has also been associated with significantly decreased signature of cytotoxic CD8 T-cell[45]. These transcriptional changes are virtually identical to the changes observed in our mouse lung cancer LLC model with constitutive overexpression of Adam2 (Fig. 4F), further supporting our findings. In addition, *ADAM2*-expressing human LUAD also exhibited increased expression of E2F, G2M and MYC, targets known to accelerate lung adenocarcinoma progression (Fig. 6D).

Together, these data show that *ADAM2* is a cancer-testis antigen expressed in a wide variety of tumor tissues, including a high pro-portion of human lung cancers, and profoundly affects the tumor immune microenvironment with implications for immune therapy.

### Discussion

Among the most promising recent advances in oncology is the development of cancer immunotherapies, which largely focus on reactivating endogenous antitumor immune responses. Immune-checkpoint inhibitors are designed to break the immune tolerance imposed by the tumor, particularly against cytotoxic T-cells[90–94] and have shown remarkable efficacy against cancers with high mutational

burden[95,96]. Similarly, adoptive cell transfer (ACT) of T cells isolated from patients, genetically modified and activated in vitro followed by infusion back into the same patient has proven as a powerful ther-apeutic strategy[97]. While very effective, immunotherapies result "only" in a ~50% response rate and mere 20% of patients experience a durable survival benefit, indicating the existence of resistance and other immune escape mechanisms. This raises several questions: What makes certain tumors resistant to immunotherapy and what can we do about it? Are there other immunotherapy targets that we can exploit? Can we combine treatments to enhance the effect of immu-notherapies? Are there genetic features that render tumors a priori less or more likely to respond and if so, could we use this information to stratify patients into optimal treatment arms? Given the cost of immunotherapies and the burden on health systems, identifying patients who are likely to respond to immunotherapy versus those who will benefit more from other treatments is paramount. From a patient's perspective, a predictive evaluation of treatment success would be extremely valuable as patients who would unlikely respond to immunotherapy can be evaluated for receiving alternative thera-pies. As such, cataloging the genetic changes that determine the sen-sitivity to immunotherapy would greatly aid clinical decision-making and impact patients' health.

The explosion of sequencing capabilities is expected to change clinical practices by personalizing treatments based on the genetic make-up of a given tumor and will certainly also help refine immuno-oncology. However, one key bottleneck on the path towards 'Precision ImmunoOncology' is our fragmented understanding of the functional consequence of most genetic alterations: which mutations are mere bystanders, which are real cancer drivers, and which can be used to determine therapy responses? Determining the effects, a given genetic alteration might have on the response to immunotherapy is exceed-ingly hard and requires a model system that not only ensures a native microenvironment with an intact immune system, but also where tumors can be genetically manipulated and functionally interrogated, ideally in a multiplexed and high-throughput manner. Previous studies have relied mainly on co-culture systems[35,40] or transplantation models[37].

To assess how the genomic landscape shapes antitumor immu-nity, we focused on somatic gene alterations that correlate with immune cytolytic activity in the TCGA cohort of 8709 tumors, as spearheaded by Rooney et al.[45]. Immune cytolytic activity is based on transcript levels of granzyme A and perforin, two key cytolytic effec-tors, that are dramatically upregulated upon NK and CD8$^+$ T-cell acti-vation and during productive clinical responses to immunotherapies[45]. As expected, this approach identified *B2M* (beta-2-microglobulin), *HLA-A*, *-B* and *-C* and *CASP8* (Caspase 8) as significantly mutated genes and *PDL-1/2*, *IDO1/2* as significantly overexpressed genes, highlighting not only loss of antigen presentation and blockade of extrinsic apop-tosis as key strategies to overcome cytolytic activity but also validating the approach. In addition to these known immune regulators, Rooney et al. identified an additional ~600 significant gene alterations

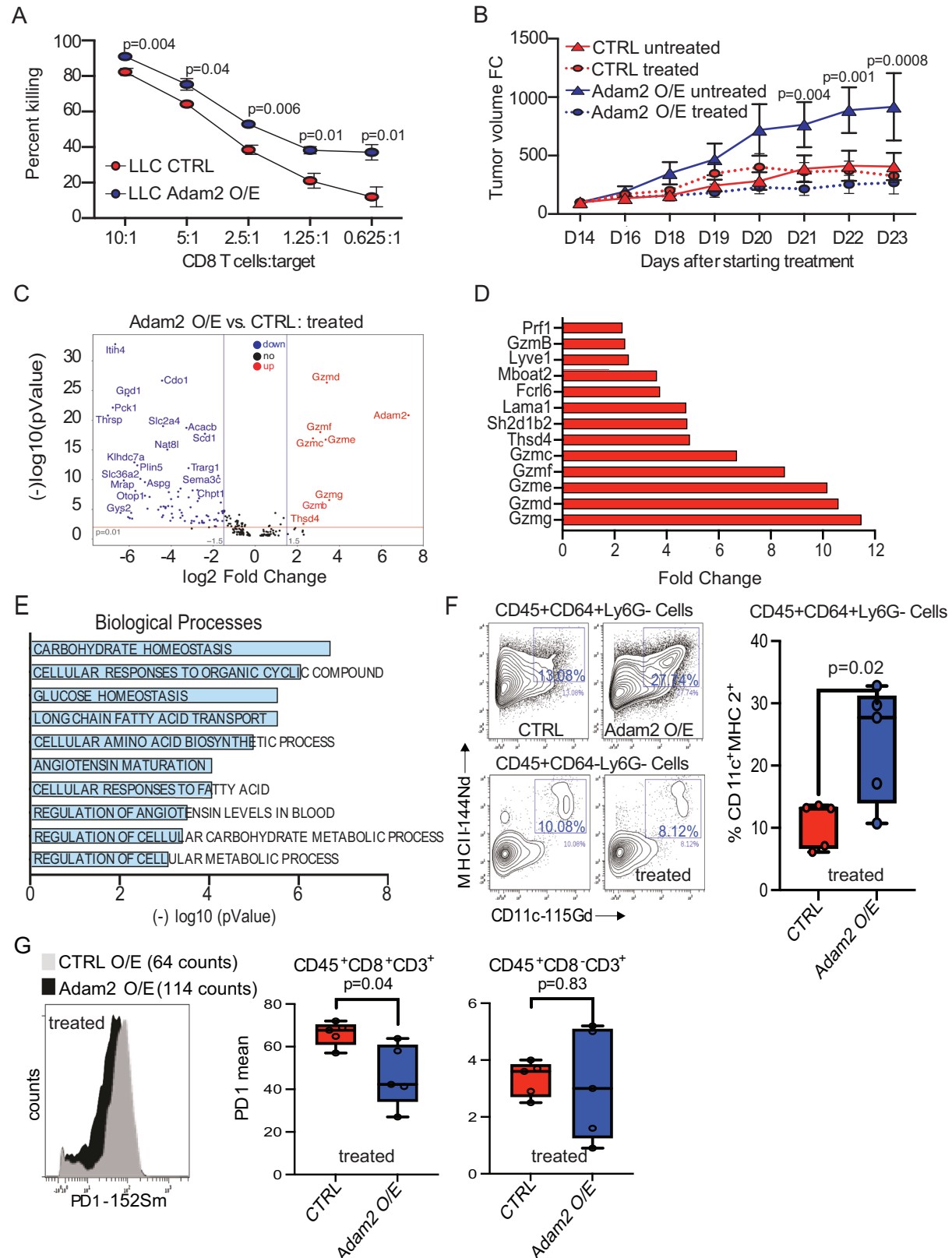

(mutations, CNV, expression) that correlate with cytotoxic index. To functionally assess the tumorigenic and immune-regulatory potential of these genes, we have established a corresponding, multiplexed, and barcoded lentiviral CRISPR knockout library that allowed us to model loss-of-function of these 600 genes a direct and autochthonous manner in the lung of *Kras*[G12D] and *Braf*[V600E] mice.

Our top immune-resistance gene is *Serpinb9*. SERPINB9 directly inhibits GZMB and is highly expressed in cytotoxic T-cell and NK cells to protect those cells from their own GZMB. However, tumor cells hijack this system to protect themselves from cytotoxic attack. Interestingly, Jiang, P et al. identified SERPINB9 as an immunosuppressor using a computational method integrating data from the

**Fig. 5 | ADAM2 increases cytolytic activity of TAA-specific CD8 T cells. A** The effect of Adam2 overexpression in LLC cells on CD8⁺ OT-I T-cell-mediated killing. CTRL or Adam2 O/E cells labeled with CFSE were used as targets (*T*) and cocultured with activated OT-I CD8 T cells (*E*) at different *E*:*T* ratios. Flow cytometry analysis was used to calculate the ratio of killed cells at different *E*:*T* ratios. Data presents mean ± s.e.m. of four technical replicates analyzed by two-sided student's *t*-test examined. One representative experiment out of three independent experiments is shown. **B** Tumor growth of CTRL or Adam2 O/E LLC cells allografted into C57BL/6 mice, which were subsequently treated with OT-I T cells (*n* = 4) or left untreated (*n* = 4). Three independent experiments were performed. **C** Volcano plot showing differentially expressed genes in OT-I treated CTRL and Adam2 O/E tumors isolated from C57BL/6 mice where log2 FC indicates the mean expression and (−)log10 adjusted *p* value level for each gene. The blue dots denote downregulated gene expression, the red dots denote upregulated expression, and black dots denote the gene expression without marked difference. Data represents three independent biological samples for each group. **D** Bar plot showing 12 top upregulated genes between OT-I treated Adam2 O/E versus CTRL tumors isolated from C57BL/6 mice. Data represents three independent biological samples for each group. **E** Bar graph showing Gene Ontology of the DEGs (FC > 2, *p* < 0.05) downregulated in OT-I treated Adam2 O/E compared to CTRL tumors assigned to Biological Process. Data represents three independent biological samples for each group. **F** CyTOF analysis showing the expression of MHCII and CD11c in CD45⁺ CD64⁺ Ly6G⁻ cells from OT-I treated CTRL and Adam2 O/E tumors isolated from C57BL/6 mice (*n* = 6 biologically independent samples for each group examined over two independent experiments). Box and whiskers plots illustrate the median, first and third quartiles, maximum and minimum of relative MHCII and CD11c abundance between two groups analyzed by two-sided student's *t*-test. **G** CyTOF analysis showing PD1 surface expression on pre-gated CD45⁺CD8⁺CD3⁺ cell population from OT-I treated CTRL and Adam2 O/E tumors isolated from C57BL/6 mice. Bar plots representing the frequency of PD1 surface expression on CD45⁺CD8⁺CD3⁺ and CD45⁺CD8⁻CD3⁺ cells from OT-I treated CTRL and Adam2 O/E tumors isolated from C57BL/6 mice (*n* = 6 for each group examined over two independent experiments). Box and whiskers plots illustrate the median, first and third quartiles, maximum and minimum of relative PD1 abundance between two groups analyzed by two-sided student's *t*-test.

TCGA, PRECOG and METABRIC, which was further substantiated by Jiang, L et al. showing that SERPINB9 inhibition might constitute a viable therapeutic avenue[46,47]. Here, we show an autochthonous mouse model of *Serpinb9* loss in the lung, corroborate their data, and further extend their findings showing that combining SERPINB9 inhibition with TAA-directed T-cell therapy could yield superior efficacy.

Our top enriched sensitizing gene is *Adam2*. Adam2 is a putative cancer-testis antigen, albeit its relevance is still being revealed[67,98]. We found that *ADAM2* is expressed in ~16% of human LUAD patients and its expression correlates with *ADAM2* copy number gain or amplification. Similar to our mouse model, we found significant reduction of interferon and TNF cytokine signaling in the 13.9% of the TCGA LUAD patients with *ADAM2* expression. Interestingly, directly adjacent to *ADAM2* lies *IDO1* and *IDO2* and all three genes are commonly co-amplified in cancer and their amplification is anti-correlated with the cytotoxic index in human tumors[45]. Both, *IDO1* and *IDO2* are known to suppress T-cell immune responses indicating the existence of a strong immune-suppressive gene island on chromosome 8p11[21–23].

In mouse tumor models, we could show that *Adam2* expression is induced in *Kras*^G12D but not *Braf*^V600E tumors and its expression is further increased upon immunotherapy, indicating that *Adam2* is an oncogene- and immune-responsive bona fide cancer-testis antigen. Using loss-of-function and gain-of-function experiments, we showed that Adam2 expression had a dramatic effect on tumor growth by restricting type I and II IFN responses as well as several other cytokine signaling pathways thereby restraining the endogenous immune surveillance machinery, affecting predominantly T cells (please see schematic in Supplementary Fig. 38). It is well-documented that IFNs are required to augment immune functions and tumors lacking IFNs expression fail to mount an antitumor immune response and thus grow unrestricted, which is precisely what we observed in Adam2 overexpressing tumors. In contrast, we observed increased interferon and cytokine profiles and increased CD8 T cells in *Adam2* knockout tumors concomitant with reduced tumor growth. We thus described a cancer-testis antigen with not only immunogenic but also with strong immune-modulatory and tumor-promoting functions.

Interestingly, we also found that Adam2 expression not only reduces endogenous cytokine signaling and T-cell responses, but also paradoxically enhanced cytotoxicity of adoptively transferred TAA-specific cytotoxic T cells as well as immune-checkpoint blockade. One possible explanation is that the reduced endogenous interferon responses, which at first allow the tumors to evade the immune surveillance, also leads to reduced expression of interferon-inducible checkpoint molecules such as Pdl1, Lag3, Tigit and Tim3 in established tumors. As such, Adam2 expression and the associated decreased IFN signaling appears to result in a less exhausted tumor microenvironment, which is highly permissive to ex vivo expanded, adoptively transferred cytotoxic T cells or rejuvenated endogenous cytotoxic T cells. Conversely, *Adam2* knockout tumors exhibited increased CD8 T-cell exhaustion presumably due to the prolonged IFN signaling and concomitant reduced ACT efficacy (Supplementary Fig. 38). While the immune stimulatory roles of IFNs are better known, the immune-suppressive immunoregulatory effects of persistent IFN signaling are being increasingly recognized not only during chronic viral infection[99–101] but also in cancer[50,51,102,103]. Consistently, several functional screens have shown that blocking tumor interferon-γ (IFNγ) signaling sensitizes tumors to immune-checkpoint blockade[35,37,38]. However, to our knowledge no screen so far has picked up Adam2 as immunomodulator. This might be routed in the fact that conventional screens using pooled cell populations are limited to assessing cell autonomous functions. In contrast, our method generates individual clones of knockout cells within the lung and thus is perfectly suited to assess non-cell autonomous functions such as Adam2' effect on cytokine signaling. Further studies will be needed to elucidate the precise mechanism of how Adam2 regulates interferon and cytokine signaling as well as activation of cytotoxic T cells.

These findings were further corroborated by whole transcriptome analysis of 510 low versus high *ADAM2* expressing TCGA LUAD cases where analysis of the 50 cancer hallmark pathways identified suppression of IFNγ, TNFα via NFκβ, IL6/JAK/STAT3 signaling and complement pathways in ADAM2^hi-expressing tumors. IFNγ signaling in particular has emerged to have dual function – pro and antitumor activity[50–52,104]. Our results indicate that Adam2-mediated reduction of IFNγ signaling facilitates tumor growth primarily through downregulation of pathways associated with antigen presentation and overall activation of endogenous antitumor T-cell responses. In contrast, adoptive transfer of in vivo activated antigen-specific CD8 T cells allowed for more efficient recruitment (CXCL9, CXCL10), cross-presentation of TAA, checkpoint inhibition (PD1), and cytotoxic function.

Clinically, our data indicate that patients with tumors expressing ADAM2 would likely respond favorably to immune-checkpoint blockade, CAR-T-cell therapy or transfer of ex vivo expanded and activated autologous TAA-specific T cells. In addition, inhibiting ADAM2 surface expression or treatment with ADAM2 blocking antibodies (alone or in combination with immune-checkpoint inhibitors) might represent a therapeutic avenues for a substantial proportion of cancer patients whose tumors have aberrant ADAM2 expression to reinvigorate their endogenous immune responses to combat their cancers.

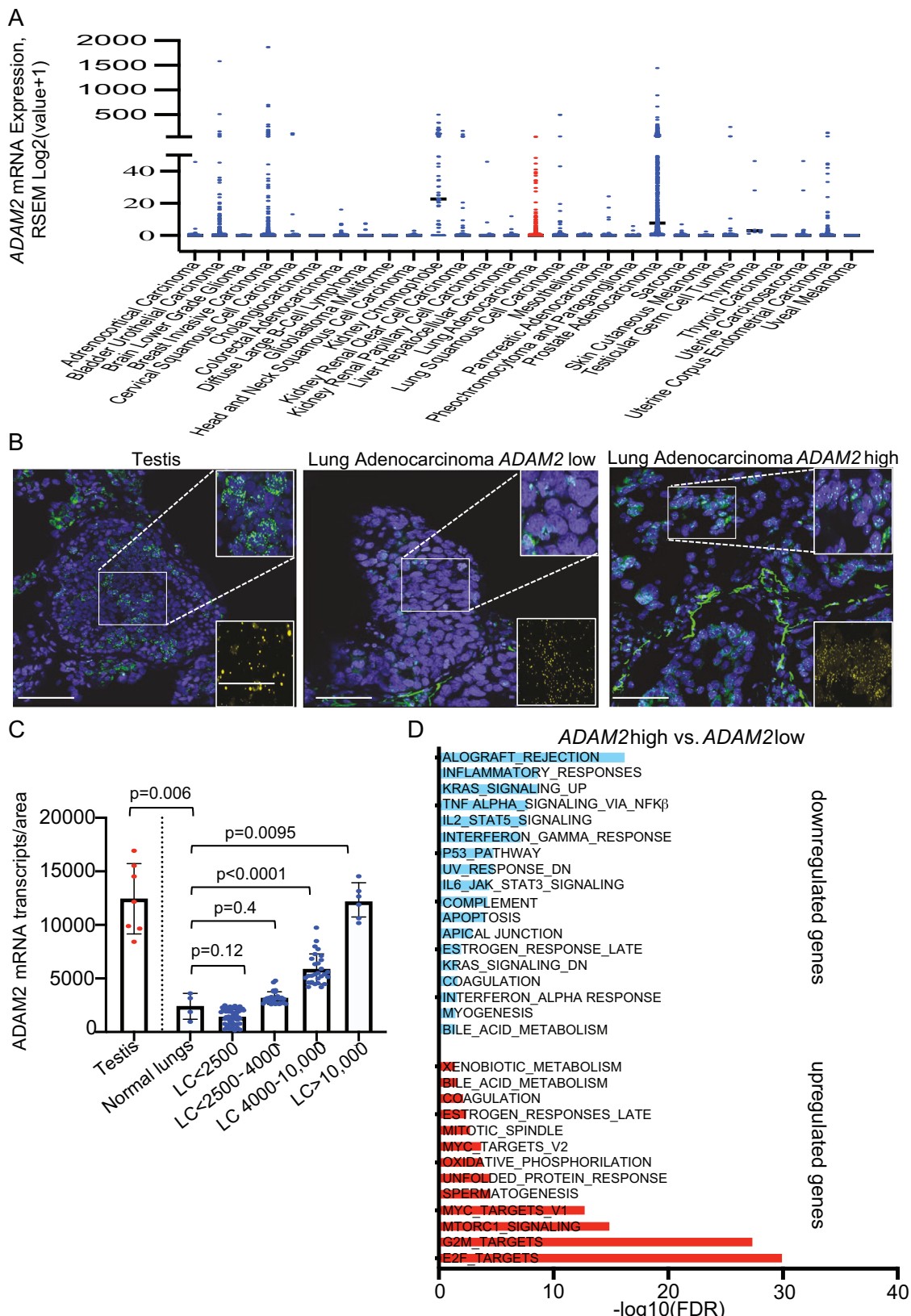

**Fig. 6 | ADAM2 is a bona fide cancer-testis antigen associated with reduced inflammatory responses and increased cancer hallmark pathways in human LUAD. A** Pan-cancer TCGA analysis reveals increased ADAM2 mRNA expression in multiple cancers. **B** RNAscope analysis of ADAM2 (green) and EPCAM (yellow) expression in testis and LUAD samples. **C** Quantification of ADAM2 transcripts from (**B**) in indicated tissues ($n = 5$ normal lungs; $n = 8$ testis; $n = 96$ LUAD). Data were analyzed by two-sided student's $t$-test and present mean ± s.e.m. **D** Pathway enrichment analysis of Cancer Hallmark gene sets in 510 TCGA LUAD samples. Bar plots showing the pathways that were significantly enriched for upregulated and downregulated genes in the TCGA LUAD cohort (FDR < 0.05). Pathways enriched for upregulated genes are shown in red and pathways enriched for downregulated genes are shown in blue. -log10 transformed FDR-adjusted $p$ values are shown on the $x$-axis.

Overall, our study highlights how direct in vivo CRISPR/Cas9-mediated gene editing can be used to integrate cancer genetics with mouse modeling to elucidate how cancer associated genetic alterations control response to immunotherapies and to identify additional reputed immunotherapy targets.

## Methods

This study has been approved by the Ethics Board of the Toronto Centre for Phenogenomics, Lunenfeld-Tanenbaum Research Institute, Mount Sinai Hospital, Toronto, Ontario, Canada.

### Animals

Equal numbers of male and female animals were used throughout the study without any bias. Animal husbandry, ethical handling of mice and all animal work were carried out according to guidelines approved by Canadian Council of Animal Care and under protocols approved by the Centre for Phenogenomics Animal Care Committee (18-0272H). The animals used in this study, LSL-Kras$^{G12D}$ (008179)[105], LSL-Braf$^{V600E}$ (017837)[106], R26-LSL-CAS9-GFP (026175), FVB.129S6(B6)Gt(ROSA)26Sor$^{tm1(Luc)Kael}$/J (005125), C57BL/6-Tg(TcraTcrb)1100Mjb/J (003831, also known as OT-I), Gt(ROSA)26Sor$^{tm4(ACTB-tdTomato,-EGFP)Luo}$/J (007576, also known as mT/mG), Gt(ROSA)26Sor$^{tm1(CAG-Brainbow2.1)Cle}$/J (013731, also known as R26R-Confetti), NOD.Cg-Prkdc$^{scid}$ IL2rg$^{tm1Wjl}$/Sz/J (005557, also known as NSG), NU/J (002019, known as Nude) and C57BL/6J (000664) were purchased from the Jackson Laboratory. CRISPR screen in the LSL-Kras$^{G12D}$-CAS9-LUC and LSL-Braf$^{V600E}$-CAS9-LUC was performed in a F1 FVBNxC57BL/6J background. Genotyping was performed by PCR using genomic DNA prepared from mouse ear punches. All details of the used animal models are listed in Supplementary Data 8.

### Cell lines

A549 and H125 cell lines were purchased from ATCC. The 293FT cell line was purchased from Thermo Fisher Scientific. LLC1 cancer cell line was received as a gift from Dr. Hansen He (The Princess Margaret Cancer Centre). All cell lines were tested for mycoplasma and cultured in DMEM (Gibco) media supplemented with 10% fetal bovine serum (FBS) and antibiotics. For some experiment, puromycin or blasticidin selection at 10 µg/ml was used.

### Lentiviral constructs and library construction

pLKO-sgRNA-Cre plasmid[107] was modified to express OVA-peptide SIINFEKL (gBlock gene fragments, IDT technologies), hereafter pLKO-sgRNA-Cre-P2A-OVA. sgRNAs targeting mouse homologs of human genes associated with immune cytolytic activity, selected from Rooney et al.[45], were obtained from Hart et al.[108] (4 sgRNAs/gene) and non-targeting sgRNAs were obtained from Sanjana et al.[109], ordered as a pooled oligo chip (CustomArray Inc., USA) and cloned into sgRNA-Cre-P2A-OVA using BsmBI restriction sites. For validation of top enriched hits, individual sgRNA 1 and 2 targeting SerpinB9 and Adam2 were ordered from Sigma and cloned into the same sgRNA-CRE-P2A-OVA plasmid backbone using BsmBI site. We excluded frequent and known immune-checkpoint inhibitors such as CTLA4, PDCD1(PD1), PDL1 from the immune genes library. The non-targeting sgRNAs as well as an sgRNA (actgccataacacctaactt) targeting the permissive TIGRE locus[110] were designed not to target in the mouse genome as negative control.

For overexpression analysis, V5 tagged mouse Adam2, human ADAM2, and human SERPINB9 were obtained from the ORFeome collaboration provided by Fritz Roth, originally from CCSB, DFCI, Harvard. Codon-optimized mouse SerpinB9 was ordered as gBlock fragment from Twist Bioscience. These were cloned into original PLX306 (kindly provided by David Root; Adgene #41391) or modified to express OVA-peptide SIINFEKL (PLX306-puromycin-P2A-OVA) construct using getaway cloning system. For in vivo experiments puromycin was replaced by iCRE.

### Lentivirus production and transduction

Large-scale production and concentration of lentivirus were performed as previously described[111–114]. Briefly, 293FT cells (Invitrogen R700-07) were seeded on a poly-L-lysine coated 15 cm plates and transfected using PEI (polyethyleneimine) method in a non-serum media with lentiviral construct of interest along with lentiviral packaging plasmids psPAX2 and pPMD2.G (Adgene plasmid 12259 and 12260). Eight to 12 h post transfection media was added to the plates supplemented with 10% Fetal Bovine Serum (FBS) and 1% Penicillin-Streptomycin antibiotic solution (w/v). After 48 h, the viral supernatant was collected and filtered through a Stericup-HV PVDF 0.45-µm filter, and then concentrated ~2000-fold by ultracentrifugation in an MLS-50 rotor (Beckman Coulter). Viral titers were determined first in vitro by infecting the R26-LSL-tdTomato MEFs and FACS based quantification. Viral titers were further determined in vivo using R26-LSL-Confetti mice, which function as a stochastic multicolor Cre-recombinase reporter strain. Upon Cre-mediated recombination, one out of four fluorescent proteins are expressed from a single genomic locus, which enables labeling and lineage-tracing individual recombined cells, and as such, we used these mice to determine the viral concentration required to transduce lung epithelium at clonal density in vivo. We used intranasal instillation at P2 (the saccular stage of lung development) to administer LV-sgRNA-Cre-OVA lentivirus to the lungs of (LSL)-Kras$^{G12D}$ or LSL-Braf$^{V600E}$ mice crossed to multicolor LSL-Confetti mice. Cre-mediated excision of the LSL-cassettes induced activation of Kras$^{G12D}$ or Braf$^{V600E}$ as well as expression of one of the four fluorescent proteins encoded in the Confetti reporter cassette. In vivo viral transduction efficiency was determined by injecting decreasing amounts of a single viral aliquot of known titer, diluted to a constant volume of 2× and 10× per intranasal instillation at P2. The percent of infection was analyzed by BLI measuring the total flux [p/s] in lung. Ten microliters of $1-1.5 \times 10^8$ PFU at P2 was determined as optimal concentration for efficient induction of lung tumorigenesis by week 3–4 in Kras and Braf LUAD mouse models. The intranasal instillation of lentivirus into 2-day-old mouse pups was chosen as this significantly increases the number of cells that can be transduced and number of tumors that form in the mouse lungs of (LSL)-Kras$^{G12D}$ or LSL-Braf$^{V600E}$ mice, which is a pre-requisite to screen a complex sgRNA library and achieve adequate coverage.

### Deep sequencing: sample preparation, pre-amplification, and sequence processing

Genomic DNA from epithelial and tumor cells were isolated with the DNeasy Blood & Tissue Kit (Qiagen). Genomic DNA concentration was quantified using Qubit dsDNA BR Assay (cat no. Q32853). Forty micrograms genomic DNA of each lung tumor ($n = 20$) was used as template in a pre-amplification reaction by nested primers v2.1-F1 gagggcctatttcccatgattc and v2.1-R1 gttgcgaaaaagaacgttcacgg with 25 cycles and Q5 High-Fidelity DNA Polymerase (NEB), followed by unique barcoded primer combination for pool of all individual 50 µl reactions for each genomic DNA sample. Five microliters of PCR1 product was run on a 1% agarose gel to visualize a product of ~600 bp. Five microliters of PCR1 product as template was amplified using unique i5 and i7 index primer combinations with 8 cycles and Q5 High-Fidelity DNA Polymerase (NEB) for each individual sample to allow pooling of sequencing libraries. The following primers were used:

FW:5′AATGATACGGCGACCACCGAGATCTACAC**TATAGCCT**ACACTCTTTCCCTACACGACGCTCTTCCGATCTtgtggaaaggacgaaaCACCG-3′

RV:5′CAAGCAGAAGACGGCATACGAGAT**CGAGTAAT**GTGACTGGAGTTCAGACGTGTGCTCTTCCGATCTATTTTAACTTGCTATTTCTAGCTCTAAAAC-3′

The underlined bases indicate the Illumina (D501-510 and D701-712) barcode location that were used for multiplexing. PCR products were run on a 2% agarose gel, and a clean ~200 bp band was isolated

using Zymo Gel DNA Recovery Kit as per manufacturer instructions (Zymoresearch Inc.). Final samples were quantitated using Qubit dsDNA BR Assay and sent for Illumina Next-seq sequencing (20 million reads per 5 pooled lungs—2 × 5 untreated and 2 × 5 treated) to the sequencing facility at Lunenfeld-Tanenbaum Research Institute (LTRI). Sequenced reads were aligned to sgRNA library using Bowtie version 1.2.2 with options −v 2 and −m 1. CRISPR screen hits were obtained and identified using Model-based Analysis of Genome-wide CRISPR-Cas9 Knockout (MAGeCK) Robust Rank Aggregation (RRA) platform[115]. Importantly, the in vivo screen generated both positive and negative gene profiles (FDR < 0.25) with respective enrichment or depletion of at least two sgRNAs per gene.

### Analysis of genome editing efficiency
LSL-Cas9-GFP MEFs were cultured and infected with lentivirus carrying Cre and corresponding sgRNAs. Cells were live sorted for GFP expression and expanded further to extract genomic DNA using DNeasy Blood & Tissue Kit (Qiagen). Genomic DNA from tumors (GFP sorted and/or unsorted cells were used) from the mice injected with single sgRNAs was also isolated using the same kit. PCR was performed flanking the regions of sgRNA on genomic DNA from WT MEFs, cells infected with respective virus or tumors and sent for Sanger sequencing. Sequencing files along with chromatograms were uploaded to https://www.deskgen.com/landing/tide.html or https://ice.synthego.com/#/ and genome editing efficiency was estimated.

### In vitro T-cell activation and expansion
Mouse T cells: Spleens were harvested from OT-I mice and dissociated to obtain single-cell suspension. Red blood cells were lysed with ACK lysis buffer. Cells were resuspended at $1 \times 10^6$ cells/ml in T-cell media [RPMI-1640 (Gibco) + 10%FBS + 1% penicillin/streptomycin + 40 μM 2-β-mercaptoethanol (Sigma-Aldrich)]. Medium was supplemented with 2 μg/ml of SIINFEKL (OVA 257–264, AnaSpec) and human interleukin-2 (hIL-2, PeproTech) at 30 U/ml. After 2 days, equal amount of new medium supplemented with IL-2 was added. Cells were used for in vitro assays following 4 days activation. For Supplementary Fig. 3a: Ex vivo OT-I were isolated from spleen and LNs and purified using CD8a (Ly-2) microbeads (#130-117-044, Miltenyi Biotech). CD8+ OT-I cells were labeled with 1 μM CFSE (CellTrace™ CFSE proliferation kit, Molecular Probes) and cultured in triplicates with plate-bound anti-CD3 (5 μg/ml) and anti-CD28 (2 μg/ml) mAb in 200 μl of T-cell media supplemented with 5 ng/ml of IL-7 and 30 U/ml of IL-2 (PeproTech) for 4 and 7 days. Activation and proliferation of CD8+ OT-I cells were determined by flow cytometry.

Human T cells: As described previously, activated and expanded day 15 human DNT (γδ) cells were kindly provided by Dr. Li Zhang's lab (Toronto General Hospital, UHN)[116]. Briefly, human blood was collected from healthy adult donors after receiving informed consents in accordance with UHN Research Ethics Board (05-0221-T) and NHLBI approved protocols. DNTs were enriched by depleting CD4+ and CD8+ cells using RosetteSep™ human CD4- and CD8-depletion cocktails (Stemcell Technologies). The CD4 and CD8 depleted cells were cultured in 24-well plates pre-coated with 5 μg/ml anti-CD3 antibody (OKT3, eBioscience) for 3 days in RPMI-1640 (Gibco) supplemented with 10% FBS (Sigma) and 250 IU/ml IL-2 (Proleukin). Fresh IL-2 and OKT3 were added to the DNT cultures every 2–4 days. After 10 days of activation/expansion, 0.1 μg/ml of OKT3 250 IU/ml IL-2 was added to culture every 2–4 days. DNTs were harvested between day 15–20 and purity was assessed by flow cytometry prior to experiments. The mean purity of DNTs used in the study was ~94%, and cells were used for functional studies.

### Synthetic peptides and inhibitors
Synthetic peptides were generated by AnaSpec and purified with HPLC to ≥ 95 % purity and verified by Mass Spectrometry.

H2K$^b$-restricted peptide epitope of OVA (257–264) · SIINFEKL were dissolved in PBS at a concentration of 2 mg/ml. For in vivo experiments mice were i.p. immunized with 100 μg of OVA (SIINFEKL) emulsified in Complete Freund's adjuvant (CFA) or Incomplete Freund's adjuvant (IFA). Blocking antibodies CTLA4 (clone 9H10), PDL1 (clone 10 F.9G2) were obtained from BioXcell and administered every other day throughout the experiment at a dose of 150 μg per mouse per treatment. All procedure were performed in accordance with TCP SOP #SAF034.

### Adoptive T-cell transfer (ACT)
For Supplementary Fig. 3b–d: CD8 T cells were isolated from gender-matched OT-I spleen and LNs using CD8a (Ly-2) microbeads (#130-117-044, Miltenyi Biotech). $10 \times 10^6$ OT-I cells were IV injected (groups 2 and 3) into C57BL6/J mice followed by i.p. injection of 100 μg of OVA (SIINFEKL) emulsified in CFA and IFA (d7). Spleens were isolated 3 days after immunization. Activation and proliferation of CD8+ OT-I cells were determined by flow cytometry.

For lung cancer treatments: ACT of gander-matched, purified $15–20 \times 10^6$ OT-I cells were i.v. transferred into 3–4 weeks old LSL-Kras$^{G12D}$-CAS9 -LUC or LSL-Braf$^{V600E}$-CAS9-LUC mice after lung tumor induction, which was assessed by bioluminescence imaging (BLI). Mice were immunized on day 1 (100 μg of OVA in CFA) and day 7 (100 μg of OVA in IFA) after ACT. In some experiments, spleen, LNs, and/or lungs were isolated for total cell number enumeration and flow cytometric analysis on day 7 or at endpoints. For tumor tissues, the entirety of each sample was acquired and the total number of CD3+CD8+Tom- T cells and transferred OT-I cells was assessed. In some experiments lungs were perfused with 4% PFA and isolated for H&E staining and immunofluorescence analysis. All procedure were performed in accordance with TCP SOP #SAF034.

### In vitro killing assay
**Mouse T cells.** Four days activated and expanded splenocytes (effectors, E) from OT-I crossed to mT/mG mice were cocultured with pre-plated 20,000 LLC transduced with either sgNTC, sg sg*Serpinb9*, LLC-CTRL-SIINFEKL or ADAM2-SIINFEKL cells (targets, T) and labeled with 1 μM CFSE (Molecular Probes) in triplicates in 48-well plates at varying E:T ratios for 4 h. To determine the cytotoxicity induced by CD8 T cells, drop in CFSE (as read out of targets being killed) was measured by flow cytometry. Following formula was used: percent of killing = 100 · total number of PI⁻CFSE⁺ cells with effectors/total number of PI⁻CFSE⁺ without effectors × 100%.

**Human T cells.** Donor-derived (UPN119, UPN133) day 15–20 activated and expanded DNT (γδ) T cells (effectors, E) were cocultured with CFSE (1 μM) labeled 20,000 pre-plated human lung cancer cells (A549-*SERPINB9* or H125-*SERPINB9*) in triplicates in 48-well plates at varying E:T ratios for ~16–18 h (overnight). The cytotoxicity induced by DNTs was determined by flow cytometry where percent of killing = 100 − total number of PI⁻CFSE⁺ cells with DNTs/total number of PI⁻CFSE⁺ without DNTs × 100%.

### Western blotting
Cell lysates were generated using RIPA lysis buffer supplemented with protein inhibitor (Roche, #4693159001) and protein concentration was determined using Pierce™ BCA protein assay. Denatured lysates (20 μg) were applied to a 4–15% Mini-PROTEAN TGX precast protein gels (Biorad) or to a 10% SDS-PAGE and blotted using standard procedures. For protein detection, blots were incubated with primary V5 tag monoclonal (#R960-25, Life Technologies) antibody overnight and with secondary antibody (goat anti-mouse IgG-HRP (#1706516, Biorad)) for 1 h. Direct-Blot™ HRP anti-GAPDH (#607904, Biolegend) was used as a loading control. Chemiluminescence was used to visualize the protein bands (Biorad). In some experiments tumors were

first sonicated in RIPA buffer and protocol described above for western blotting was followed.

## Tumor implantation

Mice were anesthetised with 2–2.5% isoflurane with oxygen. LLC-CTRL-SIINFEKL or LLC-ADAM2-SIINFEKL cell lines were gently injected subcutaneously in the upper right backside of mice. $1 \times 10^5$ cells were injected per mouse for all the experiments. After tumor onset, tumor growth was measured every day or every second day by digital caliper. Intravenous injection of $15–20 \times 10^6$ OT-I was performed on D19 after tumor reached the volume between 100–600 mm³, followed by immunization with OVA/CFA on day 1 and OVA/IFA on day 7. Humane intervention points were called when tumor size reached 1700 mm³, according to SOP AH009. Tumor-bearing NSG mice were treated with $20 \times 10^*6$ activated T cells after tumors have reached size between 100 and 400 mm³. C57BL/6J mice were treated with anti-PDL1 (cat# BP0101), anti-CTLA4 (cat# BP0164) or IgGg2a (cat# BE0090) isotype control, 150 μg/mouse every second day for 2 weeks. The maximal tumor size permitted by the TCP ethics committee is 1700 mm³ and this maximal tumor burden was not exceeded other than in rare instances where tumors grew in severely immune-compromised NSG mice, where tumors grew exponentially and even within 1 day could rapidly exceed this maximum.

## Preparation of tumor tissues for flow cytometry

Tumor tissues (fresh or frozen) were minced into small pieces using surgical blade and scalpel (08-957-5D, Fisher Scientific) and processed using the tumor dissociation kit (130-096-730, Miltenyi Biotec) as recommended by supplier. Single-cell suspensions from lung or implanted tumors were obtained using kit and the gentleMACS Octo Dissociator. Dissociated cells were passaged through 70 μm cell strainer (BD), collected in a 50 ml falcon tube, and resuspended in staining buffer (1%BSA in PBS).

## Flow cytometry

Single-cell suspensions from spleen, LNs, lung or implanted tumors were washed with FACS buffer (DPBS + 1%BSA + 1 mM EDTA + 0.1% sodium azide), incubated with FC block (CD16/32) for 30′ at 4 °C, stained with appropriate antibodies and washed twice in FACS buffer. Dead cells were excluded from all data by forward and side scatter and 4′,6-diamidino-2-phenylindole, dihydrochloride (DAPI, Molecular Probes−5 mg/ml, used 1/50,000) or fixable viability dye eFluor™ 450 (1/1000, Invitrogen). Doublets were excluded by forward scatter with and side scatter with and height. Cells were stained by standard staining techniques and analyzed on Fortessa flow cytometer (BD Biosciences). For intracellular staining, the cells were permeabilized using eBioscience™ Foxp3/Transcription factor Staining Buffer Set (00-5523-00,Thermo Fisher Scientific) according to manufacturer's recommendation. Data Files were analyzed using Flow-Jo (Tree Star).

## Immunofluorescence

Cells or tissue sections were fixed with 4% paraformaldehyde for 10 min. Following fixation, slides were rinsed 3 times with PBS for 5 min, permeabilized using 0.5% Tween-20 in PBS at 4 °C for 20 min and rinsed with 0.05% Tween-20 in PBS for 3 × 5 min at room temperature. Samples were blocked at room temperature with blocking serum (1% BSA, 1% gelatin, 0.25% goat serum 0.25% donkey serum, 0.3% Triton-X 100 in PBS) for 1 h. Samples were incubated with primary antibody diluted in blocking serum overnight at 4 °C followed by 3 washes for 5 min in PBS. Secondary antibody was diluted in blocking serum with DAPI and incubated for 1 h at room temperature in the dark. Following incubation, samples were washed 3 times for 5 min in PBS. Coverslips were added on slides using MOWIOL/DABCO based mounting medium and imaged under microscope next day. For

quantification, laser power and gain for each channel and antibody combination were set using secondary antibody only as control and confirmation with primary positive control and applied to all images. Images were captured and expression of the specified genes were quantified with fluorescent Nikon eclipse Ti inverted microscope.

## RNAscope

Custom-designed 20 ZZ probe targeting 1279-2224 bp of NM_009618.3 mouse Adam2, named mm-Adam2 (cat# 1113181-C3); catalog probes eGFP (cat # 538851-C1) and tdt Tomato-C2 (cat# 317048-C2) were ordered from Advanced Cell Diagnostics, ACD. Adam2, eGFP, Tom in situ hybridization was measured by RNAscope assay (Advanced Cell Diagnostics, ACD) according to the manufacturer's protocol using RNA scope^R Multiplex Fluorescent Reagent kit v2 (323-100). Briefly, paraffin-embedded normal (negative ctrl) and tumor-bearing lungs isolated from LSL-Kras^{G12D}-CAS9-LUC mice were cut into 5 μm sections and hybridized at 40 °C for 2 h. Adam2 ORF expressing tumor-bearing lungs or normal testis were used as positive controls. Hybridization signal was amplified using AMP 1, 2, 3 and developed using appropriate HRP signal. Images were captured with fluorescent Nikon eclipse Ti2 inverted microscope.

For human samples, custom-designed 20 ZZ probe targeting 63-1542 bp of NM_001464.5 human ADAM2, named hsADAM2, catalog probe (cat # 1141961-C1) and catalog probe targeting EPCAM (cat # 310288-C2) were ordered from Advanced Cell Diagnostics, ACD. Human paraffin-embedded tissue arrays for testis (cat#TE481A, https://www.tissuearray.com/tissue-arrays/Testis/TE481a) and for lung adenocarcinoma (cat #LC10013c, https://www.tissuearray.com/tissue-arrays/Lung/LC10013c) were ordered from US biomax.

## RNA sequencing

Total RNA was prepared from tumor tissues using TRIzol reagent (Invitrogen) or the Quick-RNA MiniPrep kit (R1055, Zymo Research) treated with ezDNase (Invitrogen). The RNA samples were quality checked by LTRI Sequencing facility using 5200 fragment analyzer system, with all samples passing the quality threshold of RQN score >9.3 except two samples with a score of 7.6. The library was prepared using an Illumina TrueSeq mRNA sample preparation kit at the LTRI sequencing facility, and complementary DNA was sequenced on an Illumina Next-seq platform. Sequencing reads were aligned to mouse genome (mm10) using Hisat2/bowtie2 version 2.1.0 and counts were obtained using featureCounts (Subread package version 1.6.3). Differential expression was performed using DEseq2 release 3.8. Gene set enrichment analysis was performed using GSEA computational method software 4.2.0 released by Broad institute. Samples were processed using the Hallmark gene sets obtained from MSigDB (https://www.gsea-msigdb.org/gsea/msigdb).

## RNA isolation, cDNA synthesis, and real-time QPCR analysis

FACS-sorted tumor RNA samples were treated using TRIzol (Invitrogen), treated with ezDNase (Invitrogen), and reverse transcribed into cDNA using SuperScript IV VILO (Invitrogen). Primers were designed to span exon junctions using Primer3Plus and were validated. Real-time quantitative PCR (qRT-PCR) reactions were performed on a QuantStudio5 (applied Biosystems) in 384-well plates containing 6.25 ng cDNA, 150 nM of each primer, and 5 μl 2X PowerUp Syber Green Master Mix (Applied Biosystems) in a 10 μl total volume. Relative mRNA levels were calculated using the comparative Ct method normalized to Ppib mRNA.

## Differential expression and pathway enrichment analysis of TCGA LUAD samples with ADAM2 expression

LUAD tumor samples were first split into two groups based on their expression of the *ADAM2* gene using consensus normalized RNA-seq

data of the TCGA PanCanAtlas project (unc.edu_PANCAN_IlluminaHi-Seq_RNASeqV2.geneExp_whitelisted.tsv). LUAD samples were assigned to the ADAM2-high group (FPKM-UQ ≥ 1) or the ADAM2-low group (FPKM-UQ < 1). The ADAM2-low group was n = 439 and ADAM2-high, n = 71. Unnormalized gene expression counts (STAR) for the TCGA LUAD cohort were then downloaded using the TCGAbiolinks R (v. 2.18.0)[117]. Differential gene expression analysis was performed on the unnormalized STAR counts by comparing the two sample groups using the DESeq2 R package (v. 1.30.1)[118]. The analysis focused on protein coding genes and filtered non-coding RNA genes. Two separate pathway enrichment analyses were conducted for genes that were upregulated and downregulated in the ADAM2-high group, respectively, using the ActivePathways R package (v. 1.1.1)[119]. Gene sets representing the 50 cancer hallmark pathways from the MSigDB database[120] were used in the enrichment analysis. A gene significance value cutoff of 0.05 and gene sets of 50 to 1000 genes were used as the parameters for ActivePathways. Significantly enriched pathways were highlighted (FDR < 0.05). Genes in each significant pathway were visualized with respect to log2-transformed fold change (FC) and significance (FDR) in ADAM2-high vs ADAM2-low LUAD samples.

**Pathway results.** Bar plots showing the pathways that were significantly enriched for upregulated and downregulated genes in the TCGA LUAD cohort (FDR < 0.05). Pathways enriched for upregulated genes are shown in red and pathways enriched for downregulated genes are shown in blue. -log10 transformed FDR-adjusted *p* values are shown on the x-axis.

**Pathway contributing gene plots.** For each pathway, all the genes that contributed to its enrichment are displayed. The FDR-adjusted p-value (-log10) for its differential expression is shown on the y-axis and the log2 transformed absolute value of its fold change is represented by the size of the circle. The absolute value of the log2 fold change was capped at 3 for visualization.

## Histology
**IHC.** Mouse lung tissues were perfused with and submerged in 4% PFA overnight at RT. Following fixation lungs were washed 3 times with PBS for 5 min, transferred to 70% EtOH (for long term storage or an immediate use), paraffin-embedded, and sectioned at 5 μm for IHC staining. Tissue sections were dewaxed in xylene and rehydrated in a graded series of alcohol (100%, 100%, 96%, 90%, 80%, 70% ethanol). For heat-induced epitope retrieval, tissue sections were incubated in Tris-EDTA buffer (pH 9.2) within a decloaking chamber (Biocare Medical) for 30 min in 95 °C. Tissue sections were then cooled and blocked in Tris buffered saline (TBS) containing 0.1% Tween-20, 3% bovine serum albumin, and 5% goat serum. Tissue sections were then incubated with specified antibodies listed in Supplementary Data 8. Quantification analysis was performed with QuPath software. Briefly, cell segmentation was performed based on optical density (OD), and positive cell detection was made by finding cutoff value with using the mean nuclear chromogen color intensity feature.

**Immunofluorescence.** Mouse lung tissues were perfused in 4% PFA overnight at RT. Following fixation lungs were washed 3 times with PBS for 5 min, embedded in OCT cryostat sectioning medium and stored at −80 °C until sectioned. The tissues were cut at 5 μm thickness, mounted onto poly-L-lysine coated slides, permeabilized and blocked at room temperature with blocking serum (1% BSA, 1% gelatin, 0.25% goat serum 0.25% donkey serum, 0.3% Triton-X 100 in PBS) for 1 h. Samples were incubated with primary antibody diluted in blocking serum overnight at 4 °C followed by 3 washes for 5 min in PBS. Secondary antibody was diluted in blocking serum with DAPI and incubated for 1 h at room temperature in the dark. Following incubation, samples were washed 3 times for 5 min in PBS. Coverslips were added on slides using

MOWIOL/DABCO based mounting medium and imaged under microscope next day. Antibodies used are listed in Supplementary Data 8. For quantification, laser power and gain for each channel and antibody combination were set using secondary only control and confirmation with primary positive control and applied to all images.

**IMC.** Tissues were formalin-fixed and paraffin-embedded, and 5um slices were mounted onto microscope slides. Tissue sections were dewaxed in xylene and rehydrated in a graded series of alcohol (100%, 100%, 96%, 90%, 80%, 70% ethanol). For heat-induced epitope retrieval, tissue sections were incubated in Tris-EDTA buffer (pH 9.2) within a decloaking chamber (Biocare Medical) for 30 min in 95 °C. Tissue sections were then cooled and blocked in Tris buffered saline (TBS) containing 0.1% Tween-20, 3% bovine serum albumin, and 5% goat serum. Tissue sections were then incubated with antibodies at a dilution of 5 μg/ml overnight in 4 °C. This was followed by incubation with $^{191}$Ir/$^{193}$Ir for 5 min in room temperature, washing three times in TBS, and air-dried prior to imaging. Antibody information including metal tag, clone, company, catalog number, and lot number can be found in Supplementary Data 8. Images were acquired using a Hyperion Imaging System (Fluidigm). Tissue sections were laser-ablated in a rasterized pattern at 200 Hz.

**IMC data analysis pipeline.** Data were preprocessed, segmented, and analyzed using an in-house integrated flexible analysis pipeline ImcPQ available at https://github.com/JacksonGroupLTRI/ImcPQ. The analysis pipeline is implemented in Python. Briefly, data were converted to TIFF format and segmented into single cells using the pipeline to classify pixels based on a combination of antibody stains to identify membranes/cytoplasm and nuclei. The stacks were then segmented into single-cell object masks. Single cells were clustered into cell categories based on pre-specified markers and clustered into immune cell types. To visualize number of cells per image the cell counts were normalized by the image area (total number of pixels) and displayed as cell density. *Data normalization and cell segmentation*: IMC raw data were converted to TIFF format without normalization. ImcPQ pipeline was used for segmentation and to process images to single-cell data. The analysis stacks were generated based on membranes/cytoplasm and nuclei markers. First, image layers, or channels, are split into nuclear or cytoplasm/membrane channels and added together to sum all markers that represent nuclei or cytoplasm/membrane. Then Mesmer model were used for segmentation as deep learning method[121]. The resulting single-cell mask was used to quantify the expression of each marker of interest and spatial features of each cell. Single-cell marker expressions are summarized by mean pixel values for each channel. The single-cell data were normalized and scaled per marker channel. *Clustering and quantification*: Single cells were clustered into groups of phenotypically similar cells using PhenoGraph v.2.0 as an unsupervised clustering method and then aggregation of these clusters into larger groups based on their mean marker correlations to identify cellular metaclusters. In a first step, the data were overclustered to detect and separate cell subpopulations. PhenoGraph was used with default parameters excluding DNA markers. For high-dimensional clustering, markers were used to cluster lymphoid (CD3+ or CD8+). Clusters with similar marker expression that represent the same established biologically phenotypes were merged. CD8+ cells were gated based on RFP and Granzyme B phenotypes. For list of markers used in clustering see Supplementary Data 8. For quantification, the normalized cell density (cell counts normalized by the image area) and their proportion among all cells analyzed are reported.

## CyTOF analysis
Cingle cell suspensions from implanted tumors (1–2 × 10⁶ cells per mouse) were washed with 1 mL staining media (SM: PBS+ 1% BSA) and

pre-treated with 2.4G2 antibody (BD Biosciences: 553141) to block FcRII/FcRIII receptors for 15′ at ambient temperature. Cells were then stained for 30′ at ambient temperature with a cocktail of 27 metal-tagged antibodies each diluted to pre-determined optimal concentrations in SM. As indicated in Suppl. Supplementary Data 8, purified carrier-free antibodies purchased from various vendors were metal tagged in-house using MaxPar X8 labeling kits (Standard Bio-Tools), except for CD45-89Y which was purchased directly from Standard BioTools. Cells were then washed by centrifuging through 10 volumes of SM at 300 × g for 5′, followed by two more washes in 2 mL protein-free phosphate buffered saline (PBS, Gibco: 20012-050) prior to staining dead cells for 5′ in 200 mL PBS containing 1 mM natural abundance Cisplatin (BioVision: 1550-1000). After two more PBS washes, cellular was labeled DNA with 100 nM [191/193]Iridium in PBS containing 0.3% saponin (Sigma-Aldrich: S7900) and 1.6% methanol-free ultra-pure formaldehyde (Analychem Corp.: 18814-20) for 24–48 h at 4 °C. Cells were then washed twice by centrifuging through PBS at 800 × g, 5′ prior to re-suspending in Maxpar Cell Acquisition Solution (Standard BioTools: 201241) with 4-element EQ normalization beads (Standard BioTools: 201078). Samples were acquired at -300 cells/second on a Helios instrument with a wide-bore injector according to the manufacturer's protocols. The Helios software (v6.7.1014) was used to generate and normalize FCS 3.0 data files which were then uploaded to CytoBank (Beckman Coulter Enterprise license) where manual gating was performed to remove debris, dead cells and doublets using standard techniques, prior to sequentially gating on specific populations as shown in Fig. 5F, G.

## Image generator
In Fig. 1A human body image is a modified version of image 'Designed by Freepik' - Image by Freepik.

## Statistical analysis
Statistical analyses were performed using the R, R studio software program (version 3.6.2, https://www.r-project.org/) and GraphPad Prism 9 (GraphPad software). Differences between groups were calculated by two-tailed Student's $t$-test, Wilcoxon Rank-Sum test (when data was not normally distributed) or log-rank test. All quantitative data are expressed as the mean ± SE. $P < 0.05$ was considered significant in all the analyses. Pearson correlation factor analysis was performed for measurement of sgRNA abundance, which was close to 1. Where applicable, FDR < 0.05 and $p < 0.05$ was considered significant: *$p < 0.01$, **$p < 0.001$, ***$p < 0.0001$, ****$p < 0.00001$. All experiments were performed using 2 to 3 independent replicates. No data were excluded from the analysis.

## Reporting summary
Further information on research design is available in the Nature Portfolio Reporting Summary linked to this article.

## Data availability
The RNA-seq data generated in this study have been deposited in the NCBI Gene Expression Omnibus GEO database under accession code GSE200628. The TCGA publicly available data used in this study are available in the TCGA database under [https://www.cancer.gov/ccg/research/genome-sequencing/tcga] and [http://www.cbioportal.org]. The remaining data are available within the article, Supplementary Information or Source Data file. Source data are provided with this paper.

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

## Acknowledgements

We thank all members of our laboratories for helpful comments and for their insight and assistance. We also thank The Centre for Phenogenomics, Network Biology Collaborative Centre and Flow Cytometry facility at LTRI as well as the Flow Cytometry Facility at the University of Toronto, CyTOF analysis platform at SickKids hospital, BioBox RNA-seq Data Analytics Platform and Histology core at UHN. Funding: This work was supported by a Krembil Foundation Grant, a CCSRI Innovation grant (#706454-1) and a CIHR project grant (#438792) to D.S. and a Terry Fox Research Institute Program Projects Grant to J.W. and D.S (TFRI Project #1107) and a Canadian Cancer Society Impact Grant (grant#: 704121) and a Canadian Institutes of Health Research (grant#: 419699) to L.Z. D.D. is a recipient of the TD and Medicine by Design Fellowship, S.K.L. is a Canadian Cancer Society Fellowship recipient (BC-F-16#31919). J.B. is a recipient of the Fonds de recherché du Québec – Santé (FRQS) Fellowship, and P.P.R. is a scholar of the FRQS.

## Author contributions

D.D. performed all experiments. E.C and H.W.J preformed IMC staining. A.A.M., S.A. and N.A. performed TCGA pathway analysis. J.R. contributed to supervision of TCGA bioinformatic analysis. D.K. and A.V. helped with pathology analysis. J.B. and P.P.R. preformed bioinformatic analyses. J.M.B., R.T., S.M and S.K.L. helped with mouse experiments. K.T. helped with microscopy quantification. A.A. and M.N. and J.W. helped with RNA scope technology. Y.L. helped with cloning and WB analysis. G.M. helped with genotyping and lab work. J-B.L. and L.Z. provided human T cells. C.G. performed CyTOF analysis. D.S. coordinated the project and together with D.D. designed the experiments and wrote the manuscript.

## Competing interests

The authors declare no competing interests.
