## [Peer Review File · Nature Communications]

In vivo CRISPR screens reveal Serpinb9 and Adam2 as regulators of immune therapy response in lung cancerREVIEWER COMMENTS

Reviewer #1 (Remarks to the Author): with expertise in lung cancer, cancer immunology

Dervovic et al evaluated the immune-modulatory capabilities of 573 genes (identified in a previous study) associated with altered cytotoxicity in human cancers using CRISPR/Cas9 in vivo screens in mouse lung cancer models. In addition to the known immune evasion factor Serpinb9, they identified a potential cancer testis antigen Adam2 as a new immune modulator. Adam2 was also found to be expressed, in part through gene amplification, in human cancer including LUAD. Studies in mice indicate that Adam2 functions as an oncogene as knockout of Adam2 attenuated tumor growth while overexpression accelerated tumor growth. These effects were diminished in immunodeficient mice suggesting a role of antitumor immune response. Importantly, knockout of Adam2 in tumors increased T cell infiltration and enhanced interferon I/II and TNF cytokine signaling. Paradoxically, Adam2 overexpression enhances ACT treatment response possibly by reducing expression of the immune checkpoint inhibitors (ICI) Pd-I1, Lag3, Tigit and Tim3 in the tumor microenvironment. Overall, this is a well-written and very well-performed study. The CRISPR/Cas9 in vivo screen has significant novelty by itself and will be of interest for cancer researchers. The discovery of Adam2 as a novel immune response modulator is also of significant interest. However, there are also concerns that need to be addressed.

1. The mechanism of action of Adam2 proposed by the authors is attenuation of interferon I/II and TNF cytokine signaling. Indeed, this may explain how Adam2 restrains antitumor immunity. Precisely how Adam2 accomplishes this will likely be investigated in a future study. However, the authors should determine whether a similar correlation exists between Adam 2 and TNF/interferon signaling and T cell activity in human cancer, e.g., LUAD. This can be done by bioinformatic gene expression analysis.

2. In the setting of Adam2 KO and overexpression, the authors should test the impact of ICI treatment. Since ICI response relies on pre-existing immunity, the expectation is that ICI response will be enhanced by Adam2 KO and reduced by Adam2 overexpression (unlike ACT).

3. The authors need to make a more direct connection between Adam 2 effect on TNF/interferon signaling, the purported main mechanism of action, and T cell activity in the context of ICI and/or ACT. They imply that adoptively transferred cytotoxic T-cells show greater cytotoxic activity in Adam2 overexpressing tumors because of restrained expression of Pd-I1, Lag3, Tigit and Tim3. The authors should test whether neutralizing TNF/interferon signaling in control tumors also enhances ACT response or whether a different Adam2 controlled mechanism is at play. In addition, I believe it will be more meaningful to evaluate Pd-1, Lag3, Tigit and Tim3 expression on T cells rather than in the overall TME.

Minor comment: interferon is referred to as IFN or INF, suggestion to use IFN only.

Reviewer #2 (Remarks to the Author): with expertise in lung cancer, cancer immunology

In this manuscript, Dervovic and colleagues explore novel modulators of tumor immunity in lung cancer. To do so, the authors perform an impressive in vivo CRISPR loss-of-function screen focused on ~600 immunomodulatory genes within genetically engineered mouse models of lung cancer. Using both KrasG12D and BrafV600E driven models, the authors discovered Serpinb9 as a common sgRNA hit depleted in both oncogenic models. Additionally, they found Adam2 to be a Kras-specific sgRNA enriched in tumors, and they discovered that Adam2 O/E decreased interferon response pathways, thereby promoting tumor growth in a treatment naïve setting; however, Adam2 O/E surprisingly promoted antigen-specific cell killing by ex vivo activated and expanded CD8 T cells. These results begin to decipher a novel mechanism of immune suppression that may inform how to treat patients with amplification/expression of this protein.

The authors completed an impressive amount of work for this manuscript; however, one major concern was the lack of focus with too many different and somewhat incomplete vignettes. While the models and the in vivo screen are highly novel and exciting, there are several issues with the

manuscript.

I have the following additional questions and concerns for the authors:

1. It is confusing that the authors have an enriched hit (Adam2) and a depleted hit (Serpinb9) that, when knocked out individually, have the same effect on tumor growth. Overall, the findings about Adam2 were presented in such a way that was convoluted and hard to follow. It took multiple reads to grasp what the authors were concluding from these results, so it would be helpful if these data and their conclusions were presented with improved clarity.
2. Kras/Braf activating mutations should be confirmed in lung tumors via allele specific PCR (or if shown previously, please cite).
3. It is unclear why the authors crossed their mice with the LSL-Confetti mouse. This model does not appear to have been utilized for tumor cell tracing, and the authors instead relied on GFP-Cas9 and BLI to demarcate tumor cells and follow tumor progression, respectively. Can the authors comment on the reason for the Confetti mouse?
4. Also, can the authors comment about the use of such young mice for the intranasal instillation of the LV-Cre? It seems that the immune system may not be fully developed at postnatal day 2. Do the authors think that the screen results may differ if adult mice are used?
5. In Supp Fig 12, the Braf lung sections do not contain GFP to the same degree as the Kras lung sections, with almost no GFP detected at all. Is the section supposed to be showing a tumor-containing lung region and if so, then why doesn't GFP indicate the tumor as it does for the Kras lung?
6. The westerns included in Supp Fig 18a and 26a are the same. Please update as these should be different experiments and thus different mouse tumor lysates.
7. In general, the figure legends were lacking in sufficient detail to know exactly what is depicted within the panel. For example, in Supp Fig 12, what do the arrows indicate? Also, TOM is denoted in yellow at the top of the first image, but the legend does not address what that is. I assume it is the tdTomato but clarification would be useful. Similarly, in Supp Figs 6a,b and Supp Figs 13a,b, are the individual bars representing tumor replicates? If so, please indicate.
8. Also in Supp Fig 6b, treated tumors do not appear to be included in the figure but the legend suggests there should be treated tumors.
9. Fig 3b seems to be skipped in the text. Also, there is not a legend for the different colored lines in this panel (and the similar graphs shown in the supplement). What are the blue/green lines versus the red/pink lines?
10. The authors used a human co-culture system to support the murine Serpinb9 findings; however, no human model was utilized for the more novel Adam2 hit. Demonstrating that Adam2 modulates human lung cancer immunity would add strength to the manuscript.
11. Can the authors demonstrate gene knockout efficiency in an additional manner? For example, it would be helpful to show protein loss with KOs of Serpin and Adam2.
12. The authors utilized both nude and NSG immunocompromised models to test the immune modulatory effects of Adam2. While the NSG results are very clear, the nude mouse results trend towards an increase in tumor growth with Adam2 O/E; in fact, they seem to be almost twice the size, so even if not statistically significant, it does appear that Adam2 may impact non-T cell immune populations. Have the authors explored the role of NK cells in their model?
13. The readout of CFSE for T-cell mediated cytotoxicity of tumor cells is not a direct measure. I would suggest a more appropriate assay, such as LDH cytotoxicity assay to measure tumor cell killing

by T cells.

14. CD44 results with IFN stimulation in ctrl or Adam2 OE lines were not included in Supp Fig 24 but was mentioned in the text. Please include or remove from text.

15. Adam2 was found to be a Kras-specific hit. From the human LUAD patient data in Supp Fig11, it would be helpful for the authors to indicate co-occurrence or mutual exclusivity, as TCGA does provide this analysis. Also, in patients with Kras + Adam2 alterations, is there a difference in overall or progression free survival?

16. The authors describe alterations to CXCL13 and CCL5 with Adam2 O/E via cytokine array (Supp Fig 16). However, it is difficult to appreciate the degree of this change. The results should be confirmed with a quantitative measure such as ELISA to confirm the array data.

17. The typos/errors within the manuscript are distracting from the data and conclusions. The authors should read the manuscript carefully for these. Examples include CTRL vs CTLR; IFN vs INF, missing data or data that isn't referenced in the text, etc.

Reviewer #3 (Remarks to the Author): with expertise in CRISPR screens, cancer immunology

The manuscript by Dervovic et al. uses CRISPR in vivo screens to identify regulators of immunotherapy in lung cancer models. They initially develop a small sgRNA library of genes, that have been reported by Rooney et al to be associated with cytolytic activity. The small sgRNA library vector also contains a CRE and an OVA encoding sequence to recombine KRAS and BRAF into their active forms and presentation of antigen to OT-1 T cells, respectively, in the lung (transgenic for the LSL_KRASG12D and LSL_BRAFV600E). Once the lungs of P2 embryos are infected, they wait for tumours to develop and then transplant OT1 T cells into these animals. These are activated by OVA/CFA and OVA/IFA injection into the mice. The authors identify some known and unknown factors that enhance/ reduce OT-1 mediated killing of the lung tumours and focus on SerpinB9 and ADAM2. While the initial idea is very nice, the validation and interpretation of the data is at this stage not sufficient for publication of this manuscript.

It starts with the analysis of the CRISPR screen and I couldn't find the overall number of guides found in the tumours before and after ACT. Could the authors please add this information?

The authors also need to explain what the different colours in Supp Figure 4 and 5 represent? I assume these are the 4 different sgRNAs for each gene in the library? The two untreated versus two treated samples look to be taken from one mouse each. Does each point reflect a technical replicate of the reads? Also, the raw reads do not really give any information. The sgRNAs, whose read counts are changed between untreated and treated samples should be presented as a proportion of total reads in each group. E.g. Serpinb9 reads were x% in the untreated and x% in the treated of the total read counts. This should be combined for all treated and untreated and for all genes which showed in the RRA score to be dropped out. How the RRA scores were determined should also be mentioned in the Materials and Methods.

When it comes to the validation of Serpinb9 in Figure 2a and b, the authors observe a clear slowing of tumour growth in Serpinb9 KO samples when looking at the luminiscore. However, the survival of the animals was not affected in the slightest in the KRAS G12D model by the absence of Serpinb9 plus/ minus ACT(Fig2c). In the BRAF V600E experiments it appears from the luminiscore again that the SerpinB9 cells have a slower tumour growth (2B). However, the survival seemed only a little affected by the treatment with ACT (Fig 2d). Interestingly while the luminiscore was strongly reduced in the control animals that was not at all reflected in the survival curve in Fig 2d when comparing control

animals treated and untreated. It appears to me that there are some differences in the beginning of the tumour onset, which is not translating into enhanced survival at later stages. When comparing the 9 weeks luminiscore in Fig2 b it appears that there are no malignant cells in the treated Serpinb9 animals, but at week 10 the first animals from that same group dies already. The authors should check whether the tumour cells at time of death of the mice are still deficient for Serpinb9 and explain how no tumour detection by luminescence can lead to death.

The Supp Figure 7 c (overall in that Figure the timepoints and ratios between b and c/d were different; why was this?) clearly shows that Serpinb9 sgRNA transduced cells are barely alive when they are used for the killing assay in vitro? 26% (sgSerpin) vs 62% (NT) based on the FSC/SSC blots. Would this explain why these cells are more sensitive to killing by the CD8 T cells? Would this also explain why the Serpinb9 sgRNAs were found as a "drop out", i.e. is generally survival/ growth of Serpinb9 deficient cells negatively impacted? The authors should repeat these killing assays in vitro with a much more viable tumour cell population lacking Serpinb9 for the killing assays.

As far as the Figure legend states the 3 individual dots in Supp Fig 7b and Fig2e represent 1 experiment with 3 technical replicates. In Figure 2f/g (according to Supp Figure 9) that is similar. While these are representative experiment, it would be important to know how often these experiments were repeated?

In the second part of the manuscript the authors focus on enriched sgRNAs from their screen and Adam2 – which is only found in the KRAS G12D ACT screen – is pursued. The interesting observation is made that Adam2 KO in KRAS G12D models slows down tumour growth and enhances survival of the mice. However, ACT treatment induces a faster tumour growth and shortens the life span of mice. This is a very surprising observation, and I would have expected that ACT has no impact on the survival of these mice in the absence of Adam2. The authors postulate that an immune suppressive microenvironment is generated in the absence of Adam2, but then - as speculated in the discussion - ACT plus checkpoint inhibitors should reverse the phenotype of accelerated death of mice. This is an important experiment, which should be performed by the authors.

Minor:

In addition Supp Figure 4 and 5 a,b,c should be annotated with the sgRNAs which they chose to pursue. Where is SerpinB9 or Stat1 for example on these blots?

In Supp Figure 6a, b the bars are not explained. What are the 4 black and 2 grey bars in a and where are the grey bars in b? Why is the last bar in b so low?

The western in Supp Figure 8 for Serpinb9 look very convincing, but the IF looks like there are not such great differences. Could the authors please discuss and include a notion why a nuclear stain for Serpinb9 was used. This information is missing in the manuscript.

Reviewer #4 (Remarks to the Author): with expertise in CRISPR screens, cancer immunology

In this study, the authors performed mini-pool CRISPR screens in mouse lung cancer models and identified a few regulator genes of the T-cell antitumor immune response. The top hit Serpinb9 corroborated well with the previous data-driven findings from Jiang et al., Nat Med 2018 and this work formally proved Serpinb9's role in an autochthonous murine model. Meanwhile, the other hit Adam2 is an intriguing finding. Although Adam2 may serve as an oncogene, Adam2 does promote the

cytotoxicity of adoptively transferred T cells. This result suggested the potential of developing anti-ADAM2 CAR T cells. In general, this paper is well written. The work is quite systematic. I only have a few suggestions for the authors to consider.

1, Have the authors compared Adam2 OE tumors upon anti-CTLA4+PDL1 combination and control treatment? The adoptive T-cell transfer depends on a positive Adam2 level. However, will Adam2 OE promote or repress the immune checkpoint blockade (ICB)? ICB may enhance the infiltration of pre-existing antitumor T cells. Such an experiment should formally justify the authors' discussion that ICB won't work in ADAM2-high tumors.

2, In the 2nd Introduction paragraph, claiming the ICB response rate is less than 13% is not correct, as some tumor types, such as Hodgkin's lymphoma, have a much higher response fraction. Please specify the tumor types and treatment settings while claiming such this low number.

3, In the 2nd Introduction paragraph, please add citations in the sentence introducing different CAR T therapies in NSCLC (e.g., MSLN, MUC1, NY-ESO-1, GPC3, PSCA, EGFR, ROR1, HER2, PDL1).

4, Page 14, paragraph 2: The author's name of this previous study is Peng Jiang, instead of Pen Jiang. Please correct the typo.

5 (Optional), The ADAM2 function on promoting ACT efficacy is still not sufficiently studied. The current data in the last result section is mostly phenotypic without disclosing further mechanisms. A transcriptomics profiling in bulk tumors was limited by the loss of cellular identity. The authors may consider single-cell RNA-seq or spatial transcriptomics Visium platforms to further explore this phenomenon. However, I understand the difficulty of figuring out underlying mechanisms. Thus, these suggestions are optional for the current revision.

Dear reviewers,

We are grateful for your critical reviews of our manuscript entitled "In vivo CRISPR screens reveal Serpinb9 and Adam2 as regulators of immune therapy response in lung cancer". We are extremely pleased to hear that the manuscript was well-received and that in vivo screens for immune regulators are of interest for cancer researchers. We are also very excited for your interest in cancer testis antigen Adam2 as a novel immune response modulator in lung cancer. Considering that the manuscript was well received, we have addressed most of the concerns raised by reviewers, which include: additional experimental work and improved readability with an elaborate schematic outlining the Adam2's proposed function of as a tumor promoter and in facilitating ACT/ICB. We also corrected all typos and added any missing information. The reviewer comments in their entirety can be found below, followed by our response in green font. For ease of reading, we also tracked all changes to the main manuscript in green font.

In summary, we feel we made 5 major improvements to the manuscript during the revisions:

1, Reviewers #1, #3 and #4 asked about the effect of ADAM2 on immune checkpoint blockade (ICB). We have now tested anti-PDL1 and anti-CTLA4 ICB with or without concomitant OT1 ACT and found that ADAM2 expression also significantly improves ICB.

2, To gain additional mechanistic insights, we performed new tumor profiling studies and found the opposite immune phenotype in Adam2 knock-out tumors compared to Adam2 overexpressing tumors.

3, We added a schematic summarizing our results from all mouse models explaining how Adam2 functions as a 'double edged' sword by (1) initially functioning as a tumor promoter blocking cytokine signaling and endogenous anti-tumor immune responses and (2) allowing enhanced response to immunotherapy (ACT and ICB) due to the less exhausted tumor microenvironment.

4, We now added additional explanations to highlight that our results are in direct support of the emerging concept of the opposing functions of interferons: On the one hand interferons function in an immune stimulatory manner and are essential to mount an endogenous adaptive response. However, prolonged interferon signaling also promote T cell exhaustion through PDL1 and other immune checkpoint mechanisms and thus can actually block cytotoxicity of T-cells. As such, blocking interferons was shown to have beneficial effects on ICB – which is exactly what we see in our model.

5. Finally, our findings have now been robustly validated in human LUAD:

- Pan-cancer analysis for *ADAM2* showed a high frequency of expression in multiple cancers.
- In lung cancer, 71/510 (13.9%) of LUAD and 26/498 (5.2%) of LUSC tumors showed *ADAM2* expression, which often correlates with *ADAM2* amplification.
- These findings have now been confirmed by RNA scope analysis in human LUAD samples.
- A new pathway analysis revealed significant downregulation of genes associated with inflammatory responses (IFN α and IFN γ ; TNF α ; IL6/JAK/STAT3 signaling) in *ADAM2* expressing human LUAD. These findings complement what we had observed in our murine studies and further our interest in cancer testis antigens re-expression in human cancers.

Please, find below a detailed point-by-point response to all comments:

Reviewer #1 (Remarks to the Author): with expertise in lung cancer, cancer immunology Dervovic et al evaluated the immune-modulatory capabilities of 573 genes (identified in a previous study) associated with altered cytotoxicity in human cancers using CRISPR/Cas9 in vivo screens in mouse lung cancer models. In addition to the known immune evasion factor Serpinb9, they identified a potential cancer testis antigen Adam2 as a new immune modulator. Adam2 was also found to be expressed, in part through gene amplification, in human cancer including LUAD. Studies in mice indicate that Adam2 functions as an oncogene as knockout of Adam2 attenuated tumor growth while overexpression accelerated tumor growth. These effects were diminished in immunodeficient mice suggesting a role of antitumor immune response. Importantly, knockout of Adam2 in tumors increased T cell infiltration and enhanced interferon I/II and TNF cytokine signaling. Paradoxically, Adam2 overexpression enhances ACT treatment response possibly by reducing expression of the immune checkpoint inhibitors (ICI) Pd-11, Lag3, Tigit and Tim3 in the tumor microenvironment. Overall, this is a **well-written and very well-performed study**. The CRISPR/Cas9 in vivo screen **has significant novelty by itself and will be of interest for cancer researchers**. The discovery of Adam2 as a novel immune response modulator is also of significant interest. However, there are also concerns that need to be addressed.

1. The mechanism of action of Adam2 proposed by the authors is attenuation of interferon I/II and TNF cytokine signaling. Indeed, this may explain how Adam2 restrains antitumor immunity. Precisely how Adam2 accomplishes this will likely be investigated in a future study. However, the authors should determine whether a similar correlation exists between Adam 2 and TNF/interferon signaling and T cell activity in human cancer, e.g., LUAD. This can be done by bioinformatic gene expression analysis.

We want to thank the reviewer for this very insightful idea! We have now teamed up with an expert in bioinformatic pathway analysis, Dr. Juri Reimand (Ontario Cancer Research Institute), and performed gene expression analysis for ADAM2 in human LUAD cancer. Pathway enrichment analysis of TCGA datasets in LUAD patients was conducted by using ActivePathways and default parameters. Separate pathway analyses were conducted for genes up-regulated and down-regulated in LUAD patients with high ADAM2 expression. Gene sets representing 50 cancer hallmark pathways of the MSigDB database were used for the enrichment analysis. Genes in each significant pathway were visualised with respect to log2-transformed fold-change and significance (FDR<0.05) in ADAM2-high vs ADAM2-low LUAD samples. Here, we report that **ADAM2 expression is associated with significantly lower IFN I/II and TNF alpha signaling pathways in human LUAD and thus perfectly mirrors the results obtained from the mouse experiments of LUAD**. We think that this is very strong data supporting our findings and have dedicated a whole new figure and several supplementary figures/tables to this data (**please see new FIGURE 6, Supplementary Figures 29-36 and Supplementary Table 4-6**). In this figure, we also present our new RNAscope data, which corroborates the TCGA RNAseq data and shows that ~50% of human LUAD show aberrant ADAM2 expression in EPCAM+ lung cancer cells (Fig. 6B). We now also show that ADAM2 expression is also found in a plethora of other cancers such as breast (9.5%), bladder (16.7%), prostate (72.4%) and renal (74.7%) cancers, indicating that ADAM2 might play a much broader role and further increasing the importance and implications of our findings (Fig. 6A).

2. In the setting of Adam2 KO and overexpression, the authors should test the impact of ICI treatment. Since ICI response relies on pre-existing immunity, the expectation is that ICI response will be enhanced by Adam2 KO and reduced by Adam2 overexpression (unlike ACT).

We want to thank this reviewer for this comment as we also speculated that ICI response would be reduced by Adam 2 overexpression. Importantly, we have now tested this experimentally and offer the following new data/conclusions:

To test how Adam2 impacts ICI treatment, we transplanted LLC cells overexpressing Adam2 and control LLC CTRL cells in B6 mice and treated the mice with PDL1 or CTLA4 blocking antibodies (Ab) once the tumor reached 100mm³. PDL1 blocking Ab as well as CTLA4 blocking Ab administered in mice bearing Adam2 tumors showed pronounced and significant tumor reduction with α PDL1 (D20, p=0.002); and α CTLA4 (D20, p=0.04) treatment respectively, while control tumors showed little or no reduction in growth at that level of α PDL1 or α CTLA4 ICI. In addition, combining PDL1 or CTLA4 inhibition with ACT led to almost complete tumor stasis of Adam2 overexpressing and control tumors, indicating strong cooperative effects (please see new Suppl. Fig. 26D and E).

This data is virtually the same as observed with OT-I ACT, where OT-I ACT had a strong effect on Adam2 overexpressing LLC tumors (please see original Figure 5B):

This reviewer and we have speculated that ICI response will be dampened by Adam2 overexpression with the argument that ICI response relies on pre-existing immunity. We concur that a complete attenuation of interferon I/II and TNF cytokine signaling, and thus complete block of anti-tumor immunity could have elicited such a response. However, it is important to point out that Adam2 O/E is **reducing** and not completely attenuating interferon I/II and TNF cytokine signaling and anti-tumor immunity. This can be nicely shown when comparing growth of Adam2 O/E LLC tumors in C57Bl/6 mice versus NSG mice as shown in the original Fig. 4A and B:

The fact that Adam2 O/E tumors are growing out much faster in NSG mice than in C57Bl/6 mice shows that there is still residual anti-tumor immunity despite Adam2 O/E, which can be harnessed by ICI.

Considering this data showing a similar response to ICI and ACT, we interpret that the dampened immune response leads to a less chronically exhausted immune milieu, which can be re-invigorated by ICI treatment or ACT. We have clarified that Adam2 is reducing (rather than blocking) the inflammatory tumor immune milieu added this new vignette to the discussion on page 15/16:

'One possible explanation is that the reduced endogenous interferon response also reduces expression of interferon-inducible check-point molecules such as PD-L1, Lag3, Tigit and Tim3 in the tumor microenvironment as seen in our mouse model. As such, Adam2 expression appears to result in a less exhausted tumor microenvironment, which is highly permissive to ex vivo expanded, adoptively transferred cytotoxic T-cells. This could explain why Adam2 overexpressing tumors are actually more sensitive to OT-I cell-mediated killing and immune-checkpoint blockade.'

3. The authors need to make a more direct connection between Adam 2 effect on TNF/interferon signaling, the purported main mechanism of action, and T cell activity in the context of ICI and/or ACT. They imply that adoptively transferred cytotoxic T-cells show greater cytotoxic activity in Adam2 overexpressing tumors because of restrained expression of Pd-1, Lag3, Tigit and Tim3. The authors should test whether neutralizing TNF/interferon signaling in control tumors also enhances ACT response or whether a different Adam2 controlled mechanism is at play. In addition, I believe it will be more meaningful to evaluate Pd-1, Lag3, Tigit and Tim3 expression on T cells rather than in the overall TME.

We agree that better mechanistic evaluation of the mode of action of Adam2 on enhanced cytolytic activity of ACT by OT-I cells in *in vivo* tumor regression is of essence. Although we do not have a conclusive mechanism, we have performed several experiments to further support our hypothesis:

To test whether limiting interferon signaling in lung tumor cells indeed enhances ACT responses, we provide the following notions:

- (1) IFN gamma activates JAK1 and JAK2 kinases leading to the phosphorylation, activation, and dimerization of STAT1 transcription factors. In direct support of our hypothesis, STAT1 as well as JAK2 scored as top ranked genes in our *in vivo* CRISPR screen in the KRas and/or BRAf model. We now substantiated these findings and genetically depleted STAT1 in *Kras*^{G12D} mouse model, which resulted in an increased overall survival in comparison to untreated control *Kras*^{G12D} mice and after ACT (p= 0.0089) (please see new Supplementary Figure 26c).

Importantly, the improved response to ICB or ACT of STAT1 and/or JAK1/2 depleted tumor cells has been observed in several other studies – for example please see reference 1,2.

- (2) We now also show that IFN gamma treatment of Adam2 O/E LLC cells, in time dependent manner, results in lower activation/phosphorylation of STAT1 compared to CTRL cells, supporting our hypothesis that Adam2 overexpression compromises IFN gamma signaling (please see new Supplementary Figure 23h).

- (3) INFs and TNF α in the tumor microenvironment is mainly produced by infiltrating cytotoxic CD8+ T cells, natural killer cells, $\gamma\delta$ T cells, Th1 polarized CD4+ T helper cells and group 1 innate lymphoid cells. As these cells are virtually all missing in immunocompromised NSG mice, we thought to assess the effect of ACT in NSG mice. ACT of OT-I cells in NSG mice bearing s.c. LLC tumors resulted in a significant reduction ($p=0.026$) of tumors compared to untreated control tumors, showing that ACT is highly effective when the endogenous adaptive and innate immune system and their associated cytokines such as IFNs and TNF α are blocked.

- (4) To further deepen the connection between Adam 2 effect on TNF/interferon signaling as suggested by this reviewer, we have now performed a new series of tumor profiling studies. Of note, in the original manuscript, we performed RNAseq experiments on LLC allograft tumors overexpressing Adam2 compared to vector only control LLC tumors. This revealed a significant reduction in cellular responses to IFN γ , IFN α/β and TNF α and reduced immune responses associated with reduced T cells exhaustion and reduced MHC-I presentation and antigen processing, which nicely explains why Adam2-overexpressing tumors grow out so aggressively in immune-competent mice, but do not exhibit a difference in growth dynamics in immune-deficient NSG mice compared to control tumors.

We now performed RNAseq experiments in autochthonous Adam2 knock-out lung tumors compared to control tumors. This revealed the exact opposite phenotype with significantly increased IFN γ and TNF α cytokine expression (as well as increased expression of other TNF and CXCL and CCL cytokines), increased MHC-I presentation and increased T-cell exhaustion marked by increased expression of Lag3, Pd-1, Tigit, and Pd-L1/2, findings that were validated by RT-PCR. As suggested by this reviewer, we are also now showing increased expression of Pd-1, Tigit, Lag3 and Infy on CD8 T-cells infiltrating Adam2 knock-out tumors using quantitative RT-PCR and FACS analysis (please see new Fig. 3g and h, new Supplementary Fig. 14a-c and 15a-c, and new Supplementary Table 4).

- (5) Lastly, we performed CyTOF analysis of Adam2 overexpressing LLC tumors, which showed lower levels of Pd-1 expression on Ag-specific CD8+ T cells, highlighting the importance of Adam2 in regulation of the intertumoral exhaustion by antigen-specific T cells (**please see new Fig. 5g**).
- (6) Lastly, it is important to note that our results are in direct support of the emerging concept of the opposing functions of interferons: On the one hand interferons function in an immune stimulatory manner and as such are essential to mount an endogenous adaptive responses. However, interferons not only augment immune functions but also promote T cell exhaustion through PDL1 and other immune checkpoint mechanisms and thus prolonged IFN signaling can actually block cytotoxicity of ex vivo or ICB-activated T-cells. As such, blocking interferons was shown to have beneficial effects on ICB – which is what we see in our model.

We have also added the following explanation and comments to the discussion to underline this notion:

‘Using loss-of-function and gain-of-function experiments, we showed that ADAM2 expression had a dramatic effect on tumor growth by restricting type I and II IFN responses as well as several other cytokine signaling pathways thereby restraining the endogenous immune surveillance machinery, affecting predominantly T-cells (please see schematic in Fig. S38). It is well-documented that IFNs are required to augment immune functions and tumors lacking IFNs expression fail to mount an anti-tumor immune response and thus grow unrestricted, which is precisely what we observed in ADAM2 overexpressing tumors. In contrast, we observed increased interferon and cytokine profiles and increased CD8 T-cells in Adam2 knock-out tumors concomitant with reduced tumor growth. To our knowledge, we describe for the first time a cancer testis antigen with not only immunogenic but also with strong immune-modulatory and tumor promoting functions.

Interestingly, we also found that ADAM2 expression not only reduces endogenous cytokine signaling and T-cell responses, but also paradoxically enhanced cytotoxicity of adoptively transferred TAA-specific cytotoxic T-cells as well as immune-checkpoint blockade. One possible explanation is that the reduced endogenous interferon responses, which at first allow the tumors to evade the immune surveillance, also leads to reduced expression of interferon-inducible check-point molecules such as PD-L1, Lag3, Tigit and Tim3 in established tumors. As such, ADAM2 expression and the associated decreased IFN signaling appears to result in a less exhausted tumor microenvironment, which is highly permissive to ex vivo expanded, adoptively transferred cytotoxic T-cells or rejuvenated endogenous cytotoxic T-cells. Conversely, Adam2 knock-out tumors exhibited increased CD8 T cell exhaustion presumably due to the prolonged IFN signaling and concomitant reduced ACT efficacy (Fig. S38). While the immune stimulatory roles of IFNs are better known, the immune suppressive immunoregulatory effects of persistent IFN signaling are being increasingly recognized not only in during chronic viral infection³⁻⁵ but also in cancer^{1,6-8}. Consistently, several functional screens have shown that blocking tumor interferon- γ (IFN γ) signaling sensitizes tumors to immune-checkpoint blockade⁹⁻¹¹. ... Further studies will be needed to elucidate the precise mechanism of how ADAM2 regulates interferon and cytokine signaling as well as activation of cytotoxic T cells.’

- 1 Benci, J. L. *et al.* Opposing Functions of Interferon Coordinate Adaptive and Innate Immune Responses to Cancer Immune Checkpoint Blockade. *Cell* **178**, 933-948 e914, doi:10.1016/j.cell.2019.07.019 (2019).
- 2 Dhainaut, M. *et al.* Spatial CRISPR genomics identifies regulators of the tumor microenvironment. *Cell* **185**, 1223-1239 e1220, doi:10.1016/j.cell.2022.02.015 (2022).
- 3 Teijaro, J. R. *et al.* Persistent LCMV infection is controlled by blockade of type I interferon signaling. *Science* **340**, 207-211, doi:10.1126/science.1235214 (2013).
- 4 Wilson, E. B. *et al.* Blockade of chronic type I interferon signaling to control persistent LCMV infection. *Science* **340**, 202-207, doi:10.1126/science.1235208 (2013).
- 5 Ng, C. T. *et al.* Blockade of interferon Beta, but not interferon alpha, signaling controls persistent viral infection. *Cell Host Microbe* **17**, 653-661, doi:10.1016/j.chom.2015.04.005 (2015).

- 6 Tumei, P. C. *et al.* PD-1 blockade induces responses by inhibiting adaptive immune resistance. *Nature* **515**, 568-571, doi:10.1038/nature13954 (2014).
- 7 Minn, A. J. & Wherry, E. J. Combination Cancer Therapies with Immune Checkpoint Blockade: Convergence on Interferon Signaling. *Cell* **165**, 272-275, doi:10.1016/j.cell.2016.03.031 (2016).
- 8 Benci, J. L. *et al.* Tumor Interferon Signaling Regulates a Multigenic Resistance Program to Immune Checkpoint Blockade. *Cell* **167**, 1540-1554 e1512, doi:10.1016/j.cell.2016.11.022 (2016).
- 9 Manguso, R. T. *et al.* In vivo CRISPR screening identifies Ptpn2 as a cancer immunotherapy target. *Nature* **547**, 413-418, doi:10.1038/nature23270 (2017).
- 10 Lawson, K. A. *et al.* Functional genomic landscape of cancer-intrinsic evasion of killing by T cells. *Nature* **586**, 120-126, doi:10.1038/s41586-020-2746-2 (2020).
- 11 Dubrot, J. *et al.* In vivo CRISPR screens reveal the landscape of immune evasion pathways across cancer. *Nat Immunol* **23**, 1495-1506, doi:10.1038/s41590-022-01315-x (2022).

Taken together, these data (although not a complete proof) supports the notion that Adam2 indeed functions by limiting type I and II IFN- and TNF α -induced signalling in lung tumor cells leading to less MHC-I antigen cross-presentation and reduced CD8-mediated anti-tumor cytotoxicity and a less exhausted tumor microenvironment. And that this less exhausted tumor microenvironment eventually allows for enhanced function of adoptively transferred TAA-specific T-cells. Without any doubt do we need further studies to elucidate the exact molecular underpinnings how Adam2 interferes with IFN/TNF-signaling and regulates expression of all these immune checkpoint receptors, but we hope that this reviewer agrees that this would be beyond the scope of the current manuscript.

Minor comment: interferon is referred to as IFN or INF, suggestion to use IFN only.

We want to apologize for this oversight, which has been corrected in the revised manuscript.

Reviewer #2 (Remarks to the Author): with expertise in lung cancer, cancer immunology

In this manuscript, Dervovic and colleagues explore novel modulators of tumor immunity in lung cancer. To do so, the authors perform an impressive in vivo CRISPR loss-of-function screen focused on ~600 immunomodulatory genes within genetically engineered mouse models of lung cancer. Using both KrasG12D and BrafV600E driven models, the authors discovered Serpinb9 as a common sgRNA hit depleted in both oncogenic models. Additionally, they found Adam2 to be a Kras-specific sgRNA enriched in tumors, and they discovered that Adam2 O/E decreased interferon response pathways, thereby promoting tumor growth in a treatment naïve setting; however, Adam2 O/E surprisingly promoted antigen-specific cell killing by ex vivo activated and expanded CD8 T cells. These results begin to decipher a novel mechanism of immune suppression that may inform how to treat patients with amplification/expression of this protein.

The authors completed an **impressive amount of work** for this manuscript; however, one major concern was the lack of focus with too many different and somewhat incomplete vignettes. While the models and the in vivo screen are **highly novel and exciting**, there are several issues with the manuscript. I have the following additional questions and concerns for the authors:

1. It is confusing that the authors have an enriched hit (Adam2) and a depleted hit (Serpinb9) that, when knocked out individually, have the same effect on tumor growth. EXPLAIN Overall, the findings about Adam2 were presented in such a way that was convoluted and hard to follow. It took multiple reads to grasp what the authors were concluding from these results, so it would be helpful if these data and their conclusions were presented with improved clarity.

We want to thank the reviewer for this comment. The screen was designed to elucidate how cells with a given gene knock-out behave during ACT treatment but not how the gene affects growth of lung tumor cells under homeostasis. As such, the screening results showed that SerpinB9 enhances ACT treatment, while Adam2 loss causes resistance to ACT treatment. Interestingly, in the individual follow-up studies, we found that loss of SerpinB9 by itself already increases the endogenous anti-tumor immune response and as such results in a slower tumor growth, which is further exacerbated by ACT. To our surprise and in agreement with this reviewer, we found that loss of Adam2 by itself also resulted in decreased tumor growth, despite that fact that it caused resistance to ACT. As such, Adam2 functions as a double-edged sword, which makes the interpretation of the data at first challenging.

To help further elucidate these phenomena, we have now performed a new series of tumor profiling studies. Of note, in the original manuscript, we performed RNAseq experiments on LLC allograft tumors overexpressing Adam2 compared to vector only control LLC tumors. This revealed a significant reduction in cellular responses to IFN γ , IFN α/β and TNF α and reduced immune responses associated with reduced T cells exhaustion and reduced MHC-I presentation and antigen processing, which nicely explains why Adam2-overexpressing tumors grow out so aggressively in immune-competent mice, but do not exhibit a difference in growth dynamics in immune-deficient NSG mice compared to control tumors.

We now performed RNAseq experiments in autochthonous Adam2 knock-out lung tumors compared to control tumors. This revealed the exact opposite phenotype with significantly increased IFN γ and TNF α cytokine expression (as well as increased expression of other TNF and CXCL and CCL cytokines), increased MHC-I presentation and increased T-cell exhaustion marked by increased expression of Lag3, Pd-1, Tigit, and Pd-L1/2, findings that were validated by quantitative RT-PCR and

FACS (please see new Fig. 3g and h, new Supplementary Fig. 14a-c and 15a-c, and new Supplementary Table 4)).

Together, this nicely shows the emerging concept of the opposing functions of interferons, which on the one hand function in an immune stimulatory manner and as such are essential to mount an endogenous adaptive responses. However, interferons not only augment immune functions but also promote T cell exhaustion through PDL1 and other immune checkpoint mechanisms and thus prolonged IFN signaling can actually block cytotoxicity of *ex vivo* or ICB-activated T-cells – which is what we see in our model.

We strongly believe that these new data help explain the phenotype much better and we have now added conclusions throughout the text and an explanation with a new schematic to the discussion (please see Supplementary Fig. 38) to help the readers conceptualize the data:

Fig. S38 Proposed functions of ADAM2

A, Adam2 is absent in normal lung. B, Adam2 is aberrantly expressed in KRas-mutant lung tumors and functions as an oncogene by reducing IFN/TNF-signaling, reducing MHC-I presentation. As such, Adam2 functions as an immune-suppressant in this setting. Forced expression of ADAM2 in LLC cells corroborated these findings and also triggered a less exhausted TME marked by reduced IFN/TNF-signaling, reduced MHC-presentation and rapid outgrowth of tumors (in an immune-system dependent manner). C, ACT-treated KRas-mutant lung tumors exhibit further elevated Adam2 expression levels and a good response to adoptive transfer of OT1 T-cells and immune checkpoint blockade (ICB). Forced expression of Adam2 in LLC cells treated with ACT or ICB showed that Adam2 overexpression indeed sensitizes to cancer immunotherapy marked by significantly elevated expression levels of granzymes and perforin and reduced expression of PD-1 on infiltrating Ag-specific cytotoxic T-cells. D, Genetically ablating Adam2 in KRas-mutant lung tumors leads to increased IFN/TNF signaling and MHC presentation, increased CD8 cells and decreased protumoral M2 macrophages and overall better immune control and less tumor burden. E, Loss of Adam2 blocks ACT and ICB: The prolonged IFN/TNF-signaling observed in KRas-mutant Adam2 KO lung tumors promotes T-cell exhaustion and blocks cytotoxicity of *ex vivo* or ICB-activated T-cells marked by increased expression of PD-1, PD-L1, TIGIT and LAG3 as well as INF and TNFs.

Editorial Note: Lung picture was generated by courtesy of Ella Fific.

We have now improved readability and clarity of our conclusions with respect to these findings throughout the text and hope the revised MS is now much easier to read and understand. We have also added the following explanation and comments to the discussion:

'Using loss-of-function and gain-of-function experiments, we showed that ADAM2 expression had a dramatic effect on tumor growth by restricting type I and II IFN responses as well as several other cytokine signaling

pathways thereby restraining the endogenous immune surveillance machinery, affecting predominantly T-cells (please see schematic in Fig. S38). It is well-documented that IFNs are required to augment immune functions and tumors lacking IFNs expression fail to mount an anti-tumor immune response and thus grow unrestricted, which is precisely what we observed in ADAM2 overexpressing tumors. In contrast, we observed increased interferon and cytokine profiles and increased CD8 T-cells in Adam2 knock-out tumors concomitant with reduced tumor growth. To our knowledge, we describe for the first time a cancer testis antigen with not only immunogenic but also with strong immune-modulatory and tumor promoting functions.

Interestingly, we also found that ADAM2 expression not only reduces endogenous cytokine signaling and T-cell responses, but also paradoxically enhanced cytotoxicity of adoptively transferred TAA-specific cytotoxic T-cells as well as immune-checkpoint blockade. One possible explanation is that the reduced endogenous interferon responses, which at first allow the tumors to evade the immune surveillance, also leads to reduced expression of interferon-inducible check-point molecules such as PD-L1, Lag3, Tigit and Tim3 in established tumors. As such, ADAM2 expression and the associated decreased IFN signaling appears to result in a less exhausted tumor microenvironment, which is highly permissive to ex vivo expanded, adoptively transferred cytotoxic T-cells or rejuvenated endogenous cytotoxic T-cells. Conversely, Adam2 knock-out tumors exhibited increased CD8 T cell exhaustion presumably due to the prolonged IFN signaling and concomitant reduced ACT efficacy (Fig. S38). While the immune stimulatory roles of IFNs are better known, the immune suppressive immunoregulatory effects of persistent IFN signaling are being increasingly recognized not only in during chronic viral infection³⁻⁵ but also in cancer^{1,6-8}. Consistently, several functional screens have shown that blocking tumor interferon- γ (IFN γ) signaling sensitizes tumors to immune-checkpoint blockade⁹⁻¹¹. Further studies will be needed to elucidate the precise mechanism of how ADAM2 regulates interferon and cytokine signaling as well as activation of cytotoxic T cells.'

2. Kras/Braf activating mutations should be confirmed in lung tumors via allele specific PCR (or if shown previously, please cite).

While we have not performed allele-specific PCR in lung tumors for Kras/Braf activating mutations, all mice were genotyped by regular PCR protocol. Please note that Kras^{LSL-G12D} and Braf^{LSL-V600E} strain carries a Lox-Stop-Lox (LSL) sequence followed by the Kras^{G12D} or the Braf^{V600E} point mutation allele commonly associated with human cancer^{12,13}. These mice were obtained from The Jackson Laboratory. To allow for the expression of the mutant Kras or Braf oncogenic protein, we have performed intranasal instillation of CRE-carrying lentiviral virus directly into the lung to activate these mutations. Please note that we have performed standard PCR continuously throughout the study (from founders to progeny).

To confirm expression of the mutant Kras^{G12D}, we mined our RNAseq data from more than 12 lung tumor obtained from Cre-transduced LSL-KRas^{G12D}-mutant mice, which clearly shows expression of the KRas^{G12D} mRNA clearly showing the GGT->GAT (reverse complement of ACC->ATC):

The original reference describing the generation of the LSL-KRas^{G12D} mice used mutant specific antibodies and Western blot analysis to confirm KRas^{G12D} in lung tumors after viral Cre inhalation: **Analysis of lung tumor initiation and progression using conditional expression of oncogenic K-ras.** Jackson EL , et al. Genes Dev 2001 15(24):3243-8

The Cre-inducible KRas^{G12D} transgene of this mouse was also sequence verified in the following publication: **Evolutionary routes and KRAS dosage define pancreatic cancer phenotypes.** Mueller S. et al. Nature 2018 Feb 1;554(7690):62-68

So far, 1285 publications have successfully used this LSL-KRas^{G12D} mouse – for details please visit <http://www.informatics.jax.org/reference/allele/MGI:2429948?typeFilter=Literature>

Cre-dependent *BRaf*^{V600E} genomic rearrangement and *BRaf*^{V600E} mRNA expression entirely restricted to the lung was shown in **A new mouse model to explore the initiation, progression, and therapy of BRAFV600E-induced lung tumors.** Dankort et al. Genes Dev 2007 Feb 15;21(4):379-84

So far, 160 publications have successfully used this LSL-Braf^{V600E} mouse – for details please visit <http://www.informatics.jax.org/reference/allele/MGI:3711771?typeFilter=Literature>

We apologize for the oversight of not having added the reference to the original papers for these mice, which we have now included into the Material and Methods section of the revised manuscript.

3. It is unclear why the authors crossed their mice with the LSL-Confetti mouse. This model does not appear to have been utilized for tumor cell tracing, and the authors instead relied on GFP-Cas9 and BLI to demarcate tumor cells and follow tumor progression, respectively. Can the authors comment on the reason for the Confetti mouse?

We apologize for not explaining the use of The R26R-Confetti mice in our study. R26-LSL-Confetti mice function as a stochastic multicolor Cre recombinase reporter strain. Upon Cre-mediated recombination, one out of four fluorescent proteins are expressed from a single genomic locus, which enables labelling and lineage-tracing individual recombined cells, and as such, we used these mice to determine the viral concentration required to transduce lung epithelium at clonal density *in vivo*. We used intranasal instillation at P2 (the saccular stage of lung development) to administer LV-sgRNA-Cre-OVA lentivirus to the lungs of (LSL)-*Kras*^{G12D} or LSL-*Braf*^{V600E} mice crossed to multicolor LSL-Confetti mice. Cre-mediated excision of the LSL-cassettes induced activation of *Kras*^{G12D} or *Braf*^{V600E} as well as expression of one of the four fluorescent proteins encoded in the Confetti reporter cassette. Viral titration was optimized first *in vitro* followed by optimization *in vivo* using Confetti mouse to allow for transduction at clonal density at a defined concentration of lentivirus of 10⁶ pfu .

We have now added this description to the Method section and added in the main text:

'To determine the viral concentration needed to transduce the lung epithelium at clonal density, we administer LV-sgRNA-Cre-OVA at postnatal day 2 (P2) to the lungs of Lox-Stop-Lox (LSL)-Kras^{G12D} or LSL-Braf^{V600E} mice crossed to multicolor LSL-Confetti mice using intranasal instillation.'

4. Also, can the authors comment about the use of such young mice for the intranasal instillation of the LV-Cre? It seems that the immune system may not be fully developed at postnatal day 2. Do the authors think that the screen results may differ if adult mice are used?

The intranasal instillation of lentivirus into 2-day old mouse pups has purely technical reasons: The classic Taylor Jacks model for transducing the lung epithelium of LSL-KRas^{G12D} mice indeed uses intranasal instillation of adenovirus into mice older than 6 weeks. Although, adenovirus allow the generation of extremely high titer virus suitable for in vivo experiments, adenovirus does not stably integrate into the host genome. This transient nature impedes multiplexed screening, as viral barcodes are lost over time, prohibiting the identification of sgRNAs by next-generation sequencing. Taylor Jacks' lab has also developed an alternative protocol using lentiviral vectors. However, lentiviral vectors do not allow generation of very high titer and thus necessitate intratracheal intubation for direct delivery into the lung. However, even intratracheal intubation of lentiviruses is not very effective and will only result in a handful of tumors and precludes any effort to screen multiple genes. If one wants to screen 500 genes at a coverage of 100, one needs 50.000 independent tumor nodules, and as such intratracheal intubation of lentiviruses into adult mice is simply inadequate.

For detailed explanations of adeno- versus lentivirus and intranasal versus intratracheal instillation, please see: Conditional mouse lung cancer models using adenoviral or lentiviral delivery of Cre recombinase. **DuPage M. et al., Nat Protoc. 2009; 4(7): 1064–1072.**

To generate sufficient number tumor nodules to enable multiplexed screening, we tested embryonic transductions of the lung epithelium using ultrasounds guided *in utero* injections to deliver lentiviral libraries to the amniotic fluid at E16.5, which is being inhaled by mouse embryos by embryonic breathing movements in order to fill and open their lung aveoli. This results in wide-spread transduction of the embryonic lung epithelium. Importantly, much of the epithelium at this stage consists of stem cells, which incorporate the lentivirus and propagate its cargo (sgRNA, Cre) into adulthood. Similarly, we found that intranasal transduction of neonatal mice also results in wide-spread and long-lasting transduction of the lung epithelium and the generation of hundreds to thousands of independent clones. Importantly, we wanted to target alveolar stem cells (ATII), which start to develop at e17.5-P4 (saccular stage of lung development), allowing for propagation of our cargo into adulthood. Of note, the lung tumors using MOI predetermined to induce single clonal infection do not develop before week 3-4, the time-point when ACT is performed.

In summary, the early time point of lung transduction enabled us to generate sufficient tumor clones to perform multiplexed screening.

We do not think that the early transduction affects our results as (1) the lung is functionally fully developed at birth (and just keeps growing and maturing thereafter) and (2) lung tumors anyway do not develop until week 3-4. However, we obviously cannot rule out that some hits maybe affected by developmental postnatal growth of the lung, but for the reasons outlined above, such a highly multiplexed screen can simply not be done in adult mice and as such constitutes the limitation of the experimental system. With regards to the two hits we followed up mechanistically, it is important to note that (3) we confirmed the effect of SerpinB9 and Adam2 by allografting lung tumor cells into immune-competent adult mice and as such can be certain that they are not affected by the neonatal experimental design.

We have added this explanation to the Methods section and are in the mist of submitting a methods paper describing the detailed procedure and the advantages compared to lentiviral transduction of adult mice.

5. In Supp Fig 12, the Braf lung sections do not contain GFP to the same degree as the Kras lung sections, with almost no GFP detected at all. Is the section supposed to be showing a tumor-containing lung region and if so, then why doesn't GFP indicate the tumor as it does for the Kras lung?

Thank you for noticing this and we apologize for this oversight. Multiple lungs and sections were imaged, and lungs from LSL-*Braf*^{V600E} that contain GFP (i.e. activation of the oncogene), are now included (please see new Suppl Fig 10).

6. The westerns included in Supp Fig 18a and 26a are the same. CHANGE ADD THE RIGHT ONES Please update as these should be different experiments and thus different mouse tumor lysates.

Our apologies again for this mistake. The Westerns for two different treatment groups – untreated vs treated mice are indeed different and this has been corrected in the revised version (please see new Suppl Fig 17a).

7. In general, the figure legends were lacking in sufficient detail to know exactly what is depicted within the panel. For example, in Supp Fig 12, what do the arrows indicate? EXPLAIN MORE DETAILED Also, TOM is denoted in yellow at the top of the first image, but the legend does not address what that is. I assume it is the tdTomato but clarification would be useful. Similarly, in Supp Figs 6a,b and Supp Figs 13a,b, are the individual bars representing tumor replicates? If so, please indicate.

We apologize and thank this reviewer to point out these mistakes, which we now corrected in the revised manuscript. Arrows in Supp. Fig. 12 (new Suppl Fig. 10) are showing Adam2 transcripts detected by RNAscope probe Mm-Adam2-C3 using Atto647 (far red fluorescence) – depicted with color red.

TOM in yellow represents RNAscope probe tdtTomato-C2, which was detected by Atto550 and represents detection of OT1-T cell within lungs (please see new Suppl Fig 10).

Supp. Fig. 6 a, b represent different individual lungs isolated from *Kras*^{G12D} or *Braf*^{V600E} mouse models transduced with SerpinB9 sgRNA. These are 5 and 4 independent lungs tested for knock-out efficacy of SerpinB9. This has now been clarified in the legend.

Supp. Fig. 13 a, b (now Suppl. Fig. 11 a, b) represent different lungs isolated from *Kras*^{G12D} mouse model transduced with Adam2 sg1 or Adam2 sg2. These are 4 and 5 independent lungs tested for knock-out efficiency of Adam2. This has now been clarified in the legend.

8. Also in Supp Fig 6b, treated tumors do not appear to be included in the figure but the legend suggests there should be treated tumors.

We apologize for this oversight. Treated lungs are now included in the figure.

9. Fig 3b seems to be skipped in the text. Also, there is not a legend for the different colored lines in this panel (and the similar graphs shown in the supplement). What are the blue/green lines versus the red/pink lines?

We want to apologize for this oversight. Fig 3b represents 4 different sgRNAs targeting Adam2 and the graphs show the relative representation of these sgRNAs in replicates of 5+5 untreated versus 5+5 ACT treated KRas^{G12D}-lung tumors.

ADAM2_CTTAGCCAGTCCCAAACAGA
ADAM2_GTATACAGTTATGACAACGC
ADAM2_TGAACACGTGATCTACCAAG
ADAM2_TGTGACAAGAAGTATGCAGG

The Fig. 3b figure legend has now been updated and a more detailed description is now added to the methods section.

10. The authors used a human co-culture system to support the murine Serpinb9 findings; however, no human model was utilized for the more novel Adam2 hit. Demonstrating that Adam2 modulates human lung cancer immunity would add strength to the manuscript.

We agree that assessing Adam2 in a human co-culture system would be very nice, but obtaining matched cytotoxic T cells that would kill human cancer cells that overexpressing ADAM2 or where ADAM2 was knocked out proved technically very challenging.

So, while we have not yet been able to use a human co-culture system to support our murine Adam2 findings, we have now performed extensive bioinformatic gene expression analysis of ADAM2-expressing versus ADAM2-non expressing human TCGA LUAD tumors. Pathway enrichment analysis was conducted by using ActivePathways and gene sets representing 50 cancer hallmark pathways of the MSigDB database. Importantly, this analysis showed that expression of ADAM2 is associated with significantly downregulated IFN α and IFN γ as well as TNF alpha signaling pathways in human LUAD and thus mirrors precisely the results obtained from the mouse experiments of LUAD. **(please see new Fig. 6D and Suppl. Fig. 30-37)**

11. Can the authors demonstrate gene knockout efficiency in an additional manner? For example, it would be helpful to show protein loss with KOs of Serpin and Adam2.

Despite enormous efforts to find antibodies that would recognize mouse Serpinb9 and Adam2 proteins by Western blot, IHC or IF, we have so far unfortunately failed. As such, we hope that the manifold knock-out experiments in mice, mouse cells and human cells using at least 2 independent sgRNAs with confirmed targeting efficacy as well as the gain-of-function experiments using overexpression constructs that were validated using Western Blot for Serpinb9 and Adam2 and IF for Serpinb9 suffice. We will try to generate new Adam2 antibodies for our follow-up studies, but this unfortunately is beyond the time scope to this current revision.

12. The authors utilized both nude and NSG immunocompromised models to test the immune modulatory effects of Adam2. While the NSG results are very clear, the nude mouse results trend towards an increase in tumor growth with Adam2 O/E; in fact, they seem to be almost twice the size, so even if not statistically significant, it does appear that Adam2 may impact non-T cell immune populations. Have the authors explored the role of NK cells in their model?

We very much agree with this reviewer's assessment. NSG mice not only lack T and B cells, but also have defective signaling pathways for IL2, IL4, IL7, IL9, IL15, and IL21 resulting in the absence and/or

functional impairment of not only the adoptive but also the innate immune cells, including NK cells. Nude mice on the other hand only lack T-cells while NK cells, macrophages etc. are largely normal. The fact that loss of T-cells significantly reduces the oncogenic potential of Adam2 certainly indicates that Adam2 functions to a large part by inhibiting T-cells. However, given that this is not a complete but rather a partial effect and that NSG mice completely block Adam2's oncogenic function, indicates that other mechanism and cell types are involved such as NK cells or B-cells.

We have now analyzed our RNAseq data in more detail and indeed, we see upregulation of several NK makers such as Klrg1 (3.3-fold), Nkp46 (NC1, 1.9-fold), Klrb1a (1.9-fold), Klrd1 (1.7-fold), Sh2d1b2 (7.1-fold), KLRB1A (1.9-fold), KLRB1B (1.9-fold), Klrd1 (1.7-fold) but no effect on B-cell markers. This is indeed interesting and suggest that NK cells may have a role in addition to T-cells. However, we hope that this reviewer agrees that a detailed exploration of this phenomenon is beyond the scope of the current paper, but will be focus of our future explorations.

In addition, we have performed Cytof analysis and found a significant increase ($p=0.02$) of CD11c⁺ MHC2⁺ DC-like macrophages among CD45⁺ TILs in ADAM2 O/E tumors. These cells can act as antigen presenting cells and are able to cross-present tumor antigens to TILs (**please see new Fig. 5F**). In addition, expression of PD1 by CD8⁺ TCRb⁺ T cells is significantly downregulated in the Adam2 O/E tumors, which may be related to enhanced MHC cross-presentation by CD11c⁺ MHC2⁺ cells.

Together, these data indicate that other cells such as CD11c⁺ MHC2⁺ DC-like macrophages (and potentially also NK cells) play a role in addition to T-cells and this data has now been added to the revised manuscript.

13. The readout of CFSE for T-cell mediated cytotoxicity of tumor cells is not a direct measure. I would suggest a more appropriate assay, such as LDH cytotoxicity assay to measure tumor cell killing by T cells.

While CFSE is not direct measure of T cell mediated toxicity and chromium release assay has been used as 'gold standard' for measurement of cytotoxicity by T and NK cells, CFSE-based or flow-cytometry based assays have been broadly accepted as proxy for the use of chromium release assay¹⁴⁻¹⁷. Disadvantages of chromium release assay are multiple: use of radioactive compounds, poor loading and high spontaneous release. The major advantage for use of flow-cytometry based killing assay lies in the ability to discriminate effector cells from target cells. The lack of discrimination between effectors and targets cannot be observed with the use of either, MTT or LDH assay. MTT and LDH are commonly used with drug-related cytotoxicity but even with drug-related cytotoxicity, use of incucyte is readily employed. Within the narrow time frame of the revisions and the scarcity of patient-derived $\gamma\delta$ T-cells we have unfortunately not been able to perform the LDH assay. Given that the CFSE assays are generally accepted in the field and consistently gave highly significant results in a dose-dependend manner, we really hope that this reviewer agrees that this will be sufficient.

14. CD44 results with IFN stimulation in ctrl or Adam2 OE lines were not included in Supp Fig 24 but was mentioned in the text. Please include or remove from text.

Thank you for noticing. We have now removed CD44.

15. Adam2 was found to be a Kras-specific hit. From the human LUAD patient data in Supp Fig11, it would be helpful for the authors to indicate co-occurrence or mutual exclusivity, as TCGA does provide this analysis. Also, in patients with Kras + Adam2 alterations, is there a difference in overall or progression free survival?

Again, we want to thank this reviewer for this insightful question. Indeed, we now analyzed the TCGA LUAD data and found that ADAM2 expression was observed commonly in LUAD with oncogenic KRAS mutations (12 out of 74 samples, =16.2%), while only one out of 19 LUAD with oncogenic BRAF mutations exhibited ADAM2 expression (please see new Supplementary Fig. 29e).

Now, while this is in line with our findings in our mouse models, I just want to note that co-occurrence (or mutual exclusivity) is neither a necessary nor a sufficient condition for genes involved in the same pathway. In line with this, several other KRas/Braf-wildtype LUAD cancers also showed prominent ADAM2 expression (many of which harbor EGFR amplifications or harbor EGFR or NF1 mutations). In addition, in the 586 TCGA LUAD samples, 'only' 230 samples have mutational and transcriptomic data and of those 'only' 33% harbor oncogenic KRas mutations (74 samples), 'only' 7% harbor oncogenic BRaf mutations (19 samples) and 'only' respectively 13.9% exhibit ADAM2 expression (29 samples). So, while the TCGA data show the right trend and follow the mouse data, unfortunately, this limited search space is underpowered to make strong statistical arguments:

A	B	Neither	A Not B	B Not A	Both	Log2 Odds Ratio	p-Value	q-Value	Tendency
KRAS: MUT	BRAF: MUT	138	75	17	0	<-3	<0.001	0.003	Mutual exclusivity
ADAM2: EXP	KRAS: MUT	133	22	68	7	-0.684	0.205	0.308	Mutual exclusivity
ADAM2: EXP	BRAF: MUT	185	28	16	1	-1.276	0.339	0.339	Mutual exclusivity

Similarly to the co-occurrence/exclusivity data, the available TCGA LUAD data is a little too sparse to make strong statistical survival analysis, but again, the KRAS-mutant and ADAM2-expressing LUAD cases show a trend towards reduced disease free survival (median 12.98 month) compared to the unaltered control group (median 35.58 month) and if we compare those two groups in isolation, the Gehan-Breslow-Wilcoxon test gives a significant p-

value of 0.0436. However, when considering all groups, a LogRank and the Gehan-Breslow-Wilcoxon test shows non-significance:

It is also important to note that interpretation of all these results has to be done with caution, as they can be confounded by many different variables that are not controlled for in these analyses. In addition, it is important to note that there are just 7 cases with KRAS mutation and concomitant ADAM2 expression. As such, these data are again underpowered to do rigorous statistical testing and thus we prefer not to include these data into the manuscript, as it could be over interpreted by some readers. It is also important to note that while survival differences are certainly nice to see, it would not substantially change the interpretation of our results as non-ADAM2-expressing LUAD certainly have found other ways to suppress the immune system using e.g. well-known immune-checkpoints such as PDL-1 expression or downregulation of MHC-I antigen presentation etc. . As more data come available esp. from clinical trials comparing ICI and ACT responders versus non-responders, we will certainly revisit this analysis and include into future publications.

16. The authors describe alterations to CXCL13 and CCL5 with Adam2 O/E via cytokine array (Supp Fig 16). However, it is difficult to appreciate the degree of this change. The results should be confirmed with a quantitative measure such as ELISA to confirm the array data.

We agree with this reviewer about the need to profile the changes in cytokine expression in a more unbiased and comprehensive manner. We have thus now performed additional transcriptional profiling of four ACT-treated control and four ACT-treated Adam2 knock-out lung tumors (**please see new Suppl. Table S4**). GSEA revealed significantly upregulated ‘cytokine activity’ in Adam2 knock-out lung tumors (**please see new Suppl. Fig. S14c**). Interestingly, several immune-modulatory cytokines such as Ifny, several Tnf superfamily ligands (Tnf α , Tnfsf4, Tnfsf9, Tnfsf11, Tnfaip2), and interferon-stimulated chemokines such as Cxcl10, Ccl3 and Ccl5 were amongst the top-upregulated genes and used quantitative RT-PCR to confirm these findings (**please see new Fig. S15a**).

17. The typos/errors within the manuscript are distracting from the data and conclusions. The authors should read the manuscript carefully for these. Examples include CTRL vs CTLR; IFN vs INF, missing data or data that isn’t referenced in the text, etc.

We thank you for bringing this to our attention and the typos/errors as well as missing data or data that are not referenced are now fixed.

Reviewer #3 (Remarks to the Author): with expertise in CRISPR screens, cancer immunology

The manuscript by Dervovic et al. uses CRISPR in vivo screens to identify regulators of immunotherapy in lung cancer models. They initially develop a small sgRNA library of genes, that have been reported by Rooney et al to be associated with cytolytic activity. The small sgRNA library vector also contains a CRE and an OVA encoding sequence to recombine KRAS and BRAF into their active forms and presentation of antigen to OT-1 T cells, respectively, in the lung (transgenic for the LSL_KRASG12D and LSL_BRAFV600E). Once the lungs of P2 embryos are infected, they wait for tumours to develop and then transplant OT1 T cells into these animals. These are activated by OVA/CFA and OVA/IFA injection into the mice. The authors identify some known and unknown factors that enhance/ reduce OT-1 mediated killing of the lung tumours and focus on SerpinB9 and ADAM2. While the **initial idea is very nice**, the validation and interpretation of the data is at this stage not sufficient for publication of this manuscript.

1. It starts with the analysis of the CRISPR screen and I couldn't find the overall number of guides found in the tumours before and after ACT. Could the authors please add this information?

We have missed to add table with sgRNAs and this information is now added as Suppl. table 2 and 3.

2. The authors also need to explain what the different colours in Supp Figure 4 and 5 represent? I assume these are the 4 different sgRNAs for each gene in the library? The two untreated versus two treated samples look to be taken from one mouse each. Does each point reflect a technical replicate of the reads? Also, the raw reads do not really give any information. The sgRNAs, whose read counts are changed between untreated and treated samples should be presented as a proportion of total reads in each group. E.g. Serpinb9 reads were x% in the untreated and x% in the treated of the total read counts. This should be combined for all treated and untreated and for all genes which showed in the RRA score to be dropped out. How the RRA scores were determined should also be mentioned in the Materials and Methods.

We are truly sorry for not specifying the technical aspects of the screen in more detail, which we have now added to the revised manuscript. For ease of reading we have broken down this comment into the following sub-sections:

- 2.1 The authors also need to explain what the different colours in Supp Figure 4 and 5 represent? I assume these are the 4 different sgRNAs for each gene in the library?

Yes, supplementary figure and 5 describes 4 different sgRNAs targeting the top depleted or top enriched genes obtained from the screens performed on untreated and ACT-treated lungs isolated from *Kras*^{G12D}Cas9 and *Braf*^{V600E} Cas9 mice. The screening library was designed to target 4sgRNA/gene. The total number of genes targeted is 573, which equals to 2292 sgRNAs. For example, for Adam2 for different colors represent following 4 different sgRNAs:

ADAM2_CTTAGCCAGTCCCAAACAGA
ADAM2_GTATACAGTTATGACAACGC
ADAM2_TGAACACGTGATCTACCAAG
ADAM2_TGTGACAAGAACTATGCAGG

The exact same pattern has been used for the rest of the genes, which were targeted with 4 different sgRNAs selected from the published TKO library.¹⁸ The graphs represent 4 different sgRNAs targeting Adam2 and the graphs show the relative representation of these sgRNAs in replicates of untreated versus ACT treated KRasG12D-lung tumors. The figure legends have now been updated and a more detailed description is now added to the methods section.

2.2 The two untreated versus two treated samples look to be taken from one mouse each. Does each point reflect a technical replicate of the reads?

No, each datapoint represents biological replicates, which were made up by 5 lungs each:

1. Untreated replicate 1: 5 untreated lungs
2. Untreated replicate 2: 5 untreated lungs
3. ACT treated replicate 1: 5 treated lungs
4. ACT treated replicate 2: 5 treated lungs

Experimentally, we followed the procedure: Genomic DNA from every single transduced lung was prepared separately and equal amounts of every single lung was pre-amplified with nested primers (=PCR1) separately. The PCR product from 5 lungs was then pooled in equal ratios to perform the second PCR2 reaction with NGS specific primers.

Each point thus represents a total of 5 pooled lungs (untreated and ACT treated) in biological replicates (i.e. independent transductions).

2.3 Also, the raw reads do not really give any information. The sgRNAs, whose read counts are changed between untreated and treated samples should be presented as a proportion of total reads in each group. E.g. Serpinb9 reads were x% in the untreated and x% in the treated of the total read counts.

The graphs do not show raw reads but normalized reads. These graphs are a standard output of the MAGeCK CRISPR algorithm to visualize distribution of individual sgRNAs between replicates and biological conditions. To clarify, we added ‘normalized’ read counts to the y-axis label and further clarified this in the figure legend and MM.

The sgRNAs, whose read counts are changed between untreated and treated samples are normalized and as such already represented as a proportion of total reads. For example: Normalized Serpinb9 reads in the untreated and treated samples to total number of reads are shown below.

	untreated (group 1)	untreated (group 2)	treated (group 3)	treated (group 4)
Serpinb9_exon_03_CTTGGGTGCAAAGGGACAGA	2214.5	2186.7	954.91	1009.6
Serpinb9_exon_05_CACCTCTGCTTCTTCAGCAA	9045.8	8628.1	5123.2	5344.4
Serpinb9_exon_06_CGTCGATTCAGAAACCAGGC	1901.3	1747	381.97	389.17
Serpinb9_exon_07_CGCCTGCACCTCCTTCACAT	4265.7	4338.6	1968.8	1985.9

sgRNA counts, fastq files obtained by NGS and library of the genes are further processed by MAGeCK, where paired comparison between test samples -t (treated group 3 and 4) and -c control samples (untreated group 1 and 2) are compared to identify positively and negatively selected sgRNAs alone or with respect to different treatments. Since the control is specified, the method of normalization is automatic = default median.

2.4 How the RRA scores were determined should also be mentioned in the Materials and Methods.

The RRA scores method detects genes that are ranked consistently better than expected under null hypothesis of uncorrelated inputs and assigns a significance score for each gene. The underlying probabilistic model makes the algorithm parameter-free and robust to outliers, noise and errors. Significance scores also provide a rigorous way to keep only the statistically relevant genes in the final list. This feature is incorporated and automatically generated by MAGeCK software. The top 10 enriched and top 10 depleted genes with input sgRNAs (4 sgRNAs/gene in our study) are then ranked and presented in the line graph as shown in the suppl. Fig 4 and Fig 5.

We have now also performed MAGeCK Flute analysis, which provides several strategies to remove potential biases within sgRNA-level read counts and gene-level beta scores. The downstream analysis of this package includes identification of essential, non-essential, and target-associated genes, as well as biological functional category analysis, pathway enrichment analysis and protein complex enrichment analysis of these genes. The package also visualizes genes in multiple ways to benefit users exploring screening data.¹⁹ Unlike MAGeCK, MAGeCK Flute is designed for top 10 (enriched + depleted) genes based on the expression of all sgRNAs represented within the library. Please see below.

Gratifyingly, MAGeCK and MAGeCK Flute show that SerpinB9 and Adam2 as top-scoring depleted and enriched genes. We have now added the MAGeCK Flute analysis also to the **new Supplemental Fig. 4E**.

- When it comes to the validation of Serpinb9 in Figure 2a and b, the authors observe a clear slowing of tumour growth in Serpinb9 KO samples when looking at the luminoscope. However, the survival of the animals was not affected in the slightest in the KRAS G12D model by the absence of Serpinb9 plus/ minus ACT (Fig2c). In the BRAF V600E experiments it appears from the luminoscope again that the SerpinB9 cells have a slower tumour growth (2B). However, the survival seemed only a little affected by the treatment with ACT (Fig 2d). Interestingly while the luminoscope was strongly reduced in the control animals that was not at all reflected in the survival curve in Fig 2d when comparing control animals treated and untreated. It appears to me that there are some differences in the beginning of the tumour onset, which is not translating into enhanced survival at later stages. When comparing the 9 weeks luminoscope in Fig2 b it appears that there are no malignant cells in the treated Serpinb9 animals, but at week 10 the first animals from that same group dies already. The authors should check whether the tumour cells at time of death of the mice are still deficient for Serpinb9 and explain how no tumour detection by luminescence can lead to death.

First, we agree with the observations of this reviewer and provide the following explanation below and in the text of the manuscript: Figure 2A and 2B clearly shows significantly slower tumor growth starting at tumor onset at either week 3 or week 4 after LV-Cre inhalation. The tumor growth based on BLI was

calculated as the fold change from the initial tumor burden for each and every lung and mouse used in the experiments. Thus, there are indeed “differences in the beginning of the tumour onset”.

While the survival was not affected in the Kras^{G12D} mouse model, the survival in Braf^{V600E} mouse model was slightly improved with the SerpinB9-KO and ACT treatment, which we think just reflects the different biology of these two different models. It is also important to note that ACT-treatment by itself significantly delays death with a median survival of 16 weeks compared to untreated mice with 7 weeks in the KRas (as shown in Fig2C and D). It is also important to note that we are just infusing OT-I cells once at the outset of tumor growth at 4 weeks of age and further optimization with multiple OTI infusions would be needed to maximize ACT efficacy (as done in the clinic), to further improve overall survival. It is conceivable that adoptively transferred OT-1 cells are effective for the first few weeks, but might lose their potency or simply die after 8-10 weeks post-injection, which would explain that there is no added benefit anymore at ~12-14 weeks. It is also important to note that we expect non-edited tumor cells or tumor cells with a Serpinb9-editing other than loss-of-function as well as evolution of tumor cell evasion mechanisms to eventually kick-in, which ultimately would lead to tumor outgrowth and death of the animal. The Luminoscore graph show fold changes of luminescence, but it is important to point out that at 4 weeks of age, all mice do show already quite tumor burden by BLI and histology shows on average 600 lung tumors, which can easily grow out within 2-3 weeks, cause breathing problems and thus necessitate sacrificing the animals. Lastly, as we are showing overall survival in these KM graphs rather than lung-cancer specific survival, it is important to note that some of these mice had to be sacrificed not because of lung tumor burden but due to other health complications. We have added the following sentences to the text:

This is likely due to the single infusion of OT-I cells at the outset of tumor growth and further optimization would be needed to maximize ACT efficacy. However, in the Braf^{V600E}-lung cancer model, ACT-treatment significantly increased the overall survival of mice transduced with sgSerpinb9 compared to control mice (Fig. 2d, Supplementary Fig. 6h). Eventually, unedited Serpinb9-wildtype escapers and other immune-editing mechanisms lead to tumor growth and death. It is also important to mention that the survival analysis shown in Fig. 2C and D depict overall survival and some of the mice had to be sacrificed not because of lung tumor burden but due to other health complications.

We have performed TIDE or ICE analysis on the end-point tumors showing that KRas and Braf mice indeed exhibit very good SerpinB9 depletion. Interestingly, untreated mice showed between 73%-96% knock-out efficiency while some of the ACT-treated mice showed lower KO efficacy (23.5%, 41.3% and 81.5%), potentially indicating that the ACT-treatment more efficiently killed the Serpinb9 knock-out cells allowed escapers to grow out (please see **Suppl Fig. 6 A and B**).

4. The Supp Figure 7c (overall in that Figure the timepoints and ratios between b and c/d were different; why was this?) clearly shows that Serpinb9 sgRNA transduced cells are barely alive when they are used for the killing assay in vitro? 26% (sgSerpinB9) vs 62% (NT) based on the FSC/SSC blots. Would this explain why these cells are more sensitive to killing by the CD8 T cells? Would this also explain why the Serpinb9 sgRNAs were found as a “drop out”, i.e. is generally survival/ growth of Serpinb9 deficient cells negatively impacted?

The graph in Fig 2E represents killing assay performed at 4 hours after coculturing LLCs (targets) with CD8 T cells (effectors), while the graph and the FACS blots in Supp Figure 7b and c (now Supp Figure 7c and d) represents killing assay performed after coculturing LLCs (targets) with CD8 T cells (effectors) over night. We wanted to demonstrate how stable the phenotype is even after an extended period of co-culturing.

While the survival of the cells in SerpinB9 deficient LLCs is indeed lower in the FACS plots of the overnight samples, and could explain that survival/growth of SerpinB9 deficient cells in negatively impacted, % of killing *in vitro* was calculated as the fold change between CFSE+ live cells with the treatment relative to the initial cell population of CFSE+ live cells without the treatment. In addition, we have now added the FACS plots of the 4h time point, which clearly show that the NT and sgSerpinb9 cells show the same viability (please see **new Supp Figure 7b**), while the sgSerpinb9 cells still clearly showing enhanced CD8 killing (Fig 2e). Thus, these data were normalized to reflect the killing capacity of LLCs with the CD8 T cells.

To further substantiate these data, we now also included the FACS blots showing SERPINB9 overexpression in human lung cancer cell lines HT125 and A549. Here we can clearly see that overexpression of SERPINB9 has the opposite effect and protects cancer cells from cytotoxic T-cells. Importantly, overexpression of SERPINB9 did not have any effect on cell survival or growth and as such we think this is strong data indicating that the effect of SERPINB9 in modulating T-cell mediated killing is independent of any potential effect on cell survival/growth (please see new **Suppl Fig. 8cont. f and g**).

5. As far as the Figure legend states the 3 individual dots in Supp Fig 7b and Fig2e represent 1 experiment with 3 technical replicates. In Figure 2f/g (according to Supp Figure 9) that is similar. While these are representative experiment, it would be important to know how often these experiments were repeated?

All killing assay were performed in triplicates (as technical replicates) and were repeated at least twice for each time point (4 hours and overnight) and at different effector:target ratios. Importantly, the finding that genetic ablation of *Serpinb9* sensitizes cells towards cytotoxic T cell killing was highly significantly different and apparent in all experiments using mouse as well as human cell lines.

6. In the second part of the manuscript the authors focus on enriched sgRNAs from their screen and Adam2 – which is only found in the KRAS G12D ACT screen – is pursued. The interesting observation is made that Adam2 KO in KRAS G12D models slows down tumour growth and enhances survival of the mice. However, ACT treatment induces a faster tumour growth and shortens the life span of mice. This is a very surprising observation, and I would have expected that ACT has no impact on the survival of these mice in the absence of Adam2. The authors postulate that an immune suppressive microenvironment is generated in the absence of Adam2, but then - as speculated in the discussion - ACT plus checkpoint inhibitors should reverse the phenotype of accelerated death of mice. This is an important experiment, which should be performed by the authors.

We agree with this reviewer that ACT plus ICB would be an interesting experiment, but the complexity and scale of an autochtonouse CRISPR mouse model with two genotypes (control versus Adam2 KO) and six treatment arms (IgG, ACT+IgG, α PD-L1 ICB, α CTLA4 ICB, ACT and α PD-1 ICB, ACT and α CTLA4 ICB) is simply very challenging in the timeframe of revisions. However, as we think this is an important point, we decided to test how Adam2 impacts ICI treatment using a allograft model by transplanting LLC cells overexpressing Adam2 vs LLC CTRL cells in B6 mice and treated the mice with PDL1 or CTLA4 blocking antibodies (Ab) +/- ACT once the tumor reached 100mm³. PDL1 blocking Ab as well as CTLA4 blocking Ab administered in mice bearing Adam2 tumors showed pronounced and significant tumor reduction with aPDL1 (D20, p=0.002); and aCTLA4 (D20, p=0.04) treatment

respectively, while control tumors showed little or no reduction in growth at that level of aPDL1 or aCTLA4 ICI. In addition, combining PDL1 or CTLA4 inhibition with ACT led to almost complete tumor stasis of Adam2 overexpressing and control tumors, indicating strong cooperative effects (please see new Supp. Fig. 26 d, e).

In light of these new data, we conclude that ICI similarly to ACT results in an effective anti-cancer immune response in Adam2 expressing tumors potentially due to the less chronically exhausted immune milieu, which can be re-invigorated by ICI treatment or ACT.

This reviewer is absolutely correct that in the initial submission we have speculated that ICI response will be reduced by Adam2 overexpression with the argument that ICI response relies on pre-existing immunity. We concur that a complete attenuation of interferon I/II and TNF cytokine signaling, and thus complete block of anti-tumor immunity could have elicited such a response. However, it is important to point out that Adam2 O/E is **reducing** and not completely attenuating interferon I/II and TNF cytokine signaling and anti-tumor immunity. This can be nicely shown when comparing the growth of Adam2 O/E LLC tumors in C57Bl/6 mice versus NSG mice as shown in the original Fig. 4A and B:

The fact that Adam2 O/E tumors are growing out much faster in NSG mice than in C57Bl/6 mice shows that there is still residual anti-tumor immunity despite Adam2 O/E, which can be harnessed by ICB treatment.

Considering these new data showing a similar response to ICI and ACT, we interpret that the dampened immune response leads to a less chronically exhausted immune milieu, which can be re-invigorated by ICI treatment or ACT. We have clarified that Adam2 is reducing (rather than blocking) the inflammatory tumor immune milieu added this new vignette to the discussion on page 15/16:

Reviewer #4 (Remarks to the Author): with expertise in CRISPR screens, cancer immunology

In this study, the authors performed mini-pool CRISPR screens in mouse lung cancer models and identified a few regulator genes of the T-cell antitumor immune response. The top hit Serpinb9 corroborated well with the previous data-driven findings from Jiang et al., Nat Med 2018 and this work formally proved Serpinb9's role in an autochthonous murine model. Meanwhile, the other hit Adam2 is an intriguing finding. Although Adam2 may serve as an oncogene, Adam2 does promote the cytotoxicity of adoptively transferred T cells. This result suggested the potential of developing anti-ADAM2 CAR T cells. In general, **this paper is well written. The work is quite systematic. I only have a few suggestions** for the authors to consider.

1. Have the authors compared Adam2 OE tumors upon anti-CTLA4+PDL1 combination and control treatment? The adoptive T-cell transfer depends on a positive Adam2 level. However, will Adam2 OE promote or repress the immune checkpoint blockade (ICB)? ICB may enhance the infiltration of pre-existing antitumor T cells. Such an experiment should formally justify the authors' discussion that ICB won't work in ADAM2-high tumors.

We want to thank this reviewer and note that the same question and insight has been raised by two additional reviewers of the MS. Hence, this experiment was important to gain and insight into the function of Adam2 with/without ICB.

To test how Adam2 impacts ICB treatment, we transplanted LLC cells overexpressing Adam2 vs LLC CTRL cells in B6 mice and treated the mice with PDL1 or CTLA4 blocking antibodies (Ab) once the tumor reached 100mm³. PDL1 blocking Ab as well as CTLA4 blocking Ab administered in mice bearing Adam2 tumors showed pronounced and significant tumor reduction with aPDL1 (D20, p=0.002); and aCTLA4 (D20, p=0.04) treatment respectively, while control tumors showed little or no reduction in growth at that level of aPDL1 or aCTLA4 ICI. In addition, combining PDL1 or CTLA4 inhibition with ACT led to almost complete tumor stasis of Adam2 overexpressing and control tumors, indicating strong cooperative effects (please see new **Suppl. Fig. 26 d, e**).

This data is virtually the same as observed with OT-I ACT, where OT-I ACT had a strong effect on Adam2 overexpressing LLC tumors (please see original Figure 5B).

In light of this data, we conclude that ICI similarly to ACT results in an effective anti-cancer immune response in Adam2 expressing tumors potentially due to the less chronically exhausted immune milieu, which can be re-invigorated by ICI treatment or ACT. We have clarified that Adam2 is reducing (rather than blocking) the inflammatory tumor immune milieu, added this new vignette to the discussion on page 15/16 and changed our discussion that 'ICB won't work in ADAM2-high tumors' accordingly.

2. In the 2nd Introduction paragraph, claiming the ICB response rate is less than 13% is not correct, as some tumor types, such as Hodgkin's lymphoma, have a much higher response fraction. Please specify the tumor types and treatment settings while claiming such this low number.

This statement is now corrected, and we corrected/added that ICB results in very high response rate in tissue-specific cancers, such as Hodgkin's lymphoma, with high 66.3% and 87% ORR in the Checkmate 205 and 039 trial, respectively.

3. In the 2nd Introduction paragraph, please add citations in the sentence introducing different CAR T therapies in NSCLC (e.g., MSLN, MUC1, NY-ESO-1, GPC3, PSCA, EGFR, ROR1, HER2, PDL1).

Thank you for noticing and citations are now added to the 2nd introduction paragraph.

4. Page 14, paragraph 2: The author's name of this previous study is Peng Jiang, instead of Pen Jiang. Please correct the typo.

Thank you for noticing again, the name of the author of the manuscript "Signatures of T-cell dysfunction and exclusion predict cancer immunotherapy response" is now corrected.

5. (Optional), The ADAM2 function on promoting ACT efficacy is still not sufficiently studied. The current data in the last result section is mostly phenotypic without disclosing further mechanisms. A transcriptomics profiling in bulk tumors was limited by the loss of cellular identity. The authors may consider single-cell RNA-seq or spatial transcriptomics Visium platforms to further explore this phenomenon. However, I understand the difficulty of figuring out underlying mechanisms. Thus, these suggestions are optional for the current revision.

Thank you for suggesting experiment to delineate the mechanism of Adam2, and as eluded to it will take some time to figure out the precise mechanism of Adam2. However, to provide more mechanistic insights how Adam2 functions to promote ACT efficacy, we have now performed several additional profiling studies:

1, In the original manuscript, we performed RNAseq experiments on LLC allograft tumors overexpressing Adam2 compared to vector only control tumors. This revealed a significant reduction in cellular responses to IFN γ , IFN α /b and TNF α and reduced immune responses associated with reduced T cells exhaustion and reduced MHC-I presentation and antigen processing, which nicely explains why Adam2-overexpressing tumors grow out so aggressively in immune-competent mice, but do not exhibit a difference in growth dynamics in immune-deficient NSG mice compared to control tumors.

We now performed additional bulk RNAseq experiments in autochthonous Adam2 knock-out lung tumors compared to control tumors. This revealed the exact opposite phenotype with significantly increased IFN γ and TNF α cytokine expression (as well as increased expression of other TNF and CXCL and CCL cytokines), increased MHC-I presentation and increased T-cell exhaustion marked by increased expression of Lag3, Pd-1, Tigit, and Pd-L1/2, findings that were validated by RT-PCR. To start elucidating the cellular compartments involved, we are also now showing increased expression of Pd-1, Tigit, Lag3 and Inf γ on CD8 T-cells infiltrating Adam2 knock-out tumors using quantitative RT-PCR and FACS analysis (please see new Fig. 3g and h, new Supplementary Fig. 14a-c and 15a-c, and new Supplementary Table 4).

2, As this reviewer pointed out, there RNAseq efforts provide only limited insights into the mechanism how Adam2 affects the cellular composition and activation of cellular compartments within Adam2 knock-out or overexpressing tumors. As suggested by this reviewer, we have now also performed CyTOF analysis and found a significant increase ($p=0.04$) of CD11c⁺ MHC2⁺ cells among CD45⁺ TILs in ADAM2 O/E tumors (**please see new Fig. 5h**). We also found a trend towards more CD64⁺ CD11c⁺ Macs among Sirpa⁺ Ly6G⁻ cells, suggesting a higher degree of macrophage activation in the ADAM2 O/E tumors. Cytof analysis also confirmed the significant downregulation of PD1 by CD8⁺ TCRb⁺ T cells in the Adam2 O/E tumors (**please see new Fig. 5g, h**):

However, our CyTOF analysis was somewhat limited by the availability of a validated mouse immune cell antibody panel and thus we want to note that a detailed mechanistic study of Adam2 within the tumor microenvironment will be the scope of the follow-up manuscript(s).

Overall, our new data further strengthen our hypothesis that Adam2 modulates type I and II IFN as well as TNF α signalling and leads to a less exhausted TME. To help guide readers through our data and our interpretation, we have now also added a schematic as Fig. 28 outlining our results and working hypothesis how ADAM2 might function as an oncogene and work in facilitating ACT and ICB:

Fig. S38 Proposed functions of ADAM2

A, Adam2 is absent in normal lung. B, Adam2 is aberrantly expressed in KRas-mutant lung tumors and functions as an oncogene by reducing IFN/TNF-signaling, reducing MHC-I presentation. As such, Adam2 functions as an immune-suppressant in this setting. Forced expression of ADAM2 in LLC cells corroborated these findings and also triggered a less exhausted TME marked by reduced IFN/TNF-signaling, reduced MHC-presentation and rapid outgrowth of tumors (in an immune-system dependent manner).

C, ACT-treated KRas-mutant lung tumors exhibit further elevated Adam2 expression levels and a good response to adoptive transfer of OT1 T-cells and immune checkpoint blockade (ICB). Forced expression of Adam2 in LLC cells treated with ACT or ICB showed that Adam2 overexpression indeed sensitizes to cancer immunotherapy marked by significantly elevated expression levels of granzymes and perforin and reduced expression of PD-1 on infiltrating Ag-specific cytotoxic T-cells.

D, Genetically ablating Adam2 in KRas-mutant lung tumors leads to increased IFN/TNF signaling and MHC presentation, increased CD8 cells and decreased protumoral M2 macrophages and overall better immune control and less tumor burden.

E, Loss of Adam2 blocks ACT and ICB: The prolonged IFN/TNF-signaling observed in KRas-mutant Adam2 KO lung tumors promotes T-cell exhaustion and blocks cytotoxicity of ex vivo or ICB-activated T-cells marked by increased expression of PD-1, PD-L1, TIGIT and LAG3 as well as INF and TNFs.

Editorial Note: Lung picture was generated by courtesy of Ella Fific.

References:

- 1 Benci, J. L. *et al.* Opposing Functions of Interferon Coordinate Adaptive and Innate Immune Responses to Cancer Immune Checkpoint Blockade. *Cell* **178**, 933-948 e914, doi:10.1016/j.cell.2019.07.019 (2019).
- 2 Dhainaut, M. *et al.* Spatial CRISPR genomics identifies regulators of the tumor microenvironment. *Cell* **185**, 1223-1239 e1220, doi:10.1016/j.cell.2022.02.015 (2022).
- 3 Teijaro, J. R. *et al.* Persistent LCMV infection is controlled by blockade of type I interferon signaling. *Science* **340**, 207-211, doi:10.1126/science.1235214 (2013).
- 4 Wilson, E. B. *et al.* Blockade of chronic type I interferon signaling to control persistent LCMV infection. *Science* **340**, 202-207, doi:10.1126/science.1235208 (2013).
- 5 Ng, C. T. *et al.* Blockade of interferon Beta, but not interferon alpha, signaling controls persistent viral infection. *Cell Host Microbe* **17**, 653-661, doi:10.1016/j.chom.2015.04.005 (2015).
- 6 Tumeh, P. C. *et al.* PD-1 blockade induces responses by inhibiting adaptive immune resistance. *Nature* **515**, 568-571, doi:10.1038/nature13954 (2014).
- 7 Minn, A. J. & Wherry, E. J. Combination Cancer Therapies with Immune Checkpoint Blockade: Convergence on Interferon Signaling. *Cell* **165**, 272-275, doi:10.1016/j.cell.2016.03.031 (2016).
- 8 Benci, J. L. *et al.* Tumor Interferon Signaling Regulates a Multigenic Resistance Program to Immune Checkpoint Blockade. *Cell* **167**, 1540-1554 e1512, doi:10.1016/j.cell.2016.11.022 (2016).
- 9 Manguso, R. T. *et al.* In vivo CRISPR screening identifies Ptpn2 as a cancer immunotherapy target. *Nature* **547**, 413-418, doi:10.1038/nature23270 (2017).
- 10 Lawson, K. A. *et al.* Functional genomic landscape of cancer-intrinsic evasion of killing by T cells. *Nature* **586**, 120-126, doi:10.1038/s41586-020-2746-2 (2020).
- 11 Dubrot, J. *et al.* In vivo CRISPR screens reveal the landscape of immune evasion pathways across cancer. *Nat Immunol* **23**, 1495-1506, doi:10.1038/s41590-022-01315-x (2022).
- 12 Jackson, E. L. *et al.* Analysis of lung tumor initiation and progression using conditional expression of oncogenic K-ras. *Genes Dev* **15**, 3243-3248, doi:10.1101/gad.943001 (2001).
- 13 Dankort, D. *et al.* A new mouse model to explore the initiation, progression, and therapy of BRAFV600E-induced lung tumors. *Genes Dev* **21**, 379-384, doi:10.1101/gad.1516407 (2007).
- 14 Jin, Q. *et al.* Rapid flow cytometry-based assay for the evaluation of gammadelta T cell-mediated cytotoxicity. *Mol Med Rep* **17**, 3555-3562, doi:10.3892/mmr.2017.8281 (2018).
- 15 Jedema, I., van der Werff, N. M., Barge, R. M., Willemze, R. & Falkenburg, J. H. New CFSE-based assay to determine susceptibility to lysis by cytotoxic T cells of leukemic precursor cells within a heterogeneous target cell population. *Blood* **103**, 2677-2682, doi:10.1182/blood-2003-06-2070 (2004).
- 16 Yao, J. *et al.* Human double negative T cells target lung cancer via ligand-dependent mechanisms that can be enhanced by IL-15. *J Immunother Cancer* **7**, 17, doi:10.1186/s40425-019-0507-2 (2019).
- 17 Valiathan, R. *et al.* Evaluation of a flow cytometry-based assay for natural killer cell activity in clinical settings. *Scand J Immunol* **75**, 455-462, doi:10.1111/j.1365-3083.2011.02667.x (2012).
- 18 Hart, T. *et al.* Evaluation and Design of Genome-Wide CRISPR/SpCas9 Knockout Screens. *G3 (Bethesda)* **7**, 2719-2727, doi:10.1534/g3.117.041277 (2017).
- 19 Wang, B. *et al.* Integrative analysis of pooled CRISPR genetic screens using MAGeCKFlute. *Nat Protoc* **14**, 756-780, doi:10.1038/s41596-018-0113-7 (2019).

REVIEWERS' COMMENTS

Reviewer #1 (Remarks to the Author):

The authors have done an outstanding job addressing my critique. I have no additional concerns.

Reviewer #2 (Remarks to the Author):

In this revised manuscript, Dervovic and colleagues explore novel modulators of tumor immunity in lung cancer by performing an impressive in vivo CRISPR loss-of-function screen focused on ~600 immunomodulatory genes within both KrasG12D and BrafV600E driven models of lung cancer. The authors completed an impressive amount of work for this manuscript, and the revised version and the authors' responses thoroughly addressed previous comments/concerns/questions. Overall, this work will be of great interest to the larger scientific community.

Reviewer #3 (Remarks to the Author):

The authors have adequately addressed my suggestions and I have no further comments.

Reviewer #4 (Remarks to the Author):

The authors have done a fantastic job of addressing my review comments and further substantiating the mechanistic insights of their study. I would like to see this systematic study published.